# Representation Costs of Linear Neural Networks: Analysis and Design

**Zhen Dai**
Committee on Computational and Applied Mathematics
University of Chicago
Chicago, IL 60637
zhen9@uchicago.edu

**Mina Karzand**
Department of Statistics
University of California, Davis
Davis, CA 95616
mkarzand@ucdavis.edu

**Nathan Srebro**
Toyota Technological Institute at Chicago
Chicago, IL 60637
nati@ttic.edu

## Abstract

For different parameterizations (mappings from parameters to predictors), we study the regularization cost in predictor space induced by $l_2$ regularization on the parameters (weights). We focus on linear neural networks as parameterizations of linear predictors. We identify the representation cost of certain sparse linear ConvNets and residual networks. In order to get a better understanding of how the architecture and parameterization affect the representation cost, we also study the reverse problem, identifying which regularizers on linear predictors (e.g., $l_p$ quasi-norms, group quasi-norms, the $k$-support-norm, elastic net) can be the representation cost induced by simple $l_2$ regularization, and designing the parameterizations that do so.

## 1 Introduction

In a class of parameterized models, penalizing the $l_2$ norm of the parameters induces regularization on function space which can be interpreted as a complexity measure on the class of learned functions. In this paper, we study how different parameterizations induce different complexity measures.

We consider parameterized mappings $f : \mathcal{X} \times \mathbb{R}^p \longrightarrow \mathbb{R}^m$, from input $x \in \mathcal{X}$ and parameters $w \in \mathbb{R}^p$ to predictions $f(x; w)$. We denote the predictor implemented with parameters $w$ by $F(w) : \mathcal{X} \longrightarrow \mathbb{R}^m$ defined as $F(w)(x) := f(x; w)$. Then $\text{image}(F)$ is the set of functions from $\mathcal{X}$ to $\mathbb{R}^m$ which can be obtained from this class of parameterized models. We will use $\mathcal{F}$ to denote a set of functions from $\mathcal{X}$ to $\mathbb{R}^m$.

The $l_2$ regularization on parameters, either explicitly or implicitly, is a common phenomenon. As an example, in deep learning, explicit $l_2$ regularization on parameters (a.k.a. weight decay) improves generalization ([20, 37]). Implicit regularization of $l_2$ norm of parameters appears when we use gradient descent (GD) to train the model ([2, 35, 33, 19, 23, 32, 14, 25, 9, 18]). In particular, GD on homogeneous neural networks with logistic loss implicitly regularizes $l_2$ norm on weights [23, 32].

*The representation cost* ([15, 31, 27]) of a function $g$ in $\text{image}(F)$ under the parametrization $F$ is

$$R_F(g) = \min\{\|w\|_2^2 : F(w) = g\}. \tag{1}$$

Consider learning a predictor $F(w)$ with some loss function $L(\cdot)$ while controlling the $l_2$ norm of parameters $w$ by minimizing

$$\min_{w\in\mathbb{R}^p} L(F(w)) + \lambda\|w\|_2^2. \tag{2}$$

This is clearly equivalent to learning a function $g$ in $\text{image}(F)$ by controlling $R_F(g)$ defined in Eq. (1):

$$\min_{g\in\text{image}(F)} L(g) + \lambda R_F(g). \tag{3}$$

In other words, the representation cost under the parameterization $F$, $R_F(\cdot)$, captures the regularization on function space $\text{image}(F)$ induced by $l_2$ regularization on parameter space. In this paper, we are interested in understanding how different parameterizations, regularize the function space differently. Since GD on homogeneous neural networks with logistic loss implicitly regularizes $l_2$ norm on weights [23, 32], representation cost induced by $l_2$ regularization in weight space captures the implicit regularization in homogeneous models, but not necessarily in non-homogeneous models. Thus, representation costs of predictors parameterized by homogeneous models are arguably more related to implicit regularization, while representation costs of predictors parameterized by both homogeneous and non-homogeneous models are related to explicit regularization. In this paper, we first develop results for homogeneous neural networks. Then we reduce the non-homogeneous neural network to the homogeneous ones by arguing that the asymptotic behavior of its representation cost can be captured by the representation cost of some homogeneous subnetwork of the non-homogeneous network.

One way to motivate the study of representation cost is by considering the popularity of the over-parameterized models, in which the number of parameters is greater than the number of samples. Surprisingly, it has been observed that in the overparameterized regime, interpolative predictor generalizes well [7, 39, 16, 3]. One way to explain this is that although there are many predictors which perfectly fit the training data, gradient based algorithms choose the one with the smallest representation cost ([15, 23]). In these cases, representation cost operates as a regularization in the function space which enables good generalization. Thus, understanding representation cost helps us understand the generalization of the model. In particular, representation cost of predictors induces an ordering on the space of predictors. Since representation cost is determined by the specific parametrization, each parameterization induces an ordering on the function space. This can be interpreted as an *induced complexity measure* of the predictor space, where penalizing the cost is the same as minimizing the complexity.

**Definition 1.1 (Induced complexity measure).** Let $\mathcal{F}$ be a set of functions from $\mathcal{X}$ to $\mathbb{R}^m$ and $\mathcal{F}^*$ be the set of all functions from $\mathcal{F}$ to $\mathbb{R}$. We define the *equivalence relation* in $\mathcal{F}^*$ such that $h_1$ and $h_2 \in \mathcal{F}^*$ (i.e., $h_1, h_2 : \mathcal{F} \to \mathbb{R}$) are equivalent ($h_1 \cong h_2$) if there exists a strictly increasing function $\psi : \mathbb{R} \longrightarrow \mathbb{R}$ $h_1 = \psi \circ h_2$. Let $\mathcal{F}^*/\cong$ be the set of equivalence classes of $\mathcal{F}^*$ under $\cong$. [1]

Given a parameterization $F : \mathbb{R}^p \to \mathcal{F}$, let $\mathcal{F}$ be $\text{image}(F)$. In this case, for each value of parameter $w \in \mathbb{R}^p$, $F(w) \in \mathcal{F}$ is a mapping from $\mathcal{X}$ to $\mathbb{R}^m$. Let the representation cost $R_F$ under parameterization $F$ be as in Eq. (1). We say that an equivalence class $\bar{h} \in \mathcal{F}^*/\cong$ is the *induced complexity measure* of the parameterization $F$ if for any representative (i.e., element) $h$ in class $\bar{h}$, $R_F$ is a strictly increasing function of $h$.

We study the dependence of induced complexity measures on parameterizations from two perspectives: First, given a parameterization $F$, we analyze its representation cost. Second, given a regularizer on function space, we study when and how it can be the induced complexity measure of some parametrizations. In this paper, we start answering these questions by focusing on linear predictors parameterized by linear neural networks. Note that for linear networks with single outputs, the function space being parameterized does not depend on the architecture. Thus, the change in architecture only changes the induced complexity measure. This makes it appealing for highlighting and understanding what the effect of changing the architecture is in changing the induced complexity measure.

In the first part of the paper (Section 3), we identify the representation costs of various architectures. Specifically, we look into fully connected networks and convolutional networks with multiple outputs. In addition, we show how the representation cost of convolutional networks with restricted filter

---

[1]In other words, $\mathcal{F}^*/\cong$ is a partition of $\mathcal{F}^*$ into classes with the following property: given a class $\bar{h} \in \mathcal{F}^*/\cong$, any two elements $h_1, h_2 \in \bar{h}$ are equivalent (i.e. $h_1 = \psi \circ h_2$).

| Architectures (of depth $d$) | Induced Complexity Measures |
| --- | --- |
| Fully Connected Network with multiple outputs | Schatten $2/d$ quasi-norm |
| Diagonal Network with multiple outputs | Matrix $l_{2/d,2}$ quasi-norm |
| Convolutional Network with multiple outputs | Matrix $l_{2/d,2}$ quasi-norm on Fourier domain |
| Residual Network | An interpolation between two quasi-norms |

Table 1: **From Architecture to Induced Complexity Measure**

| Induced Complexity Measure | Conditions | Architectures |
| --- | --- | --- |
| $l_p$ quasi-norms | if and only if $2/p \in \mathbb{N}$ | Diagonal networks |
| $k$-support norms | if and only if $k \in [n]$ | $k$-balanced networks (Fig. 1a) |
| $l_{p,q}$ group quasi-norms | if $2/p, 2/q \in \mathbb{N}$ and $2/p \geq 2/q - 1$ | Group networks (Fig. 1) |
| Elastic nets | None | None |
| $l_{p,q}$ with overlapping groups | None | None |

Table 2: **From Induced Complexity Measure to Architecture**

width changes as their filter width changes. Then, we show that the representation cost of residual networks interpolates between the representation costs of two of their component networks. Finally, we give two characterizations of the representation costs of depth-two neural networks. The results of this part are summarized in Table 1.

In the second part (Section 4), given a few regularizers, we study when and how they can be the induced complexity measure of some architectures. Specifically, we show that $l_p$ quasi-norm can be the induced complexity measure induced by $l_2$ regularization on some linear neural networks if and only if $2/p$ is an integer. Moreover, when $2/p$ is an integer, we characterize all the architectures whose induced complexity measures are $l_p$ quasi-norms. In addition, we design architectures whose induced complexity measures are $k$-support-norm and $l_{p,q}$ group quasi-norms. Then, we show that elastic nets and $l_{p,q}$ quasi-norm with overlapping groups cannot be the induced complexity measure of any linear neural network. On the contrary, we show that there exist homogeneous parametrizations whose induced induced complexity measures are elastic nets and $l_{p,q}$ quasi-norm with overlapping groups. The results of this part are summarized in Table 2. Finally, in the conclusion, we discuss some interesting future directions.

**Further related works**: Some previous work focuses on the expressive power $\text{image}(F)$ of the model [22, 29, 38, 21]. However, as discussed in [26, 24], some other capacity control, different from network size, plays a role in deep learning. This motivates the study of representation cost and its relation to parametrizations.

Representation cost has been studied before under various models. [15] showed that the representation cost of a linear convolutional neural network of depth $d$ is strictly increasing in the Schatten-$2/d$ quasi-norm on the Fourier domain, whereas the representation cost of a linear fully connected network of depth $d$ is strictly increasing in the $l_2$ norm. [31] and [27] studied depth-two fully connected

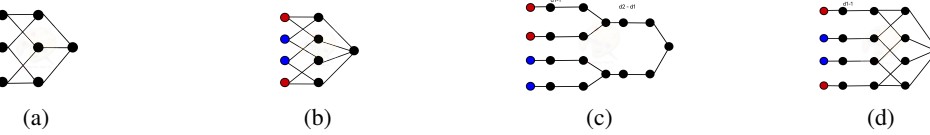

|     (a)     |     (b)     |     (c)     |     (d)     |

Figure 1: $k$-**balanced and Group networks: architectures for** $l_{p,q}$ **quasi-norms.** Figure 1b induces $l_{2,1}$ norm. Figure 1c induces $l_{2/d_2,2/d_1}$ quasi-norm. Figure 1d induces $l_{2/d_1,2/(d_1+1)}$ quasi-norm. In all plots, nodes in same color are in same group.

network with infinite width and ReLU activation. They show that the representation cost of any continuous function depends on the Laplacian of that function.

Parallel to our work, [18] studies the representation cost of linear convolutional neural network with restricted filter width (a.k.a. kernel size) and multiple channels using anlaytical tools from semidefinite programming. In spite of different approaches between our work and [18] the results on CNN with restricted filter width are similar in two papers.

Another line of work studies the relationship between neural networks with $l_2$ regularization on weights and convex optimization problems [28, 12, 11, 30]. In [28, 12, 11, 30], the authors showed that training a neural network with explicit $l_2$ regularization on weights is equivalent to a convex regularized optimization problem in some higher dimensional space. In contrast, motivated by the literature on implicit regularization of gradient descent [2, 35, 33, 19, 23, 32, 14, 25, 9, 18], we looked into the induced regularization of weight decay on function space. Some of the results in [28, 12, 11, 30] are similar to the results in our work. For instance, the results on linear convolutional neural network in [28, 12, 11, 30] suggest that explicit $l_2$ regularization on weight space is related to $l_1$ regularization on the Fourier transform of the predictor. This result was also discovered in [15] and is generalized to multiple output and restricted filter width (a.k.a. kernel size) case in our work and in [18]. On the other hand, we considered other architectures beyond fully connected and convolutional neural networks. For instance, we studied architectures that induce $k$-support norms and architectures that induce $l_{p,q}$ group quasi-norms, which are not included in [28, 12, 11, 30].

As another related work, [36] studied the equivalence between $l_2$ regularization on weights and some sparsity-inducing regularization on the function space for various architectures. They considered the architecture which induces $l_{2/d,2}$ group quasi-norms on the function space, for any $d \in \mathbb{N}$. We also studied a similar question in section 4.1.2. However, we found architectures that induce $l_{p,q}$ group quasi-norms for both the case $p > q$ and the case $p < q$. In addition, we showed that in the case $p < q$, $l_{p,q}$ quasi-norm can be induced by some linear neural network as induced complexity measure if and only if $2/p, 2/q \in \mathbb{N}$ and designed architectures that do so.

## 2 Setup

A parameterized mapping $f : \mathcal{X} \times \mathbb{R}^p \to \mathbb{R}^m$ is *homogeneous* of degree $L$ if $f(x; \lambda w) = \lambda^L f(x; w)$, for all $\lambda > 0$. A feedforward neural network $f_\mathcal{N}$ with weights (parameters) $w = (W_1, \ldots, W_d)$ and activation function $\sigma$ is defined as $f_\mathcal{N}(x; w) = \sigma(W_d \sigma(\ldots W_2 \sigma(W_1 x)))$. Note that when the activation function $\sigma$ is homogeneous (i.e. $\sigma(\lambda x) = \lambda^L \sigma(x)$ for some $L > 0$), the feedforward neural network $f_\mathcal{N}$ is also homogeneous. In particular, with ReLU activation (i.e. $\sigma(x) = \max(0, x)$) or identity activation (i.e. $\sigma(x) = x$), $f_\mathcal{N}$ is homogeneous.

A linear neural network is a neural network with identity activation function. When $\mathcal{X} = \mathbb{R}^n$ and $g$ is a linear function, we identify $g$ with the matrix $\boldsymbol{\beta} \in \mathbb{R}^{m \times n}$ such that $g(x) = \boldsymbol{\beta} x$. In particular, in the case of one-dimensional output space, we identify $g$ with the vector $\beta \in \mathbb{R}^n$ such that $g(x) = \beta^T x$. In this paper, we mainly focus on *linear neural networks*. A more general definition of linear neural networks in terms of a directed acyclic graphs will be useful in our work.

**Definition 2.1.** Let $G = (V, E)$ be a weighted directed acyclic graph, with $n$ sources $v_1, v_2, \ldots v_n$ (i.e. vertices with in-degree zero) and $m$ sinks $u_1, u_2, \ldots, u_m$ (i.e. vertices with out-degree zero). The weight of edge $e \in E$, is denoted by $g(e)$. Given parameters $w \in \mathbb{R}^p$, a function $\psi : E \longrightarrow [p]$ assigns parameters to edges such that $g(e) = w[\psi(e)]$ for all $e \in E$.

The pair $(G, \psi)$ gives a construction of a linear feedforward neural network $\mathcal{N}$ corresponding to a linear predictor $f_\mathcal{N}(\cdot; w) : \mathbb{R}^n \to \mathbb{R}^m$ as follows: Let $\phi(v) \in \mathbb{R}$ be the output flow of the node $v \in V$. Given $x \in \mathbb{R}^n$, let $\phi(v_i) = x[i]$ for all input nodes $i \in [n]$. Then, $\phi(v)$ for other nodes is defined recursively such that the output flow of each node is a weighted sum of its input flow using the weights of the graph: $\phi(v) := \sum_{u:(u,v)\in E} g(u)\phi(u)$. Then $\phi(u)$ for sink nodes $u$ give the linear predictor $f_\mathcal{N}(x; w) = (\phi(u_1), \phi(u_2), \ldots, \phi(u_m))$.

Let $F_\mathcal{N}$ be the parametrization associated with $f_\mathcal{N}$ defined as $F_\mathcal{N}(w)(x) := f_\mathcal{N}(x; w)$.

The depth $d$ of a linear feedforward neural network $(G, \psi)$ is defined as the length of the longest path from the source to the sink. We say that a linear feedforward neural network is homogeneous if every path from the source to the sink have the same length. We say that a linear feedforward neural

network is without shared weights if the map $\psi$ is a bijection. We call $v_1, \ldots, v_n$ the input nodes, $u_1, \ldots, u_m$ the output nodes, and $v \in V$ the nodes of the network $\mathcal{N}$.

Without loss of generality, we assume that for all $v \in V$, there exist a directed path from $v$ to some output node $u_j$ and a directed path from some input node $v_i$ to $v$. Otherwise, removing $v$ would not change $f_{\mathcal{N}}$. For each $v \in V$, let

$$S_v = \{i \in [n] : \text{ there exists a directed path from } v_i \text{ to } v\}. \tag{4}$$

By assumption, for all $v \in V$, $|S_v| \geq 1$.

Note that if a linear feedforward neural network $\mathcal{N}$ is homogeneous, then we can define its $l$th layer $N_l$ as the set of vertices whose distance to any input node $v_i$ is $l$. Note that this is well-defined since the length length of any path from any input node to any output node is constant in homogeneous linear feedforward neural networks.

Let $\mathcal{N}$ be a depth $d$ homogeneous feedforward linear neural network without shared weights. Then, the weights of the edges between the $l$th and $l + 1$th layer of $\mathcal{N}$ can be identified as a matrix $W_l$. In this case, the parameters $w$ is a sequence of matrices $W_1, W_2, \ldots, W_d$ with some fixed sparsity pattern, i.e. $\mathrm{supp}(W_l) = \mathsf{S}_l$ for each $l \in [d]$, where $\mathsf{S}_l$ is determined by $\mathcal{N}$. The parameterized map $f_{\mathcal{N}}$ and the parametrization $F_{\mathcal{N}}$ are given by

$$f_{\mathcal{N}}(x; w) := F_{\mathcal{N}}(w)(x) := (\prod_{l=1}^{d} W_{d-l+1})x, \tag{5}$$

for $x \in \mathbb{R}^n$, where $w = (W_1, W_2, \ldots, W_d)$. Note that Eq. (5) is an equivalent definition of $f_{\mathcal{N}}$ and $F_{\mathcal{N}}$ when $\mathcal{N}$ is homogeneous and without shared weights.

In the rest of the paper, unless stated otherwise, we will use $\mathcal{N}$ to denote a single output depth $d$ homogeneous feedforward linear neural network without shared weights. Note that $f_{\mathcal{N}}$ is homogeneous of degree $d$. Let $N_0 = [n]$ denote the input layer (we identify $v_1, \ldots, v_n$ with $[n]$) and $N_d = \{O\}$ denote the output layer. With slight abuse of notation, let $F_{\mathcal{N}}(w) \in \mathbb{R}^n$ be the vector corresponding to the linear predictor generated by $w$ on $\mathcal{N}$. Let $R_{\mathcal{N}} := R_{F_{\mathcal{N}}}$ denote the representation cost (Eq. 1) under $F_{\mathcal{N}}$. We say that $h$ is the *induced complexity measure* of $\mathcal{N}$ if it is the induced complexity measure of $F_{\mathcal{N}}$ as defined in Def 1.1.

**Notation:** We will use $\beta \in \mathbb{R}^n$ to denote a column vector, and $\beta_i$ or $\beta[i]$ to denote the $i$-th component of $\beta$. We will use $\hat{\beta}$ to denote the discrete Fourier transform of $\beta$. For groups $G_1, G_2 \ldots, G_k \subseteq [n]$, we use the definition of the $l_{p,q}$ group quasi-norm, $\|\beta\|_{p,q} = \left( \sum_{j=1}^{k} \left( \sum_{i \in G_j} |\beta_i|^q \right)^{p/q} \right)^{1/p}$. Unless stated otherwise, $G_1, G_2 \ldots,$ and $G_k$ form a partition of $[n] := \{1, 2, \ldots, n\}$. We will use $\boldsymbol{\beta} \in \mathbb{R}^{m \times n}$ to denote a matrix and $\boldsymbol{\beta}[j, k]$ to denote the element in the $j$-th row and $k$-th column of $\boldsymbol{\beta}$.

## 3 Representation cost analysis

To understand the dependence of induced complexity measure on architectures, we analyze the representation costs of some commonly used architectures. The authors in [15] studied single output fully connected network, diagonal network, and convolutional neural network (CNN) with full filter width. In this section, we first generalize their results to multiple output case. Then, we look into the non-homogeneous residual neural network and observe that its representation cost interpolates between the representation costs of two of its component networks. Finally, we characterize the representation costs of depth-two neural network in two ways.

### 3.1 Multiple output networks

#### 3.1.1 Linear fully connected network

In a linear *fully connected neural network*,

$$F_{FC(n_1, n_2, \ldots, n_{d+1})}(w) = \prod_{i=1}^{d} W_{d+1-i},$$

where $w = (W_1, W_2, \cdots, W_d)$ is the weights of the network. For $i \in [d]$, the matrix $W_i$ is in $\mathbb{R}^{n_{i+1} \times n_i}$ where $n_i \geq \min(m, n)$, $n_1 = n$ and $n_{d+1} = m$. Let $R_{FC(n_1, n_2, \ldots, n_{d+1})} :=$ $R_{F_{FC(n_1, n_2, \ldots, n_{d+1})}}$ be the representation cost under $F_{FC(n_1, n_2, \ldots, n_{d+1})}$ defined in Eq. (1).

**Theorem 1.** *Suppose that $n_i \geq \min(m, n)$ for all $i \in [d+1]$, where $n_1 = n$ and $n_{d+1} = m$. Then, for any $\boldsymbol{\beta} \in \mathbb{R}^{m \times n}$,*

$$R_{FC(n_1, n_2, \ldots, n_{d+1})}(\boldsymbol{\beta}) = d \sum_{i=1}^{r} \sigma_i^{2/d} \cong \|\boldsymbol{\beta}\|_{2/d}^{SC},$$

*where $\sigma_1, \sigma_2, \cdots, \sigma_r$ are the positive singular values of $\boldsymbol{\beta}$ and $\|\boldsymbol{\beta}\|_{2/d}^{SC} := (\sum_{i=1}^{r} \sigma_i^{2/d})^{d/2}$ is the Schatten $2/d$-quasi-norm of $\boldsymbol{\beta}$. In particular, with a single output,*

$$R_{FC(n_1, n_2, \ldots, n_{d+1})}(\beta) = d\|\beta\|_2^{2/d} \cong \|\beta\|_2.$$

The above result is similar to a result in [18]. They studied two layer multiple output convolutional neural network with filter width (a.k.a kernel size) one, and showed that its induced complexity measure is the nuclear norm.

### 3.1.2 Linear diagonal network

In a linear *diagonal network*,

$$F_{DNN}(w) = W_d \prod_{i=2}^{d} \text{diag}(w_{d+1-i}),$$

where $w = (w_1, w_2, \cdots w_{d-1}, W_d)$ is the parameters of a diagonal neural network. For $i \in [d-1]$, $w_i \in \mathbb{R}^n$, and $W_d \in \mathbb{R}^{m \times n}$. So a diagonal network consists of some *diagonal layers* followed by a *fully connected layer*. Let $R_{DNN} := R_{F_{DNN}}$ be the representation cost under $F_{DNN}$ defined in Eq. (1).

**Theorem 2.** *For any $\boldsymbol{\beta} = (\beta^{(1)}, \beta^{(2)}, \cdots, \beta^{(n)}) \in \mathbb{R}^{m \times n}$,*

$$R_{DNN}(\boldsymbol{\beta}) = d \sum_{i=1}^{n} \left\| \beta^{(i)} \right\|_2^{2/d} \cong \|\boldsymbol{\beta}\|_{2/d, 2},$$

*where $\|\boldsymbol{\beta}\|_{2/d, 2} := (\sum_{i=1}^{n} \left\| \beta^{(i)} \right\|_2^{2/d})^{d/2}$ is the matrix $l_{2, 2/d}$ quasi-norm. In particular, with a single output*

$$R_{DNN}(\beta) = d\|\beta\|_{2/d}^{2/d} \cong \|\beta\|_{2/d}.$$

### 3.1.3 Linear convolutional neural network (CNN)

In a linear *Convolutional neural network* (CNN) with filter width $q$, the parameters $w = (w_1, w_2, \cdots w_{d-1}, W_d)$, where $w_i \in \mathbb{R}^q \times \{0\}^{n-q}$ and $W_d \in \mathbb{R}^{m \times n}$. Let $h_i \in \mathbb{R}^n$ be the outputs of the nodes in the $i$th layer. For $i \in [d-1]$, the transformation from the $i$th layer to the $i+1$th layer is given by $h_{i+1}[j] = \frac{1}{\sqrt{n}} \sum_{k=1}^{n} w_{i+1}[k] h_i[(j+k-1) \mod n] =: (w_{i+1} \circledast h_i)[j]$. The last layer is fully connected and $h_d = W_d h_{d-1}$. Then, the linear map is given by $f_{CNN(q)}(w, x) = F_{CNN}(w)(x) = W_d(w_{d-1} \circledast (w_{d-1} \circledast (\ldots w_2 \circledast (w_1 \circledast x) \ldots)))$. Equivalently,

$$F_{CNN(q)}(w) = \prod_{i=1}^{d} W_{d+1-i},$$

where for each $i \in [d-1]$, $w_i[0] := w_i[n]$ and $W_i[j, k] = w_i[(k-j+1) \mod n]/\sqrt{n}$ is the circulant matrix with respect to $w_i$. Let $R_{CNN(q)}(\boldsymbol{\beta}) := R_{F_{CNN(q)}}(\boldsymbol{\beta})$ be the representation cost under $F_{CNN(q)}$ defined in Eq. (1) filter width $q$.

Let $\mathsf{F} \in \mathbb{C}^{n \times n}$ be the discrete Fourier transform matrix defined by $\mathsf{F}[j, k] = \frac{1}{\sqrt{n}} \omega_n^{(j-1)(k-1)}$, where $\omega_n = e^{2\pi i/n}$. For $\boldsymbol{\beta} \in \mathbb{R}^{m \times n}$, let $\hat{\boldsymbol{\beta}} := \boldsymbol{\beta} \mathsf{F}$.

**Theorem 3.** *For any $\boldsymbol{\beta} \in \mathbb{R}^{m \times n}$, let $\hat{\boldsymbol{\beta}} := \boldsymbol{\beta}\mathsf{F}$ and $\hat{\beta}^{(i)}$ be the i-th column of $\hat{\boldsymbol{\beta}}$. Then,*

$$R_{CNN(n)}(\boldsymbol{\beta}) = d \sum_{i=1}^{n} \left\| \hat{\beta}^{(i)} \right\|_2^{2/d} \cong \left\| \hat{\boldsymbol{\beta}} \right\|_{2/d,2},$$

*where $\left\| \hat{\boldsymbol{\beta}} \right\|_{2/d,2} := (\sum_{i=1}^{n} \left\| \hat{\beta}^{(i)} \right\|_2^{2/d})^{d/2}$ is the matrix $l_{2,2/d}$ quasi-norm. In particular, with a single output*

$$R_{CNN(n)}(\beta) = d \left\| \hat{\beta} \right\|_{2/d}^{2/d} \cong \left\| \hat{\beta} \right\|_{2/d}.$$

The same result for $d = 2$ was also discovered in [18].

Results on linear CNN with restricted filter width $q < n$ and some variations of CNN such as CNN with sum pooling and CNN with multiple channels can be found in the supplementary materials.

## 3.2 Linear non-homogeneous residual neural networks

Let $\mathcal{N}$ be a linear homogeneous feedforward neural network without shared weights. Suppose that each hidden layer of $\mathcal{N}$ contains $n$ nodes. Let $d$ be the depth of $\mathcal{N}$. Let $I_1, I_2, \ldots, I_k \subseteq [d]$ such that $|I_j| = d_j$ for each $j \in [k]$. For each $j \in [k]$, let $I_j = \{j_1, j_2, \ldots, j_{d_j}\}$, where $j_1 < j_2 < \cdots < j_{d_j}$. For each $w = (W_1, W_2, \ldots, W_d)$ and $j \in [k]$, let

$$F_{\mathcal{N}_j}(w) := \prod_{i=1}^{d_j} W_{j_{(d_j - i + 1)}} \quad \text{and} \quad F_{\mathcal{N}_{ResNet}}(w) := \sum_{j=1}^{k} F_{\mathcal{N}_j}(w). \tag{6}$$

be the parameterization for a *residual neural network* (ResNet). Let $R_{ResNet} := R_{F_{\mathcal{N}_{ResNet}}}$ be the representation cost under $F_{\mathcal{N}_{ResNet}}$, and $R_j := R_{F_{\mathcal{N}_j}}$ be the representation cost under $F_{\mathcal{N}_j}$ for each $j \in [k]$.

**Theorem 4.** *Suppose that $d_1 < d_2 < \cdots < d_k$. Then, $R_{ResNet}(\lambda\beta)/R_1(\lambda\beta) \longrightarrow 1$ as $\lambda \to 0$, and $R_{ResNet}(\lambda\beta)/R_k(\lambda\beta) \longrightarrow 1$ as $\lambda \to \infty$.*

Note that the model considered here includes sum of homogeneous models without shared weights, which is studied in [32]. A concrete example can be found in the supplementary materials.

## 3.3 Depth two neural networks

In this section, we characterize the representation costs of depth two homogeneous feedforward neural networks in two ways. We will use these two characterizations to find architectures that induce $k$-support norms [6] and $l_{2,1}$ norms as induced complexity measures.

Let $d = 2$. Note that the definition given in Eq. (4) becomes $S_h = \{i \in N_0 : (i, h) \in E\}$, for each $h \in N_1 =: N_H$.

**Lemma 5.** *For a depth-two linear homogeneous feedforward neural network $\mathcal{N}$ without shared weights, $R_{\mathcal{N}}(\beta) = 2 \min\{\sum_{h \in N_H} \|v_h\|_2 : \operatorname{supp}(v_h) \subseteq S_h, \sum_{h \in N_H} v_h = \beta\}$.*

In the above lemma, each vector $v_h$ corresponds to the linear predictor generated by the part of the network which includes the hidden node $h$ and its neighbors. Lemma 5 implies that $R_{\mathcal{N}}(\cdot)$ is a norm. Let $R_{\mathcal{N}}^*(\cdot)$ be its dual norm. Now, we give a characterization of $R_{\mathcal{N}}^*(\cdot)$.

**Lemma 6.** *For a depth-two linear homogeneous feedforward neural network $\mathcal{N}$ without shared weights, $R_{\mathcal{N}}^*(\beta) = \frac{1}{2} \max\{(\sum_{i \in S_h} \beta_i^2)^{1/2} : h \in N_H\}$.*

By Lemma 6, if there exists $h_1, h_2 \in N_H$ such that $h_1 \neq h_2$ and $S_{h_1} \subseteq S_{h_2}$, then removing $h_1$ from $\mathcal{N}$ would not change the representation cost since $\sum_{i \in S_{h_1}} \beta_i^2 \leq \sum_{i \in S_{h_2}} \beta_i^2$.

# 4 Parameterization design

In order to further understand the dependence of induced complexity measure on architectures, we study when and how regularizers can be induced as the induced complexity measure by some architectures.

In this section, we study a few regularizers such as $l_p$ quasi-norms, $l_{p,q}$ group quasi-norms with and without overlapping between groups, $k-$support norms, and elastic nets.

## 4.1 Architecture design

First, we design architectures that induce $l_p$ quasi-norms, $l_{p,q}$ quasi-norms without overlapping groups, and $k-$support norms as induced complexity measures respectively.

### 4.1.1 $l_p$ quasi-norms

In this section, we study architectures that induce $l_p$ quasi-norm, which is defined as $\|\beta\|_p = (\sum_{i=1}^n |\beta_i|^p)^{1/p}$, where $\beta \in \mathbb{R}^n$.

**Theorem 7.** *There exists a linear homogeneous feedforward neural network $\mathcal{N}$ without shared weights that induces $l_p$ quasi-norm if and only if $2/p \in \mathbb{N}$. In particular, diagonal network of depth $2/p$ induces $l_p$ quasi-norm.*

It turns out that we can capture all the architectures that induce $l_p$ quasi-norms using a simple combinatorial measure called *mixing depths*. Roughly speaking, for any $S \subseteq [n]$, the mixing depth $M_{\mathcal{N}}(S)$ is the index of the first layer that contains a node $v$ such that $S \subseteq S_v$, where $S_v$ is defined in Eq. (4).

A linear homogeneous feedforward neural network $\mathcal{N}$ without shared weights induces $l_p$ quasi-norm if and only if $M_{\mathcal{N}}(S) = 2/p$, for all $S \subseteq [n], |S| \geq 2$. The details can be found in supplementary materials on mixing depths and proofs are in supplementary materials for $l_p$ quasi-norms.

### 4.1.2 $l_{p,q}$ group quasi-norms

Similar to the previous sections, we want to know if and when $l_{p,q}$ group quasi-norm is the induced complexity measure of $\mathcal{N}$. Remember the definition of of $l_{p,q}$ quasi-norm, $\|\beta\|_{p,q} = \left(\sum_{j=1}^k \left(\sum_{i \in G_j} |\beta_i|^q\right)^{p/q}\right)^{1/p}$, where $G_1, G_2 \ldots G_k$ form a partition of $[n]$.

Unlike the results for $l_p$ quasi-norms, we do not find all the values of $p$ and $q$ such that $l_{p,q}$ group quasi-norms without overlapping groups can be induced by some homogeneous feedforward linear neural networks without shared weights.

**Theorem 8.** *If there exists a linear homogeneous feedforward neural network $\mathcal{N}$ without shared weights that induces $l_{p,q}$ group quasi-norms, then $2/p, 2/q \in \mathbb{N}$. On the other hand, if $2/p, 2/q \in \mathbb{N}$ and $2/p \geq 2/q - 1$, then there exists a linear homogeneous feedforward neural network $\mathcal{N}$ without shared weights that induces $l_{p,q}$ group quasi-norms.*

Next, we will design *group networks* that induce $l_{p,q}$ group quasi-norms. The design of group networks uses insights from *subnetworks*. Roughly speaking, a subnetwork is a restriction of the original network to some input nodes. The induced complexity measure of a subnetwork is tightly related to that of the original network. This relationship, together with the results in Section 4.1.1 inform how certain subnetworks of a group network look like, which indicates certain properties of the group network. The details can be found in supplementary materials.

Group networks consists of some diagonal layers followed by a *grouping layer* and then followed by a diagonal network (Section 3.1.2). Two examples of such networks are in Figures 1c and 1d. The *grouping layer* is the first layer that mixes information from different input nodes. Depending on whether $p < q$ or $p > q$, [2] we define two types of grouping layers:

**Definition 4.1** (**Type I and II Grouping Layers**). For each $i \in [d]$, $N_i$ is a *type I grouping layer* if $N_j$ is diagonal for all $j < i$, $|N_i| = k$, where $k$ is the number of groups, and for each $j \in [k]$, there exists $u \in N_i$ such that $S_u = G_j$.

---

[2] When $p = q$, $l_{p,q}$ quasi-norm becomes $l_p$ quasi-norm which we already studied.

For each $i \in [d]$, $N_i$ is a *type II grouping layer* if $N_j$ is diagonal for all $j < i$, $|N_i| = \prod_{j=1}^{k} |G_j|$, where $k$ is the number of groups, and for each $h \in \prod_{j=1}^{k} G_j$, there exists $u \in N_i$ such that $S_u = h$.

Next, we compute the representation costs of networks with these two types of grouping layers.

For $d_1, d_2 \in \mathbb{N}$ with $d_2 > d_1$, let $\mathcal{N}^{1;d_1,d_2}$ be the architecture with $d_1 - 1$ diagonal layers, followed by a type I grouping layer (Def 4.1), and then followed by a diagonal network of depth $d_2 - d_1$ (Section 3.1.2). See Figure 1c for an example of this kind of *group network*.

**Theorem 9.** *Let $G_1, G_2 \ldots G_k$ be a partition of $[n]$. Let $\beta_{G_j}$ be the projection of $\beta$ on $G_j$. Then for $d_2 > d_1$, $R_{\mathcal{N}^{1;d_1,d_2}}(\beta) = d_2 \sum_{j=1}^{k} \left\| \beta_{G_j} \right\|_{2/d_1}^{2/d_2} = d_2 \|\beta\|_{2/d_2,2/d_1}^{2/d_2} \cong \|\beta\|_{2/d_2,2/d_1}$.*

The same architecture when $d_1 = 1$ is also discovered in [36]. They also showed that in the group network $\mathcal{N}^{1;1,d_2}$, $l_2$ regularization on weights translate to $l_{2/d_2,2}$ regularization in the function space. Our result is stronger than the results in [36] in two ways. First, we found the architecture $\mathcal{N}^{1;d_1,d_2}$, which induces $l_{2/d_2,2/d_1}$ quasi-norms for all $d_1, d_2 \in [n]$, while they only did it for $d_1 = 1$. Second, we proved that these are all the values of $p, q$ such that $l_{p,q}$ group quasi-norms can be induced as induced complexity measures for some linear neural network, when $p < q$.

For $d_1, d_2 \in \mathbb{N}$ with $d_2 > d_1$, let $\mathcal{N}^{2;d_1,d_2}$ denote the architecture consisting of $d_1 - 1$ diagonal layers, followed by a type II grouping layer (Def 4.1), and then followed by a diagonal network of depth $d_2 - d_1$ (Section 3.1.2). Figure 1d is an example of $\mathcal{N}^{2;d_1,d_2}$.

In particular, when $d_1 = 1$ and $d_2 = 2$ (as in Figure 1b), $\mathcal{N}^{2;1,2}$ induces $l_{2,1}$ norm. This can be proved by the dual characterization of representation cost of depth-two networks in Lemma 6 and the fact that $\|\beta\|_{2,1} = \|\beta\|_{2,\infty}^*$.

**Theorem 10.** *When $d_2 = d_1 + 1$, $R_{\mathcal{N}^{2;d_1,d_2}}(\beta) = d_2 \|\beta\|_{2/d_1,2/d_2}^{2/d_2} \cong \|\beta\|_{2/d_1,2/d_2}$.*

This theorem implies that $\mathcal{N}^{2;d_1,d_2}$ induces $l_{2/d_1,2/d_2}$ quasi-norm when $d_2 = d_1 + 1$. Surprisingly, $\mathcal{N}^{2;d_1,d_2}$ does not induce $l_{2/d_1,2/d_2}$ quasi-norm when $d_2 > d_1 + 1$. The details are in supplementary materials.

### 4.1.3 The $k$-support norms

In [6], the $k$-support norm is defined as $\|\beta\|_k^{sp} = \min\{\sum_{I \in \mathcal{G}_k} \|v_I\|_2 : \text{supp}(v_I) \subseteq I, \sum_{I \in \mathcal{G}_k} v_I = \beta\}$, for $k \in [n]$, where $\mathcal{G}_k$ is the set of subsets of $[n]$ of size at most $k$.

To design an architecture which induces $k$-support-norm, we introduce the *$k$-balanced networks*. A two layer neural network is a $k$-balanced network, if it contains $\binom{n}{k}$ nodes in the hidden layer such that for each subset $I \subseteq [n]$ of size $k$, there is a node in the hidden layer which connects to input nodes in $I$. See Figure 1a for an example with $n = 3$ and $k = 2$.

**Theorem 11.** *For any $k \in [n]$, there exists a homogeneous feedforward depth two linear neural network without shared weights that induces $k$-support norm as induced complexity measure. In particular, $k$-balanced network induces $k$-support norm.*

The proof of the above theorem is an application of Lemma 5 which characterizes the representation cost of depth-two networks.

### 4.2 Limitations of homogeneous neural networks

Theorem 8 and Theorem 11 give architectures that induce $l_{p,q}$ quasi-norms and $k$-support norms. Then, it is natural to consider two regularizers related to $k$-support norms and $l_{p,q}$ quasi-norms. Elastic nets [3] is defined as $\|\beta\|_{EN} = \|\beta\|_1 + \alpha \|\beta\|_2$, and $l_{p,q}$ quasi-norms with overlapping groups is defined as in Section 4.1.2 except that $G_1, G_2 \ldots G_k$ might overlap. Contrary to the results of $k$-support norm and $l_{p,q}$ quasi-norms, elastic nets and $l_{p,q}$ quasi-norms with overlapping groups are not induced complexity measure of any architecture $\mathcal{N}$ without shared weights. The detail can be found in supplementary materials.

---

[3]Elastic nets and $k$-support norms are both interpolations between $l_1$ and $l_2$ norms.

Given these negative results, it is natural to look at non-homogeneous residual networks. We use the same definition of residual networks as in Section 3.2. Theorem 4 characterizes the asymptotic behavior of the representation costs of residual networks. The proof of the following theorem uses Theorem 4.

**Theorem 12.** *Suppose that $d_1 < d_2 < \cdots < d_k$. Let $h : \mathbb{R}^n \longrightarrow \mathbb{R}$ be a homogeneous function. If $F_{\mathcal{N}_{ResNet}}$ induces $h$ as induced complexity measure, then $F_{\mathcal{N}_1}$ also induces $h$ as induced complexity measure.*

This theorem implies that the negative results on elastic nets, $l_{p,q}$ quasi-norm with overlapping groups, and $l_p$ quasi-norms when $2/p \notin \mathbb{N}$ still hold even in the case of non-homogeneous residual networks. As a next step looking beyond homogeneous networks, we look into general form of homogeneous parameterizations which might not be associated with any networks. Surprisingly, homogeneous parameterizations can indeed induce elastic nets and $l_{p,q}$ quasi-norms with overlapping groups for all $p, q > 0$, as induced complexity measure. The details are in supplementary materials.

## 5 Conclusion

In this paper, we take the first steps in studying the dependency of induced complexity measure on the choice of parametrization. We do so by analyzing the induced complexity measures of some well-known architectures and designing architectures that induce some common regularizers on linear predictors. These directions are important for two reasons. First, it helps us understand why certain architectures generalize. Second, if we have a desired regularizer in mind, this helps us design an architecture which induces this regularizer as induced complexity measure.

For the first reason, many of the representation costs we study, when used as regularizers in learning problems, have good generalization properties. This includes $l_{p,q}$ group quasi-norms, especially in the context of multi-task or multi-class learning [13, 17], k-support norm [6], elastic net [10, 40], nuclear norm [4, 34, 1, 5], and $l_p$ quasi-norms for $p \leq 1$ in order to promote sparsity [8]. Thus, this existing understanding and analysis, together with the results in our work, explain for the benefit of using the corresponding architectures.

For the second reason, we do not mean designing an architecture from scratch based on a fully specified regularizer (as we do in section 4). Instead, we believe that building out our understanding in this regard can help us with making architectural choices about complex architectures. In these setups, we do not understand the exact representation cost, and cannot write it down and use it explicitly; but we might want to change representation cost or nudge it in particular directions through some modification of the architecture.

Answering these two questions of design and analysis in a broad sense is an important step in understanding generalization and improving our current models. The limitation of our work includes the fact that we are considering only a rather specific set of architectures, and in particular only linear models. So our study is mostly meant to build tools and understanding and set the stage for understanding more complex non-linear models. But non-linear models might behave very differently, and so we should be cautious about how many of our insights carry over.

To move beyond, there are still many unanswered questions for the linear models. For instance, for $p, q \in \mathbb{N}$ such that $2/p < 2/q - 1$, does there exist an architecture that induce $l_{p,q}$ quasi-norm?

Next step would be looking beyond linear predictors. For example, the question of analyzing representation cost for neural networks with non-linear activation functions such as ReLU is an open problem for most architectures. The other possible direction is studying the same questions (analysis and design) for functions with multiple outputs.

## 6 Acknowledgement

Zhen Dai was funded by DARPA (grant number: HR00112190040) and NSF (grant DMS 1854831).

## 7 Funding Transparency Statement

Funding in direct support of this work: DARPA grant (grant number: HR00112190040); NSF grant (grant DMS 1854831).

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
