# A Supplementary materials in Section 3.1: multiple output networks

## A.1 Fully connected networks with multiple outputs

We give a proof of Theorem 1.

**Theorem 1.** *Suppose that $n_i \geq \min(m,n)$ for all $i \in [d+1]$, where $n_1 = n$ and $n_{d+1} = m$. Then, for any $\boldsymbol{\beta} \in \mathbb{R}^{m \times n}$,*

$$R_{FC(n_1,n_2,\ldots,n_{d+1})}(\boldsymbol{\beta}) = d \sum_{i=1}^{r} \sigma_i^{2/d} \cong \|\boldsymbol{\beta}\|_{2/d}^{SC},$$

*where $\sigma_1, \sigma_2, \cdots, \sigma_r$ are the positive singular values of $\boldsymbol{\beta}$ and $\|\boldsymbol{\beta}\|_{2/d}^{SC} := (\sum_{i=1}^{r} \sigma_i^{2/d})^{d/2}$ is the Schatten $2/d$-quasi-norm of $\boldsymbol{\beta}$. In particular, with a single output,*

$$R_{FC(n_1,n_2,\ldots,n_{d+1})}(\beta) = d\|\beta\|_2^{2/d} \cong \|\beta\|_2.$$

Recall that we assumed that $n_i \geq \min(m,n)$ for all $i$.

We first prove a special case when $n_i = m = n$ for all $i \in [d+1]$.

*Special case of Theorem 1.* The idea of the proof is as follows. If we have a minimum cost representation $\prod_{i=1}^{d} W_{d+1-i}$ of $\boldsymbol{\beta}$, then the matrices $W_i$ are aligned in the sense that we can pick a singular value decomposition $W_i = U_i \Sigma_i V_i^T$ for each $i$ such that $U_i = V_{i+1}$ for all $i$. As a result, singular values of the product $\prod_{i=1}^{d} W_{d+1-i}$ equal the product of the singular values of $W_i$s. In addition, the singular values of $W_i$ equal the singular values of $W_j$ for all $i, j$. These observations immediately give the representation cost of $\boldsymbol{\beta}$. To prove this, we will choose a singular value decomposition $W_i = U_i \Sigma_i V_i^T$ for each $W_i$ such that $U_i = V_{i+1}$, and $\Sigma_i = \Sigma_{i+1}$. We prove this in two steps.

First, we will prove the case $d = 2$. We will show that if $W_2 W_1$ is a minimum cost representation of $\boldsymbol{\beta}$, then the singular values of $W_2$ and $W_1$ are the same. In addition, given any singular value decomposition (SVD) of $W_2 = U_2 \Sigma_2 V_2^T$, there exists a SVD of $W_1 = U_1 \Sigma_1 V_1^T$ such that $U_1 = V_2$.

Second, we will use this observation to prove the general depth case. We will show that if $\prod_{i=1}^{d} W_{d+1-i}$ is a minimum cost representation of $\boldsymbol{\beta}$, then any two adjacent matrices $W_{i+1}$ and $W_i$, form a minimum cost representation of their product matrix $W_{i+1} W_i$. Then, by the result in $d = 2$ case, given any SVD of $W_{i+1} = U_{i+1} \Sigma_{i+1} V_{i+1}^T$, there exists a SVD of $W_i = U_i \Sigma_i V_i^T$ such that $U_i = V_{i+1}$. We will use this observation to pick a SVD for each $W_i$. Using this, we will show that the singular values of $\boldsymbol{\beta}$ are the products of the corresponding singular values of the $W_i$s and this immediately gives the representation cost of $\boldsymbol{\beta}$.

**Depth two case:** Let $\boldsymbol{\beta}$ be a square matrix in $\mathbb{R}^{n \times n}$. Let $\boldsymbol{\beta} = U \Sigma V^T$ be the SVD of $\boldsymbol{\beta}$. We begin with the case $d = 2$. Suppose that $\boldsymbol{\beta} = W_2 W_1$, where $W_2, W_1 \in R^{n \times n}$ such that $R_{FC}(\boldsymbol{\beta}) = \|W_2\|_F^2 + \|W_1\|_F^2$, i.e., $W_2 W_1$ is a minimum cost representation of $\boldsymbol{\beta}$. Let

$$A_2 = U^T W_2, \quad \text{and} \quad A_1 = W_1 V. \tag{7}$$

Then $A_2 A_1 = \Sigma$, $\|A_2\|_F = \|W_2\|_F$ and $\|A_1\|_F = \|W_1\|_F$. Thus, $R_{FC}(\boldsymbol{\beta}) = \|W_2\|_F^2 + \|W_1\|_F^2 = \|A_2\|_F^2 + \|A_1\|_F^2$ and

$$
\begin{aligned}
R_{FC}(\boldsymbol{\beta}) &= \|A_2\|_F^2 + \|A_1\|_F^2 \\
&= \mathrm{Tr}\big(A_2 A_2^T\big) + \mathrm{Tr}\big(A_1^T A_1\big) \\
&\overset{(a)}{\geq} 2\sqrt{\mathrm{Tr}\big(A_2 A_2^T\big) \mathrm{Tr}\big(A_1^T A_1\big)} \\
&\overset{(b)}{\geq} 2\,\mathrm{Tr}(A_2 A_1) \\
&= 2\,\mathrm{Tr}(\Sigma) \\
&= 2\sum_{i=1}^{r} \sigma_i,
\end{aligned}
\tag{8}
$$

where in $(a)$ we used AM-GM inequality and in $(b)$ we used Cauchy inequality. For the equality to hold, it must be the case that $\|A_2\|_F = \|A_1\|_F$ (AM-GM in $(a)$) and $A_2 = \lambda A_1^T$ for some $\lambda \in \mathbb{R}$ (Cauchy in $(b)$). Thus,

$$A_2 = A_1^T, \quad \text{or} \quad A_2 = -A_1^T. \tag{9}$$

Let $W_2 = U_2\Sigma_2 V_2^T$ be a SVD of $W_2$. Then, by Eq. (7) and Eq. (9), $A_1 = \pm A_2^T = \pm(U^T W_2)^T = \pm(U^T U_2\Sigma_2 V_2^T)^T = V_2\Sigma_2(\pm U_2^T U)$. By Eq. (7), $W_1 = A_1 V^T = V_2\Sigma_2(\pm V U^T U_2)^T$ is a SVD of $W_1$. Thus, the singular values of $W_2$ and $W_1$ are the same.

**General depth case:** Now, we turn to the general depth case. Let $\boldsymbol{\beta} = \prod_{i=1}^{d} W_{d+1-i}$ such that $R_{FC}(\boldsymbol{\beta}) = \sum_{i=1}^{d} \|W_i\|_F^2$. Let $E_i = W_{i+1}W_i$. Then $R_{FC}(E_i) = \|W_{i+1}\|_F^2 + \|W_i\|_F^2$, since otherwise there is a representation of $\boldsymbol{\beta}$ with a smaller cost by changing $W_{i+1}$ and $W_i$ to some other matrices whose product is still $E_i$ and keep other $W_j$ the same.

Next, we pick a SVD for each $W_i$ as follows. Let $W_d = U_d\Sigma_d V_d^T$ be an arbitrary SVD of $W_d$. By the argument in the previous paragraph (by considering $\boldsymbol{\beta} = U_d = W_d W_{d-1}$), there exists a SVD $W_{d-1} = U_{d-1}\Sigma_{d-1}V_{d-1}^T = V_d\Sigma_d V_{d-1}^T$, for some $V_{d-1} \in \mathbb{R}^{n \times n}$. Then, we apply the same argument to $W_{d-2}$ and so on. At the end, we would get a SVD $W_i = U_i\Sigma_i V_i^T$ for each $W_i$ such that $U_i = V_{i+1}$, and $\Sigma_i = \Sigma_{i+1}$. Thus,

$$\boldsymbol{\beta} = \prod_{i=1}^{d} W_{d+1-i} = U_d\Sigma_d^d V_1^T. \tag{10}$$

Let $\sigma_1', \ldots, \sigma_r'$ be the singular values of $\Sigma_d$. By Eq. (10), the singular values of $\boldsymbol{\beta}$ are:

$$\sigma_j = \sigma_j'^d \tag{11}$$

for all $j$. Thus,

$$R_{FC}(\boldsymbol{\beta}) = \sum_{i=1}^{d} \|W_i\|_F^2 = \sum_{i=1}^{d}(\sum_{j=1}^{r} \sigma_j'^2) = d\sum_{j=1}^{r} \sigma_j^{2/d}. \tag{12}$$
$\square$

Now, we give a proof of Theorem 1 for the general case ($n_i$ is not a constant).

*Proof of Theorem 1.* We will first prove that $R_{FC}(\boldsymbol{\beta}) \geq d\sum_{j=1}^{r} \sigma_j^{2/d}$ and then show that $R_{FC}(\boldsymbol{\beta}) \leq d\sum_{j=1}^{r} \sigma_j^{2/d}$.

To prove $R_{FC}(\boldsymbol{\beta}) \geq d\sum_{j=1}^{r} \sigma_j^{2/d}$, we will consider a super-network of the original fully connected network. This super-network have constant widths (the number of nodes in each layer is the same) and thus we can compute the representation cost of any matrix in this network using results we just proved. We will then show that the representation cost of some matrix $\tilde{\boldsymbol{\beta}}$ in this super-network is always a lower bound for the representation cost of $\boldsymbol{\beta}$ in the original network.

To prove $R_{FC}(\boldsymbol{\beta}) \leq d\sum_{j=1}^{r} \sigma_j^{2/d}$, we will consider a subnetwork of the original network and give a representation $\prod_{i=1}^{d} W_{d+1-i}$ of $\boldsymbol{\beta}$ in this subnetwork, whose cost is $d\sum_{j=1}^{r} \sigma_j^{2/d}$.

**Lower bound:** Let $M = \max\{m, n, n_1, n_2, \ldots, n_{d+1}\}$ be the maximum width of the network. We consider a super-network of the original fully connected network by adding nodes to each layer (including the input and output layers) such that each layer have exactly $M$ nodes and then add edges to make the network fully connected. Let $\mathcal{N}$ denote this network. Let $\tilde{\boldsymbol{\beta}} \in \mathbb{R}^{M \times M}$ be defined as

$$\tilde{\boldsymbol{\beta}} = \begin{pmatrix} \boldsymbol{\beta} & 0 \\ 0 & 0 \end{pmatrix}. \tag{13}$$

Let $R_{\mathcal{N}}(\tilde{\boldsymbol{\beta}})$ be the representation cost of $\tilde{\boldsymbol{\beta}}$ under $\mathcal{N}$. For any given weights $w$ on the original fully connected network such that $F_{FC}(w) = \boldsymbol{\beta}$ and $\|w\|_2^2 = R_{FC}(\boldsymbol{\beta})$, we can get a weights $w'$ on $\mathcal{N}$ by putting zeros to the edges not in the original networks. Then, $F_{\mathcal{N}}(w) = \tilde{\boldsymbol{\beta}}$ and $\|w'\|_2^2 = \|w\|_2^2$. Thus,

$$R_{FC}(\boldsymbol{\beta}) \geq R_{\mathcal{N}}(\tilde{\boldsymbol{\beta}}). \tag{14}$$

By the proof of the special case of Theorem 1, we have

$$R_{\mathcal{N}}(\tilde{\boldsymbol{\beta}}) = d \sum_{j=1}^{r} \tilde{\sigma}_j^{2/d}, \tag{15}$$

where $\tilde{\sigma}_j$ are the singular values of $\tilde{\boldsymbol{\beta}}$. However, the non-zero singular values of $\boldsymbol{\beta}$ and $\tilde{\boldsymbol{\beta}}$ are the same. To see this, let $\boldsymbol{\beta} = U\Sigma V^T$ be a singular value decomposition of $\boldsymbol{\beta}$. Then,

$$\tilde{\boldsymbol{\beta}} = \begin{pmatrix} \boldsymbol{\beta} & 0 \\ 0 & 0 \end{pmatrix} = \begin{pmatrix} U\Sigma V^T & 0 \\ 0 & 0 \end{pmatrix} = \begin{pmatrix} U & 0 \\ 0 & \mathbb{I} \end{pmatrix} \begin{pmatrix} \Sigma & 0 \\ 0 & 0 \end{pmatrix} \begin{pmatrix} V^T & 0 \\ 0 & \mathbb{I} \end{pmatrix} \tag{16}$$

is a singular value decomposition of $\tilde{\boldsymbol{\beta}}$. Thus, by Eq. (14) and Eq. (15),

$$R_{FC}(\boldsymbol{\beta}) \geq d \sum_{j=1}^{r} \sigma_j^{2/d}. \tag{17}$$

**Upper bound:** Let $r = \text{rank}(\boldsymbol{\beta})$. Clearly, $r \leq \min(m, n)$. We extract a subnetwork $\mathcal{N}'$ from the original fully connected network as follows. We remove all but $r$ nodes in each hidden layer (do not include input and out layers). Let $\mathcal{N}'$ be the resulting network. Let $R_{\mathcal{N}'}(\boldsymbol{\beta})$ be the representation cost of $\boldsymbol{\beta}$ in $\mathcal{N}'$. Since $\mathcal{N}'$ is a subnetwork of the original fully connected network, we have

$$R_{FC}(\boldsymbol{\beta}) \leq R_{\mathcal{N}'}(\boldsymbol{\beta}). \tag{18}$$

(Since $m \neq n$, the proof is not finished yet.) Let $\boldsymbol{\beta} = \tilde{U}\tilde{\Sigma}\tilde{V}^T$ be a reduced singular value decomposition of $\boldsymbol{\beta}$, where $\tilde{\Sigma} \in \mathbb{R}^{r \times r}$, $\tilde{U} \in \mathbb{R}^{m \times r}$, and $\tilde{V} \in \mathbb{R}^{n \times r}$. Now, take $W_d = \tilde{U}\tilde{\Sigma}^{1/d}, W_1 = \tilde{\Sigma}^{1/d}\tilde{V}^T$ and $W_i = \tilde{\Sigma}^{1/d}$ for all $i \notin \{1, d\}$. Then, $\boldsymbol{\beta} = \prod_{i=1}^{d} W_{d+1-i}$ and $\sum_{i=1}^{d} \|W_i\|_F^2 = d\sum_{j=1}^{r} \sigma_j^{2/d}$. Thus,

$$R_{FC}(\boldsymbol{\beta}) \leq R_{\mathcal{N}'}(\boldsymbol{\beta}) \leq d \sum_{j=1}^{r} \sigma_j^{2/d}. \tag{19}$$

By Eq. (17) and Eq. (19), $R_{FC}(\boldsymbol{\beta}) = d\sum_{j=1}^{r} \sigma_j^{2/d}$.

$\square$

## A.2 Diagonal networks with multiple outputs

We give a proof for Theorem 2.

**Theorem 2.** *For any $\boldsymbol{\beta} = (\beta^{(1)}, \beta^{(2)}, \cdots, \beta^{(n)}) \in \mathbb{R}^{m \times n}$,*

$$R_{DNN}(\boldsymbol{\beta}) = d \sum_{i=1}^{n} \left\| \beta^{(i)} \right\|_2^{2/d} \cong \|\boldsymbol{\beta}\|_{2/d, 2},$$

*where $\|\boldsymbol{\beta}\|_{2/d, 2} := (\sum_{i=1}^{n} \|\beta^{(i)}\|_2^{2/d})^{d/2}$ is the matrix $l_{2, 2/d}$ quasi-norm. In particular, with a single output*

$$R_{DNN}(\beta) = d\|\beta\|_{2/d}^{2/d} \cong \|\beta\|_{2/d}.$$

*Proof.* Let $w = (w_1, \ldots, w_{d-1}, W_d)$ be the parameters of a diagonal neural network such that

$$F_{DNN}(w) = W_d \prod_{i=2}^{d} \text{diag } w_{d-i+1} = \boldsymbol{\beta}.$$

Let $V = \prod_{i=2}^{d} \text{diag } w_{d-i+1}$. For each $k \in [n]$, let $v_k = V[k, k]$. Then we want to minimize $\sum_{i \in [d-1]} w_i[k]^2$ subject to $\prod_{i \in [d-1]} w_i[k] = v_k$, for each $k \in [n]$. By AM-GM inequality, the minimum is $(d-1)|v_k|^{2/(d-1)}$.

Thus, it suffices to minimize

$$\sum_{k=1}^{n} (d-1)|v_k|^{2/(d-1)} + \|W_d\|_F^2$$

subject to $W_d V = \boldsymbol{\beta}$. Let $a_{ij} = W_d[i,j]$ and $a_j$ be the $j^{th}$ column of $W_d$. Let $\beta_{ij} = \boldsymbol{\beta}[i,j]$ and $\beta_j$ be the $j^{th}$ column of $\boldsymbol{\beta}$. Then the problem becomes minimize $\sum_{k \in [n]} ((d-1)|v_k|^{2/(d-1)} + \|a_k\|_2^2)$ subject to $v_k a_k = \beta_k$ for all $k \in [n]$. Without loss of generality, we can assume that $v_k > 0$ for all $k$. This breaks up into $n$ separate minimization problems and it suffices to solve one of them. Fix a $k \in [n]$. It suffice to minimize $(d-1)v_k^{2/(d-1)} + \sum_{i \in [m]} a_{ik}^2$ subject to $v_k a_{ik} = \beta_{ik}$ for all $i \in [m]$. Let $y = \sum_{i \in [m]} \beta_{ik}^2$. Let $f(x) = (d-1)x^{2/(d-1)} + y/x^2$. Then $f(x) \geq dy^{1/d}$ by AM-GM inequality, where we write the first term $(d-1)x^{2/(d-1)}$ as $(d-1)$ terms and then apply AM-GM to the whole $d$ terms. This bound can be achieved by $x^* = y^{(d-1)/2d}$. Then the minimum for the whole problem is $d \sum_{k \in [n]} \|\beta_k\|_2^{2/d}$. $\qquad\square$

### A.3 Convolutional networks with multiple outputs

Recall that for each $w_i \in \mathbb{R}^n$, the circulant matrix $W_i$ with respect to $w_i$ is defined as

$$W_i = \frac{1}{\sqrt{n}} \begin{pmatrix} w_i[1] & w_i[2] & \cdots & w_i[n] \\ w_i[n] & w_i[1] & \cdots & w_i[n-1] \\ \vdots & \vdots & & \vdots \\ w_i[2] & w_i[3] & \cdots & w_i[1] \end{pmatrix}. \tag{20}$$

The main idea of the approach is to reduce the convolutional network case to the diagonal network case. To do this, we observe that all the circulant weight matrices $W_i$ are simultaneously diagonalizable by the discrete Fourier transform matrix. Then after a change of basis, the problem is similar to the diagonal network case. Let $\mathsf{F} \in \mathbb{C}^{n \times n}$ be the discrete Fourier transform matrix defined by $\mathsf{F}[j,k] = \frac{1}{\sqrt{n}} \omega_n^{(j-1)(k-1)}$, where $\omega_n = e^{2\pi i/n}$. Note that $\mathsf{F}^* = \mathsf{F}^{-1}$. Then, the Fourier transform of the column vector $c$ is $\mathsf{F}c$ and the Fourier transform of the row vector $\beta^T$ is $\beta^T \mathsf{F}$.

Let

$$d_i = \mathsf{F} w_i, \tag{21}$$

for each $i \in [d-1]$. Then for each $i \in [d-1]$,

$$W_i = \mathsf{F} D_i \mathsf{F}^*, \tag{22}$$

where

$$D_i = \operatorname{diag}(d_i) = \mathsf{F}^* W_i \mathsf{F}. \tag{23}$$

Then

$$\boldsymbol{\beta} = W_d \mathsf{F} D_{d-1} D_{d-2} \cdots D_1 \mathsf{F}^*. \tag{24}$$

Let

$$\hat{\boldsymbol{\beta}} = \boldsymbol{\beta} \mathsf{F} \quad \text{and} \quad \hat{W}_d = W_d \mathsf{F}. \tag{25}$$

Then

$$\hat{\boldsymbol{\beta}} = \hat{W}_d \prod_{i=1}^{d-1} D_{d-i}. \tag{26}$$

Since $\mathsf{F}$ is unitary,

$$\|D_i\|_F = \|W_i\|_F \quad \text{and} \quad \left\| \hat{W}_d \right\|_F = \|W_d\|_F. \tag{27}$$

Thus,

$$\|w\|_2^2 = \sum_{i=1}^{d-1} \|w_i\|^2 + \|W_d\|_F^2 = \sum_{i=1}^{d-1} \|W_i\|_F^2 + \left\| \hat{W}_d \right\|_F^2 = \sum_{i=1}^{d-1} \|D_i\|_F^2 + \left\| \hat{W}_d \right\|_F^2. \tag{28}$$

Note that the above transformation can also be viewed as shifting from "time domain" to "frequency domain". This makes convolution becomes multiplication, by the convolution theorem. This idea will be used both in the full filter width case $q = n$ and in the restricted filter width case $q < n$. As we shall see, the filter width somehow represents the "degree of freedom". In the full filter width case, the situation is very similar to the diagonal case since we can choose the coefficients $w_i$ freely. In the restricted filter width case, however, we lose much freedom due to the sparsity control on $w_i$, which makes our approach harder to work as the lower bound by AM-GM inequality cannot be attained anymore.

**Lemma 13.** *For $x = (x_0, \ldots, x_{N-1}) \in \mathbb{R}^N$, the DFT $\hat{x} = (\hat{x}_0, \ldots, \hat{x}_{N-1})$ satisfies $\hat{x}_k = \hat{x}^*_{-k \bmod N}$. Inversely, if for $x \in \mathbb{C}^N$ we have $x_k = x^*_{-k \bmod N}$, then the inverse Fourier transform of $x$ is real.*

*Proof.* First, let $x = (x_0, \ldots, x_{N-1}) \in \mathbb{R}^n$. Let $\hat{x} = \mathsf{F}x = (\hat{x}_0, \ldots, \hat{x}_{N-1})$. Then

$$\begin{aligned}
\hat{x}_k &= \frac{1}{\sqrt{N}} \sum_{j=0}^{N-1} \omega_N^{jk} x_j \\
&= \left( \frac{1}{\sqrt{N}} \sum_{j=0}^{N-1} \omega_N^{-jk} x_j \right)^* \\
&= \hat{x}^*_{-k \bmod N}.
\end{aligned} \tag{29}$$

Now suppose that $x \in \mathbb{C}^N$ such that

$$x_k = x^*_{-k \bmod N} \tag{30}$$

Let $x' = \mathsf{F}^* x = (x'_0, \ldots, x'_{N-1})$ be the inverse Fourier transform of $x$. Without loss of generality, assume that $N$ is odd. The case that $N$ is even can be done in exactly the same way. Note that by (30), we have

$$x_0 \in \mathbb{R}. \tag{31}$$

Then

$$\begin{aligned}
x'_k &= \frac{1}{\sqrt{N}} \sum_{j=0}^{N-1} \omega_N^{-jk} x_j \\
&= \sqrt{N} x_0 + \frac{1}{\sqrt{N}} \sum_{j=1}^{\frac{N-1}{2}} (\omega_N^{-jk} x_j + \omega_N^{jk} x_{N-j}) \\
&\overset{(30)}{=} \sqrt{N} x_0 + \frac{1}{\sqrt{N}} \sum_{j=1}^{\frac{N-1}{2}} (\omega_N^{-jk} x_j + (\omega_N^{-jk} x_j)^*) \\
&= \sqrt{N} x_0 + \frac{2}{\sqrt{N}} \sum_{j=1}^{\frac{N-1}{2}} \mathrm{Re}(\omega_N^{-jk} x_j) \\
&\overset{(31)}{\in} \mathbb{R}.
\end{aligned} \tag{32}$$

$\square$

Now, we give a proof of Theorem 3

**Theorem 3.** *For any $\boldsymbol{\beta} \in \mathbb{R}^{m \times n}$, let $\hat{\boldsymbol{\beta}} := \boldsymbol{\beta}\mathsf{F}$ and $\hat{\beta}^{(i)}$ be the $i$-th column of $\hat{\boldsymbol{\beta}}$. Then,*

$$R_{CNN(n)}(\boldsymbol{\beta}) = d \sum_{i=1}^{n} \left\| \hat{\beta}^{(i)} \right\|_2^{2/d} \cong \left\| \hat{\boldsymbol{\beta}} \right\|_{2/d,2},$$

*where $\left\| \hat{\boldsymbol{\beta}} \right\|_{2/d,2} := (\sum_{i=1}^{n} \left\| \hat{\beta}^{(i)} \right\|_2^{2/d})^{d/2}$ is the matrix $l_{2,2/d}$ quasi-norm. In particular, with a single output*

$$R_{CNN(n)}(\beta) = d \left\| \hat{\beta} \right\|_{2/d}^{2/d} \cong \left\| \hat{\beta} \right\|_{2/d}.$$

*Proof.* Let $d_p[j] = D_p[j,j] = \mathsf{F}w_p$. Let $\hat{W}_d = (\hat{W}_d^{(1)}, \ldots, \hat{W}_d^{(n)})$. Since $\hat{\boldsymbol{\beta}} = \hat{W}_d D$,

$$\hat{\beta}_j = d[j] \hat{W}_d^{(j)}, \tag{33}$$

for all $j \in [n]$. Thus,

$$\left\|\hat{W_d}^{(j)}\right\|_2^2 = \left\|\hat{\beta}_j\right\|_2^2 / |d[j]|^2. \tag{34}$$

Then

$$
\begin{aligned}
\|w_n\|_2^2 &= \sum_{i=1}^{d-1} \|D_i\|_F^2 + \left\|\hat{W}_d\right\|_F^2 \\
&= \sum_{i=1}^{d-1}\sum_{j=1}^{n} |d_i[j]|^2 + \sum_{j=1}^{n} \left\|\hat{W_d}^{(j)}\right\|_2^2 \\
&= \sum_{i=1}^{d-1}\sum_{j=1}^{n} |d_i[j]|^2 + \sum_{j=1}^{n} \left\|\hat{\beta}_j\right\|_2^2 / |d[j]|^2 \\
&= \sum_{j=1}^{n}\left(\sum_{i=1}^{d-1} |d_i[j]|^2 + \left\|\hat{\beta}_j\right\|_2^2 / \prod_{i=1}^{d-1} |d_i[j]|^2\right) \\
&\geq d\sum_{j=1}^{n} \left\|\hat{\beta}_j\right\|_2^{\frac{2}{d}} \\
&= \sum_{i=1}^{n} d\left\|\hat{\beta}_i\right\|_2^{\frac{2}{d}},
\end{aligned}
\tag{35}
$$

where the second to last step follows by AM-GM inequality. Now we show that the AM-GM inequality can be attained by some $d_i$, whose inverse Fourier transform is real. For the AM-GM inequality to be attained, it suffices to let

$$d_1[j] = \cdots = d_{d-1}[j] = \left\|\hat{\beta}_j\right\|_2^{1/d}, \tag{36}$$

for all $j \in [n]$. Now let $w_i = \mathsf{F}^* d_i$ for each $i \in [d-1]$. Note that the rows of $\hat{\boldsymbol{\beta}}$ are Fourier transforms of the rows of $\boldsymbol{\beta}$, which is real. Then by lemma 13, we have

$$\hat{\beta}_j[k] = \hat{\beta}_{n-j+2}[k]^*, \tag{37}$$

and thus

$$|\hat{\beta}_j[k]|^2 = |\hat{\beta}_{n-j+2}[k]|^2, \tag{38}$$

for all $j \in [n-1], k \in [m]$. Thus,

$$\left\|\hat{\beta}_j\right\|_2^2 = \left\|\hat{\beta}_{n-j+2}\right\|_2^2, \tag{39}$$

for all $j \in [n-1]$. By Eq. (39) and lemma 13, $w_i$ is real for all $i \in [d-1]$. Thus, the bound can be attained. $\qquad\square$

## B   Supplementary materials : CNN with restricted filter width

In this section, we consider CNN with restricted filter width $q$ such that $q|n$. Unlike results in previous sections, results in this section do not give a complete characterization of the induced complexity measure of CNN with restricted filter width. Instead, we will give some results which lead to an example that sheds light on the induced complexity measure of CNN with restricted filter width.

The lemmas in this section (or some highly similar variants) were also discovered in [18]. In addition, they also studied the multiple channel CNN. We put some results on multiple channel CNN in the supplementary materials B.4. Their results on multiple channel CNN is stronger than ours. We show that the representation cost of multiple channel CNN does not depend on the number of channels when the filter width $q = 1$ or $n$. They show that the same holds for any $q \in [n]$.

For simplicity, we will consider single output $m = 1$ and depth two $d = 2$ case of CNN with restricted filter width. The extension to multiple output case is straightforward and similar to what

we did in Section 3.1. The extension to general depth case is also similar to what we did before. In addition, we will assume that

$$q|n. \tag{40}$$

Since $m = 1$, we will use $\beta$ in place of $\boldsymbol{\beta}$ in this section. Also we will use $R_{CNN(q)}(\beta)$ to denote the representation cost of a vector $\beta \in \mathbb{R}^n$ in a convolutional neural network of filter width $q$. Let $\hat{\beta} = \mathsf{F}b$ be the discrete Fourier transform of $\beta$.

## B.1  Filter width one case $q = 1$

We begin with the case $q = 1$, where the convolutional layer is simply a scaling (a multiple of the identity matrix).

**Lemma 14.** *For $\beta \in \mathbb{R}^n$,*

$$R_{CNN(1)}(\beta) = 2\sqrt{n}\|\beta\|_2 = 2\sqrt{n}\left\|\hat{\beta}\right\|_2 \cong \left\|\hat{\beta}\right\|_2 = \|\beta\|_2. \tag{41}$$

The above result shows that the induced complexity measure of CNN with filter width one is $l_2$ norm (or equivalently, $l_2$ norm on the Fourier domain). This is very different from the induced complexity measure of CNN with full filter width (i.e. $q = n$), which is $l_1$ norm on the Fourier domain. Thus, the induced complexity measure of CNN with restricted filter width is in general an interpolation between $l_2$ and $l_1$ norms on the Fourier domain.

*Proof.* When $q = 1$, we identify $w_1$ with $w_1[1]$. Let $\lambda = w_1[1]$. Let $w_2 \in \mathbb{R}^n$ be the weights in the second layer. Then we have $\lambda w_2/\sqrt{n} = \beta$. Thus,

$$R_{CNN(1)}(\beta) = \min\{\lambda^2 + \|w_2\|_2^2 : \lambda w_2 = \sqrt{n}\beta\} = \min(\lambda^2 + n\|\beta\|_2^2/\lambda^2) \overset{(a)}{=} 2\sqrt{n}\|\beta\|_2, \tag{42}$$

where we used AM-GM inequality in $(a)$. Since $\mathsf{F}$ is unitary, $\|\beta\|_2 = \left\|\hat{\beta}\right\|_2$. $\qquad\square$

Note that we clearly have $R_{CNN(q)}(\beta) \le R_{CNN(1)}(\beta) = 2\sqrt{n}\|\beta\|_2 = 2\sqrt{n}\left\|\hat{\beta}\right\|_2$ for all $q$ since we can always set some weights to be zero.

Next, we turn to the general filter width case (i.e. $q \in [n]$).

## B.2  General case $1 \le q \le n$

First, we give a lemma.

**Lemma 15.** *For $\beta \in \mathbb{R}^n$,*

$$R_{CNN(q)}(\beta) = \min_{w \in R^q \times \{0\}^{n-q}} \sum_{i=1}^{q} w[i]^2 + \sum_{j=1}^{n} \frac{n|\hat{\beta}_j|^2}{\sum_{i=1}^{q} w[i]^2 + \sum_{k=1}^{q-1} 2\sum_{i \le q-k} w[i]w[i+k] \cos\frac{2\pi(j-1)k}{n}}, \tag{43}$$

*where $\hat{\beta}_j$ denotes the jth entry of the Fourier transform of $\beta$.*

*Proof.* Let $w = (w_1, w_2)$ be the weights such that

$$F_{CNN(q)}(w) = w_2 W_1 = \beta, \tag{44}$$

where $W_1$ is the circulant matrix with respect to $w_1$. Then by Eq. (26),

$$\hat{\beta} = \hat{w}_2 D, \tag{45}$$

where $D = \mathrm{diag}(\hat{w}_1) = \mathrm{diag}(\mathsf{F}w_1)$ and $\mathsf{F}$ is the Discrete Fourier Transform matrix. Then

$$\hat{w}_2[j] = \frac{\hat{\beta}_j}{\hat{w}_1[j]}. \tag{46}$$

Note that

$$|\hat{w}_1[j]|^2 = (\mathsf{F}w_1[j])(\mathsf{F}w_1[j])^*$$

$$= \frac{1}{n}(\sum_{i=1}^{q} \omega_n^{(j-1)(i-1)} w_1[i])(\sum_{i=1}^{q} \omega_n^{(j-1)(i-1)} w_1[i])^*$$

$$= \frac{1}{n}(\sum_{i=1}^{q} \omega_n^{(j-1)(i-1)} w_1[i])(\sum_{i=1}^{q} \omega_n^{-(j-1)(i-1)} w_1[i])$$

$$= \frac{1}{n}(\sum_{i=1}^{q} w_1[i]^2 + \sum_{s>t}(\omega_n^{(j-1)(s-1)}\omega_n^{-(j-1)(t-1)} + \omega_n^{(j-1)(t-1)}\omega_n^{-(j-1)(s-1)})w_1[s]w_1[t])$$

$$= \frac{1}{n}(\sum_{i=1}^{q} w_1[i]^2 + \sum_{s>t}(\omega_n^{(j-1)(s-t)} + \omega_n^{-(j-1)(s-t)})w_1[s]w_1[t])$$

$$= \frac{1}{n}(\sum_{i=1}^{q} w_1[i]^2 + 2\sum_{s>t} \cos\frac{2\pi(s-t)(j-1)}{n} w_1[s]w_1[t])$$

$$= \frac{1}{n}(\sum_{i=1}^{q} w_1[i]^2 + \sum_{k=1}^{q-1} 2\sum_{i\leq q-k} w_1[i]w_1[i+k] \cos\frac{2\pi(j-1)k}{n}).$$

$$(47)$$

Then,

$$\|w\|_2^2 = \sum_{i=1}^{q} w_1[i]^2 + \sum_{j=1}^{n} |\hat{w}_2[j]|^2 = \sum_{i=1}^{q} w_1[i]^2 + \sum_{j=1}^{n} \frac{n|\hat{\beta}_j|^2}{\sum_{i=1}^{q} w_1[i]^2 + \sum_{k=1}^{q-1} 2\sum_{i\leq q-k} w_1[i]w_1[i+k] \cos\frac{2\pi(j-1)k}{n}}.$$

$$(48)$$

Now, minimizing over $w_1$ gives the desired result. $\qquad\square$

### B.2.1 Key lower bound of the representation cost

Now we give a lower bound for the general case.

**Lemma 16.** *For single output, depth-two CNN with filter width q such that q|n,*

$$R_{CNN(q)}(\beta) \geq 2\sqrt{\frac{n}{q}}\sqrt{\sum_{t=1}^{n/q}(\sum_{j\in\mathcal{S}_t} |\hat{\beta}_j|)^2}, \tag{49}$$

*where $\mathcal{S}_t = \{t + \frac{n}{q}v : v \in \{0, 1, \ldots, q-1\}\}$.*

To prove this lower bound, we first do a change of variable (note that $x$ and $y$ are not input and output of the network):

$$y_0 = \sum_{i=1}^{q} w[i]^2$$

$$x_k = \frac{2\sum_{i\leq q-k} w[i]w[i+k]}{\sum_{i=1}^{q} w[i]^2}, \qquad k \in [q-1]. \tag{50}$$

Let

$$y = (y_0, x_1, \ldots, x_{q-1}), \qquad x = (x_1, \ldots, x_{q-1}). \tag{51}$$

Note that the coordinates of $y$ are not independent. However, this is enough to get a lower bound. Let

$$Y = \{(y_0, x_1, \ldots, x_{q-1}) : c \in R^q \times \{0\}^{n-q}\},$$
$$X = \{(x_1, \ldots, x_{q-1}) : c \in R^q \times \{0\}^{n-q}\} \tag{52}$$

be the feasible region for $y, x$. Then by Lemma 15, we have

$$
\begin{aligned}
R_{CNN(q)}(\beta) &= \min_{c \in R^q \times \{0\}^{n-q}} \sum_{i=1}^{q} w[i]^2 + \sum_{j=1}^{n} \frac{n|\hat{\beta}_j|^2}{\sum_{i=1}^{q} w[i]^2 + \sum_{k=1}^{q-1} 2\sum_{i \leq q-k} w[i]w[i+k] \cos \frac{2\pi(j-1)k}{n}} \\
&= \min_{y \in Y} y_0 + \sum_{j=1}^{n} \frac{n|\hat{\beta}_j|^2}{y_0 + y_0 \sum_{k=1}^{q-1} x_k \cos \frac{2\pi(j-1)k}{n}} \\
&\stackrel{(a)}{=} 2\sqrt{n} \min_{x \in X} \sqrt{\sum_{j=1}^{n} \frac{|\hat{\beta}_j|^2}{1 + \sum_{k=1}^{q-1} x_k \cos \frac{2\pi(j-1)k}{n}}} \\
&= 2\sqrt{n} \sqrt{\min_{x \in X} f(x)} \\
&\geq 2\sqrt{n} \sqrt{\min_{x \in \mathbb{R}^{q-1}} f(x)},
\end{aligned}
$$

(53)

where $(a)$ follows by AM-GM inequality and the fact that scale $c$ by $t$ does not change $x$ but scales $y_0$ by $t^2$, and we define

$$
f(x) = \sum_{j=1}^{n} \frac{|\hat{\beta}_j|^2}{1 + \sum_{k=1}^{q-1} x_k \cos \frac{2\pi(j-1)k}{n}}.
$$

(54)

We prove a simple lemma.

**Lemma 17.** *For any $u, w \in [0, 2\pi)$ such that $u \neq 0$ and $qu = 2\pi k$ for some $k \in \mathbb{N}$,*

$$
\sum_{v=0}^{q-1} \cos(w + vu) = 0.
$$

(55)

*Proof.*

$$
\begin{aligned}
\sum_{v=0}^{q-1} \cos(w + vu) &= \text{Re}\left(\sum_{v=0}^{q-1} e^{i(w+vu)}\right) \\
&= \text{Re}\left(\sum_{v=0}^{q-1} e^{iw}(e^{iu})^v\right) \\
&= \text{Re}\left(\frac{e^{iw} - e^{iw}e^{iqu}}{1 - e^{iu}}\right) \\
&= \text{Re}\left(\frac{e^{iw} - e^{iw}e^{i2\pi k}}{1 - e^{iu}}\right) \\
&= \text{Re}\left(\frac{e^{iw} - e^{iw}}{1 - e^{iu}}\right) \\
&= 0. \qquad \square
\end{aligned}
$$

Now, we give a proof of Lemma 16.

**Lemma 16.** *For single output, depth-two CNN with filter width $q$ such that $q|n$,*

$$
R_{CNN(q)}(\beta) \geq 2\sqrt{\frac{n}{q}} \sqrt{\sum_{t=1}^{n/q} (\sum_{j \in \mathcal{S}_t} |\hat{\beta}_j|)^2},
$$

(49)

*where $\mathcal{S}_t = \{t + \frac{n}{q}v : v \in \{0, 1, \ldots, q-1\}\}$.*

*Proof.* By (53) and (54), it suffices to show that

$$
qf(x) \geq \sum_{t=1}^{n/q} (\sum_{j \in \mathcal{S}_t} |\hat{\beta}_j|)^2.
$$

(56)

By lemma 17, we have

$$\sum_{j \in \mathcal{S}_t} \cos \frac{2\pi(j-1)k}{n} = \sum_{v=0}^{q-1} \cos\left(\frac{2\pi(t-1)k}{n} + \frac{2\pi k v}{q}\right) = 0, \tag{57}$$

for all $k$. Thus,

$$\sum_{j \in \mathcal{S}_t}(1 + \sum_{k=1}^{q-1} x_k \cos \frac{2\pi(j-1)k}{n}) = |\mathcal{S}_t| + \sum_{k=1}^{q-1} x_k \sum_{j \in \mathcal{S}_t} \cos \frac{2\pi(j-1)k}{n}$$

$$= q + \sum_{k=1}^{q-1} x_k 0 \tag{58}$$

$$= q.$$

Now we have

$$qf(x) = q \sum_{t=1}^{n/q} (\sum_{j \in \mathcal{S}_t} \frac{|\hat{\beta}_j|^2}{1 + \sum_{k=1}^{q-1} x_k \cos \frac{2\pi(j-1)k}{n}})$$

$$= \sum_{t=1}^{n/q} (\sum_{j \in \mathcal{S}_t} (1 + \sum_{k=1}^{q-1} x_k \cos \frac{2\pi(j-1)k}{n}))(\sum_{j \in \mathcal{S}_t} \frac{|\hat{\beta}_j|^2}{1 + \sum_{k=1}^{q-1} x_k \cos \frac{2\pi(j-1)k}{n}}) \tag{59}$$

$$\overset{(a)}{\geq} \sum_{t=1}^{n/q} (\sum_{j \in \mathcal{S}_t} |\hat{\beta}_j|)^2,$$

where $(a)$ follows from Cauchy's inequality. $\qquad\square$

Next we show that the bound in Lemma 16 is tight up to a multiplicative factor of $\sqrt{q}$.

**Lemma 18.** *For $\beta \in \mathbb{R}^n$ and $q|n$,*

$$2\sqrt{\frac{n}{q}}\sqrt{\sum_{t=1}^{n/q}(\sum_{j \in \mathcal{S}_t} |\hat{\beta}_j|)^2} \leq R_{CNN(q)}(\beta) \leq 2\sqrt{n}\sqrt{\sum_{t=1}^{n/q}(\sum_{j \in \mathcal{S}_t} |\hat{\beta}_j|)^2}. \tag{60}$$

*In addition,*

$$\frac{1}{\sqrt{q}} R_{CNN(1)}(\beta) \leq R_{CNN(q)}(\beta) \leq R_{CNN(1)}(\beta). \tag{61}$$

*Proof.*

$$R_{CNN(q)}(\beta) \geq 2\sqrt{\frac{n}{q}}\sqrt{\sum_{t=1}^{n/q}(\sum_{j \in \mathcal{S}_t} |\hat{\beta}_j|)^2}$$

$$\geq 2\sqrt{\frac{n}{q}}\sqrt{\sum_{t=1}^{n/q}(\sum_{j \in \mathcal{S}_t} |\hat{\beta}_j|^2)}$$

$$= 2\sqrt{\frac{n}{q}}\sqrt{\sum_{j=1}^{n} |\hat{\beta}_j|^2} \tag{62}$$

$$= R_{CNN(1)}(\beta)/\sqrt{q}$$

$$\geq R_{CNN(q)}(\beta)/\sqrt{q}.$$

$\qquad\square$

Now, we show that both the lower and the upper bound in Theorem 16 can be attained.

**Example** Let

$$\beta = (1, 1, \ldots, 1)/\sqrt{n}, \qquad \beta' = (1, 0, \ldots, 0). \tag{63}$$

Then,

$$R_{CNN(q)}(\beta) = R_{CNN(1)}(\beta)/\sqrt{q}, \qquad R_{CNN(q)}(\beta') = R_{CNN(1)}(\beta'). \tag{64}$$

*Proof.* We begin with the lower bound. Let

$$\beta = (1, 1, \ldots, 1)/\sqrt{n}. \tag{65}$$

Then

$$\hat{\beta} = (1, 0, \ldots, 0). \tag{66}$$

Then by Lemma 14, we have

$$R_{CNN(1)}(\beta) = 2\sqrt{n}\|\beta\|_2 = 2\sqrt{n}. \tag{67}$$

Then we have,

$$
\begin{aligned}
&\sum_{i=1}^{q} w[i]^2 + \sum_{j=1}^{n} \frac{n|\hat{\beta}_j|^2}{\sum_{i=1}^{q} w[i]^2 + \sum_{k=1}^{q-1} 2\sum_{i \leq q-k} w[i]w[i+k]\cos\frac{2\pi(j-1)k}{n}} \\
&= \sum_{i=1}^{q} w[i]^2 + \frac{n}{(\sum_{i=1}^{q} w[i])^2} \\
&= \frac{1}{q}\left(\sum_{i=1}^{q} 1^2\right)\left(\sum_{i=1}^{q} w[i]^2\right) + \frac{n}{(\sum_{i=1}^{q} w[i])^2} \\
&\overset{(a)}{\geq} \frac{1}{q}\left(\sum_{i=1}^{q} w[i]\right)^2 + \frac{n}{(\sum_{i=1}^{q} w[i])^2} \\
&\overset{(b)}{\geq} 2\sqrt{\frac{n}{q}},
\end{aligned}
\tag{68}
$$

where $(a)$ follows from Cauchy's inequality and $(b)$ follows from AM-GM inequality. The bound is attained when

$$w[1] = w[2] = \cdots = w[q] = \frac{n^{1/4}}{q^{3/4}}. \tag{69}$$

Then by Lemma 15,

$$R_{CNN(q)}(\beta) = 2\sqrt{\frac{n}{q}} = R_{CNN(1)}(\beta)/\sqrt{q}. \tag{70}$$

Note that we did not use the fact that $q|n$ in the above calculation. Thus, (70) holds for all $q$ and $n$. For the upper bound, we take

$$\beta' = (1, 0, \ldots, 0). \tag{71}$$

Then $\hat{\beta}' = (1, 1, \ldots, 1)/\sqrt{n}$. Thus,

$$R_{CNN(n)}(\beta') = 2\left\|\hat{\beta}'\right\|_1 = 2\sqrt{n} = 2\sqrt{n}\|\beta'\|_2 = R_{CNN(1)}(\beta'), \tag{72}$$

where the last step follows from Lemma 14. Thus,

$$R_{CNN(q)}(\beta') = R_{CNN(1)}(\beta') = 2\sqrt{n}, \tag{73}$$

since $R_{CNN(n)}(\beta') \leq R_{CNN(q)}(\beta') \leq R_{CNN(1)}(\beta')$. $\qquad\square$

### B.2.2 Periodic and antiperiodic linear predictors

Now, we compute the representation costs of some special vectors. We say that $\beta \in \mathbb{R}^n$ is $q$-periodic if $\beta_{i+q} = \beta_i$ for all $i$, and $q$-antiperiodic if $\beta_{i+q} = -\beta_i$ for all $i$.

**Lemma 19.** *For $q|n$, if either $\beta$ is $q$-periodic or $n$ is even, $q$ is odd, and $\beta$ is is $q$-antiperiodic, then*

$$R_{CNN(q)}(\beta) = 2\sqrt{\frac{n}{q}}\left\|\hat{\beta}\right\|_1 \cong \left\|\hat{\beta}\right\|_1.$$

We give a proof of Theorem 19 by considering the periodic and antiperiodic cases separately.

Recall that

$$\mathcal{S}_t = \{t + \frac{n}{q}v : v \in \{0, 1, \ldots, q-1\}\}, \tag{74}$$

for each $t \in [n/q]$. We say that a vector $\beta$ is supported on $\mathcal{S}_t$ if $\beta[i] \neq 0$ only if $i \in \mathcal{S}_t$. Let $\mathsf{F}_n$ be the $n \times n$ discrete Fourier transform matrix. Let $\mathsf{F} = \mathsf{F}_n$.

**Periodic case**

We begin with the periodic case.

**Theorem 20.** *For any $\beta \in \mathbb{R}^n$ that is $q$-periodic,*

$$R_{CNN(q)}(\beta) = 2\sqrt{\frac{n}{q}}\left\|\hat{\beta}\right\|_1. \tag{75}$$

Before giving the proof, we first recall a lemma and give a characterization of $q$-periodic vectors. We will then use this characterization to show that the representation cost of $q$-periodic vectors can attain the lower bound in Theorem 16.

**Lemma 21.** *For $x = (x_0, \ldots, x_{N-1}) \in \mathbb{R}^N$, the DFT $\hat{x} = (\hat{x}_0, \ldots, \hat{x}_{N-1})$ satisfies $\hat{x}_k = \hat{x}^*_{-k \bmod N}$. Inversely, if for $x \in \mathbb{C}^N$ we have $x_k = x^*_{-k \bmod N}$, then the inverse Fourier transform of $x$ is real.*

**Lemma 22.** *Let $\beta \in \mathbb{R}^n$. Then $\hat{\beta}$ is supported on $\mathcal{S}_1$ if and only if $\beta$ is $q$-periodic, where $\mathcal{S}_1$ is defined in Eq. (74).*

*Proof.* For simplicity, let

$$s = \frac{2\pi i}{q}, \qquad w = \frac{2\pi i}{n} \tag{76}$$

Suppose that $\hat{\beta}$ is supported on $\mathcal{S}_1$. Then for any $k \leq n - q$, we have

$$
\begin{aligned}
\beta_{k+q} &= (\mathsf{F}^{-1}\hat{\beta})_{k+q} \\
&= \frac{1}{\sqrt{n}}\sum_{v=0}^{q-1}\hat{\beta}_{1+vn/q}e^{-(k+q-1)vs} \\
&= \frac{1}{\sqrt{n}}\sum_{v=0}^{q-1}\hat{\beta}_{1+vn/q}e^{-(k-1)vs}e^{-qvs} \\
&= \frac{1}{\sqrt{n}}\sum_{v=0}^{q-1}\hat{\beta}_{1+vn/q}e^{-(k-1)vs} \\
&= (\mathsf{F}^{-1}\hat{\beta})_k \\
&= \beta_k.
\end{aligned}
\tag{77}
$$

Thus, $\beta$ is $q$-periodic.

Now suppose that $\beta$ is $q$-periodic. Then for any $j \notin \mathcal{S}_1$, we have

$$
\begin{aligned}
\hat{\beta}_j &= (\mathsf{F}\beta)_j \\
&= \frac{1}{\sqrt{n}} \sum_{t=1}^{q} \sum_{v=0}^{n/q-1} \beta_{t+qv} e^{(j-1)(t-1+qv)w} \\
&= \frac{1}{\sqrt{n}} \sum_{t=1}^{q} \sum_{v=0}^{n/q-1} \beta_t e^{(j-1)(t-1+qv)w} \\
&= \frac{1}{\sqrt{n}} \sum_{t=1}^{q} \beta_t \sum_{v=0}^{n/q-1} e^{(j-1)(t-1+qv)w} \\
&= \frac{1}{\sqrt{n}} \sum_{t=1}^{q} \beta_t \sum_{v=0}^{n/q-1} e^{(j-1)(t-1)w} e^{(j-1)vqw} \\
&= \frac{1}{\sqrt{n}} \sum_{t=1}^{q} \beta_t e^{(j-1)(t-1)w} \sum_{v=0}^{n/q-1} e^{(j-1)vqw} \\
&= \frac{1}{\sqrt{n}} \sum_{t=1}^{q} \beta_t e^{(j-1)(t-1)w} \frac{1 - e^{(j-1)nw}}{1 - e^{(j-1)qw}} \\
&= \frac{1}{\sqrt{n}} \sum_{t=1}^{q} \beta_t e^{(j-1)(t-1)w} \frac{1 - 1}{1 - e^{(j-1)qw}} \\
&= 0.
\end{aligned}
\tag{78}
$$

Thus, $\hat{\beta}$ is supported on $\mathcal{S}_1$. $\qquad\square$

Next, we give some discussion of when the lower bound in Theorem 16 can be attained, which is central to the proof of Theorem (20). By Eq. (59), $R_{CNN(q)}(\beta) = 2\sqrt{\frac{n}{q}} \sqrt{\sum_{t=1}^{n/q} (\sum_{j \in \mathcal{S}_t} |\hat{\beta}_j|)^2}$ if and only if the Cauchy inequality (($a$) in Eq. (59)) is achieved with equality. By Eq. (53), this happens if and only if there exists weights (in the convolutional layer) $w \in \mathbb{R}^q \times \{0\}^{n-q}$ and $\lambda_1, \ldots, \lambda_{n/q} \in \mathbb{R}$ such that for all $i \in [n/q]$, for all $j \in \mathcal{S}_i$,

$$
|\hat{w}_j|^4 = \lambda_i |\hat{\beta}_j|^2,
\tag{79}
$$

where $\mathcal{S}_i$ is defined in Eq. (74). Note that in AM-GM inequality in ($a$) of Eq. (53) can always be attained by scaling the weights $w$ by some constant without changing the ratios between the $|\hat{w}_j|$s. Thus, after satisfying Eq. (79), we can always change $w$ to $\mu w$ by some appropriate $\mu \in \mathbb{R}$ so that ($a$) in Eq. (53) holds and Eq. (79) still holds with a different choice of $\lambda_i$s.

By Lemma 22, if $\beta$ is $q$-periodic, then $\text{supp}(\hat{\beta}) \subseteq \mathcal{S}_1$. Thus, the condition in Eq (79) becomes, there exists $\lambda_1 \in \mathbb{R}$ and weights $w \in \mathbb{R}^q \times \{0\}^{n-q}$ such that for all $j \in \mathcal{S}_1$

$$
|\hat{w}_j|^4 = \lambda_1 |\hat{\beta}_j|^2.
\tag{80}
$$

Since $w \in \mathbb{R}^q \times \{0\}^{n-q}$, we only care about the first $q$ columns of $\mathsf{F}$ in order to compute $\hat{w}$. Let $\mathsf{F}' \in \mathbb{R}^{n \times q}$ be the submatrix of $\mathsf{F}$ consisting of the first $q$ columns of $\mathsf{F}$. By Eq. (80), we only need information of $\hat{w}_j$ for $j \in \mathcal{S}_1$ to determine whether $R_{CNN(q)}(\beta) = 2\sqrt{\frac{n}{q}} \sqrt{\sum_{t=1}^{n/q} (\sum_{j \in \mathcal{S}_t} |\hat{\beta}_j|)^2}$. Let

$$
\tilde{\mathsf{F}} = \mathsf{F}'[\mathcal{S}_1]
\tag{81}
$$

be the $q \times q$ submatrix of $\mathsf{F}'$ consisting of the $q$ rows of $\mathsf{F}'$ whose indices are specified by $\mathcal{S}_1$. Then, we observe that

$$
\tilde{\mathsf{F}} = \sqrt{\frac{q}{n}} \mathsf{F}_q
\tag{82}
$$

is some scaling of the $q \times q$ discrete Fourier transform matrix.

Now we give the proof of the theorem.

*Proof of Theorem* (20). By lemma (22), $\hat{\beta}$ is supported on $\mathcal{S}_1$. Then

$$2\sqrt{\frac{n}{q}}\left\|\hat{\beta}\right\|_1 = 2\sqrt{\frac{n}{q}}\sqrt{\sum_{t=1}^{n/q}(\sum_{j\in\mathcal{S}_t}|\hat{\beta}_j|)^2}. \tag{83}$$

Thus, it suffices to show that the lower bound in Theorem 16 can be attained. Let $w \in \mathbb{R}^n$ be the weights in the convolutional layer. Let

$$u = w[1:q] \tag{84}$$

be the first $q$ entries of $w$. Since $\operatorname{supp}(w) \subseteq [q]$,

$$\hat{w} = \mathsf{F}w = \mathsf{F}'u, \tag{85}$$

where $\mathsf{F}'$ is the submatrix of $\mathsf{F}$ which consists of the first $q$ columns of $\mathsf{F}$. Let

$$\gamma = \hat{\beta}[\mathcal{S}_1] \in \mathbb{R}^q \tag{86}$$

be the subvector of $\hat{\beta}$ whose indices are specified by $\mathcal{S}_1$. Let

$$\hat{w}' = \hat{w}[\mathcal{S}_1] \tag{87}$$

be the subvector of $\hat{w}$ whose indices are specified by $\mathcal{S}_1$. Recall that $\tilde{\mathsf{F}}$ is the submatrix of $\mathsf{F}'$ which consists of the $q$ rows of $\mathsf{F}'$ whose induces are specified by $\mathcal{S}_1$. By Eq. (85), Eq. (87), and Eq. (82),

$$\hat{w}' = \tilde{\mathsf{F}}u = \sqrt{\frac{q}{n}}\mathsf{F}_q u = \sqrt{\frac{q}{n}}\hat{u}, \tag{88}$$

where $\hat{u}$ is the Fourier transform of $u$.

By Eq.(80), the lower bound in Theorem 16 is attained if and only if there exists $\lambda_1 \in \mathbb{R}$ and weights $c \in \mathbb{R}^q \times \{0\}^{n-q}$ such that for all $j \in \mathcal{S}_1$

$$|\hat{w}_j|^4 = \lambda_1|\hat{\beta}_j|^2. \tag{89}$$

By Eq. (86) and Eq. (87), Eq. (89) is equivalent to

$$|\hat{w}'[i]|^4 = \lambda_1|\gamma[i]|^2. \tag{90}$$

for all $i \in [q]$. By Eq. (88), it suffices to show that there exists $u \in \mathbb{R}^q$ and $\lambda \in \mathbb{R}$ such that

$$\hat{u}[i] = \lambda\sqrt{|\gamma[i]|} \tag{91}$$

for all $i \in [q]$. By lemma 13,

$$|\hat{\beta}_j| = |\hat{\beta}_{n-j+2}| \tag{92}$$

for all $j$. Thus,

$$\sqrt{|\gamma[i]|} = \sqrt{|\hat{\beta}_{(i-1)n/q+1}|} = \sqrt{|\hat{\beta}_{(q-i+1)n/q+1}|} = \sqrt{|\gamma[q-i+2]|}. \tag{93}$$

Let $\lambda \in \mathbb{R}_+$ be arbitrary. Let $\gamma'[i] = \lambda\sqrt{|\gamma[i]|}$. Let $u = \mathsf{F}_q^{-1}\gamma'$. Then by lemma 13, we have

$$u \in \mathbb{R}^q. \tag{94}$$

Thus, the lower bound can be attained. $\square$

**Antiperiodic case** Now we consider the antiperiodic case. We assume that $n$ is even and $q$ is odd. For simplicity let

$$w = \frac{2\pi i}{n}. \tag{95}$$

Let $r \in [n/q]$ be such that

$$\frac{n}{2} + 1 \in \mathcal{S}_r. \tag{96}$$

For each $j \in [n]$, let

$$j' = (j - \frac{n}{2}) + n\mathbf{1}_{j \le \frac{n}{2}}. \tag{97}$$

Then we have

$$
\begin{aligned}
\mathsf{F}[j,k] &= \frac{1}{\sqrt{n}} e^{(j-1)(k-1)w} \\
&= \frac{1}{\sqrt{n}} e^{(j'+n/2-1)(k-1)w} \\
&= \frac{1}{\sqrt{n}} e^{n(k-1)w/2} e^{(j'-1)(k-1)w} \\
&= \frac{1}{\sqrt{n}} (-1)^{k-1} e^{(j'-1)(k-1)w} \\
&= (-1)^{k-1} \mathsf{F}[j',k].
\end{aligned}
\tag{98}
$$

Let

$$
P = \begin{bmatrix}
e_{n/2+1} \\
e_{n/2+2} \\
\vdots \\
e_n \\
e_1 \\
e_2 \\
\vdots \\
e_{n/2}
\end{bmatrix}
\tag{99}
$$

is a permutation matrix. Let $\mathsf{F}'' \in \mathbb{R}^{n \times q}$ be the submatrix of $P\mathsf{F}$ consisting of the first $q$ columns of $P\mathsf{F}$. Let

$$
\tilde{\mathsf{F}}' = \mathsf{F}''[\mathcal{S}_r]
\tag{100}
$$

be the $q \times q$ submatrix of $\mathsf{F}''$ consisting of the $q$ rows of $\mathsf{F}'$ whose indices are specified by $\mathcal{S}_r$. Then by (98), we have

$$
\tilde{\mathsf{F}}' = (-1)^{n/2} \sqrt{\frac{q}{n}} \mathsf{F}_q.
\tag{101}
$$

Now we state the result.

**Theorem 23.** *Let $n$ be even and $q$ be odd. For any $\beta \in \mathbb{R}^n$ that is $q$-antiperiodic,*

$$
R_{CNN(q)}(\beta) = 2\sqrt{\frac{n}{q}} \left\| \hat{\beta} \right\|_1.
\tag{102}
$$

Before giving the proof, we first give a characterization of $q$-antiperiodic vectors. We will then use this characterization to show that the representation cost of $q$-antiperiodic vectors can attain the lower bound in Theorem 16.

**Lemma 24.** *Let $n$ be even and $q$ be odd. Then $\beta \in \mathbb{R}^n$ is $q$-antiperiodic if and only if $\hat{\beta}$ is supported on $\mathcal{S}_r$, where $\mathcal{S}_r$ is defined in Eq. (74) and Eq. (96).*

*Proof.* For simplicity, let

$$
s = \frac{2\pi i}{q}, \qquad w = \frac{2\pi i}{n}
\tag{103}
$$

Suppose that $\hat{\beta}$ is supported on $\mathcal{S}_r$. Then for any $k \leq n - q$, we have

$$
\begin{aligned}
\beta_{k+q} &= (\mathsf{F}^{-1}\hat{\beta})_{k+q} \\
&= \frac{1}{\sqrt{n}} \sum_{v=0}^{q-1} \hat{\beta}_{r+vn/q} e^{-(k+q-1)(r-1+vn/q)w} \\
&= \frac{1}{\sqrt{n}} \sum_{v=0}^{q-1} \hat{\beta}_{r+vn/q} e^{-(k-1)(r-1+vn/q)w} e^{-q(r-1)w-vnw} \\
&= \frac{1}{\sqrt{n}} \sum_{v=0}^{q-1} \hat{\beta}_{r+vn/q} e^{-(k-1)(r-1+vn/q)w} e^{-q(r-1)w} \\
&= \frac{1}{\sqrt{n}} \sum_{v=0}^{q-1} \hat{\beta}_{r+vn/q} e^{-(k-1)(r-1+vn/q)w} e^{-q\pi i} \\
&= \frac{1}{\sqrt{n}} \sum_{v=0}^{q-1} \hat{\beta}_{r+vn/q} e^{-(k-1)(r-1+vn/q)w} (-1) \\
&= -(\mathsf{F}^{-1}\hat{\beta})_k \\
&= -\beta_k.
\end{aligned}
\tag{104}
$$

Thus, $\beta$ is $q$-antiperiodic.

Now suppose that $\beta$ is $q$-antiperiodic. Then for any $j \notin \mathcal{S}_r$, we have

$$
\begin{aligned}
\hat{\beta}_j &= (\mathsf{F}\beta)_j \\
&= \frac{1}{\sqrt{n}} \sum_{t=1}^{q} \sum_{v=0}^{n/q-1} \beta_{t+qv} e^{(j-1)(t-1+qv)w} \\
&= \frac{1}{\sqrt{n}} \sum_{t=1}^{q} \sum_{v=0}^{n/q-1} (-1)^v \beta_t e^{(j-1)(t-1+qv)w} \\
&= \frac{1}{\sqrt{n}} \sum_{t=1}^{q} \beta_t \sum_{v=0}^{n/q-1} (-1)^v e^{(j-1)(t-1+qv)w} \\
&= \frac{1}{\sqrt{n}} \sum_{t=1}^{q} \beta_t \sum_{v=0}^{n/q-1} (-1)^v e^{(j-1)(t-1)w} e^{(j-1)vqw} \\
&= \frac{1}{\sqrt{n}} \sum_{t=1}^{q} \beta_t e^{(j-1)(t-1)w} \sum_{v=0}^{n/q-1} (-1)^v e^{(j-1)vqw} \\
&= \frac{1}{\sqrt{n}} \sum_{t=1}^{q} \beta_t e^{(j-1)(t-1)w} \frac{1 - (-1)^{n/q} e^{(j-1)nw}}{1 + e^{(j-1)qw}} \\
&= \frac{1}{\sqrt{n}} \sum_{t=1}^{q} \beta_t e^{(j-1)(t-1)w} \frac{1 - 1}{1 + e^{(j-1)qw}} \\
&= 0.
\end{aligned}
\tag{105}
$$

Thus, $\hat{\beta}$ is supported on $\mathcal{S}_r$. $\qquad\square$

Now we give the proof of the theorem, which is similar to the proof in the periodic case.

*Proof of Theorem* (23). By lemma (22), $\hat{\beta}$ is supported on $\mathcal{S}_r$. Then

$$
2\sqrt{\frac{n}{q}} \big\| \hat{\beta} \big\|_1 = 2\sqrt{\frac{n}{q}} \sqrt{\sum_{t=1}^{n/q} \Big(\sum_{j \in \mathcal{S}_t} |\hat{\beta}_j|\Big)^2}.
\tag{106}
$$

Thus, it suffices to show that the lower bound in Theorem 16 can be attained. Let $w \in \mathbb{R}^n$ be the weights in the convolutional layer. Let

$$u = w[1:q] \tag{107}$$

be the first $q$ entries of $w$. Let

$$\gamma = (P\hat{\beta})[\mathcal{S}_1] \in \mathbb{R}^q \tag{108}$$

be a permutation of the nonzero entries of $\hat{\beta}$. Let

$$\hat{w}' = (P\hat{w})[\mathcal{S}_1] \tag{109}$$

be the subvector of $P\hat{w}$ whose indices are specified by $\mathcal{S}_1$. Now we have

$$\hat{w}' = \tilde{\mathsf{F}}'u = (-1)^{n/2}\sqrt{\frac{q}{n}}\mathsf{F}_q u = (-1)^{n/2}\sqrt{\frac{q}{n}}\hat{u}, \tag{110}$$

where $\hat{u}$ is the Fourier transform of $u$. Note that in order for $(a)$ in (59) to be attained with equality, it suffices to show that for any $\lambda > 0$ there exists $u \in \mathbb{R}^q$ such that

$$\hat{u}[i] = \lambda\sqrt{|\gamma[i]|} \tag{111}$$

for all $i \in [q]$. By lemma 13,

$$|\hat{\beta}_j| = |\hat{\beta}_{n-j+2}| \tag{112}$$

for all $j$. In other words,

$$|\hat{\beta}_j| = |\hat{\beta}_k| \tag{113}$$

if $j + k = n + 2$. Let $f : [n] \longrightarrow [n]$ be defined by

$$f(i) = i - \frac{n}{2} + n\mathbf{1}_{i \le \frac{n}{2}}. \tag{114}$$

Then note that if $j + k = n + 2$ and $j \neq k$, then

$$f(j) + f(k) = j - \frac{n}{2} + k - \frac{n}{2} + n = j + k = n + 2. \tag{115}$$

Thus,

$$P\hat{\beta}_j = \hat{\beta}_{f(j)} = \hat{\beta}_{f(n-j+2)} = P\hat{\beta}_{n-j+2}. \tag{116}$$

Thus,

$$\sqrt{|\gamma[i]|} = \sqrt{|P\hat{\beta}_{(i-1)n/q+1}|} = \sqrt{|P\hat{\beta}_{(q-i+1)n/q+1}|} = \sqrt{|\gamma[q-i+2]|}. \tag{117}$$

Let $\gamma'[i] = \lambda\sqrt{|\gamma[i]|}$. Let $u = \mathsf{F}_q^{-1}\gamma'$. Then by lemma 13, we have

$$u \in \mathbb{R}^q. \tag{118}$$

Thus, the lower bound can be attained. $\qquad\square$

### B.2.3   Induced complexity measure of CNN with restricted filter width

In this section, we give some examples to show how the induced complexity measure of CNN with restricted filter width changes with the filter width.

Since CNN with larger filter width contains CNN with smaller filter width as a subnetwork, the representation cost $R_{CNN(q)}(\beta)$ is monotonically decreasing in $q$ for all $\beta \in \mathbb{R}^n$. Given a predictor $\beta \in \mathbb{R}^n$ such that $R_{CNN(1)}(\beta) > R_{CNN(n)}(\beta)$, one might expect that $R_{CNN(q)}(\beta)$ is strictly decreasing in $q$. However, this is not always the case as the following example shows.

**Example** Consider $q_2|n$. Let $e_j \in \mathbb{R}^n$ be the $j$th standard basis vector and

$$\beta^{(2)} = \sum_{j=0}^{n/q_2-1} e_{1+jq_2}.$$

Then, for all $1 \le q_1 \le q_2$,

$$R_{CNN(1)}(\beta^{(2)}) = R_{CNN(q_1)}(\beta^{(2)}) = \frac{2n}{\sqrt{q_2}}, \quad \text{and} \quad R_{CNN(n)}(\beta^{(2)}) = 2\sqrt{n}. \tag{119}$$

Now, we give a proof of Eq. (119).

*Proof of Eq. (119).* Let

$$\mathcal{S}_t(q_2) = \{t + v\frac{n}{q_2} : v \in \{0, 1, \ldots, q_2 - 1\}\}, \tag{120}$$

for each $t \in [n/q_2]$. For simplicity, let

$$s = \frac{2\pi i q_2}{n}, \qquad w = \frac{2\pi i}{n}. \tag{121}$$

For each $j \notin \mathcal{S}_1[q_2]$,

$$
\begin{aligned}
\hat{\beta}^{(2)}[j] &= \frac{1}{\sqrt{n}} \sum_{t=0}^{n/q_2-1} e^{tq_2(j-1)w} \\
&= \frac{1}{\sqrt{n}} \sum_{t=0}^{n/q_2-1} e^{ts(j-1)} \\
&= \frac{1}{\sqrt{n}} \frac{1 - e^{s(j-1)n/q_2}}{1 - e^{s(j-1)}} \\
&= 0.
\end{aligned}
\tag{122}
$$

For $j \in \mathcal{S}_1[q_2]$,

$$
\begin{aligned}
\hat{\beta}^{(2)}[j] &= \frac{1}{\sqrt{n}} \sum_{t=0}^{n/q_2-1} e^{tq_2(j-1)w} \\
&= \frac{1}{\sqrt{n}} \sum_{t=0}^{n/q_2-1} e^{ts(j-1)} \\
&= \frac{1}{\sqrt{n}} \sum_{t=0}^{n/q_2-1} 1 \\
&= \frac{\sqrt{n}}{q_2}.
\end{aligned}
\tag{123}
$$

Thus,

$$\hat{\beta}^{(2)} = \frac{\sqrt{n}}{q_2} \sum_{i=0}^{q_2-1} e_{1+in/q_2}. \tag{124}$$

So $\hat{\beta}^{(2)}$ has the same structure as $\beta^{(2)}$ but with a different period. Thus,

$$\left\|\hat{\beta}^{(2)}\right\|_1 = \sqrt{n}, \qquad \left\|\hat{\beta}^{(2)}\right\|_2 = \sqrt{\frac{n}{q_2}}, \tag{125}$$

and

$$
\begin{aligned}
R_{CNN(1)}(\beta^{(2)}) &= 2\sqrt{n}\left\|\hat{\beta}^{(2)}\right\|_2 = \frac{2n}{\sqrt{q_2}}, \\
R_{CNN(q_2)}(\beta^{(2)}) &\geq 2\sqrt{\frac{n}{q_2}}\sqrt{\sum_{t=1}^{n/q_2}(\sum_{j\in\mathcal{S}_t(q_2)}|\hat{\beta}_j|)^2} = \frac{2n}{\sqrt{q_2}}.
\end{aligned}
\tag{126}
$$

Since $1 \leq q_1 \leq q_2$,

$$R_{CNN(q_1)}(\beta^{(2)}) = R_{CNN(q_2)}(\beta^{(2)}) = \frac{2n}{\sqrt{q_2}}. \tag{127}$$

$\square$

By Eq. (125), the $l_1$ norm of the Fourier transform of $\beta^{(2)}$ is independent of the periodicity $q_2$. Thus, we immediately get

$$\left\|\hat{\beta}^{(1)}\right\|_1 = (1-\epsilon)\left\|\hat{\beta}^{(2)}\right\|_1 = (1-\epsilon)\sqrt{n}, \tag{128}$$

since $\beta^{(1)}/(1-\epsilon)$ only differs from $\beta^{(2)}$ in the periodicity.

Next, we look into the representation costs of two predictors $\beta^{(1)}$ and $\beta^{(2)}$ to see how the induced complexity measure of CNN changes with $q$.

**Example** Consider $q_1|q_2, q_2|n$. Let $e_j \in \mathbb{R}^n$ be the $j$th standard basis vector and $\epsilon > 0$ be a constant such that

$$(1-\epsilon)\sqrt{q_2} > \sqrt{q_1}. \tag{129}$$

Let

$$\beta^{(1)} = (1-\epsilon) \sum_{j=0}^{n/q_1-1} e_{1+jq_1} \quad \text{and} \quad \beta^{(2)} = \sum_{j=0}^{n/q_2-1} e_{1+jq_2}.$$

Then

$$R_{CNN(q_1)}(\beta^{(1)}) > R_{CNN(q_1)}(\beta^{(2)}) \quad \text{but} \quad R_{CNN(q_2)}(\beta^{(1)}) < R_{CNN(q_2)}(\beta^{(2)}). \tag{130}$$

*Proof of Eq. (130).* By Lemma 19, Eq. (119), Eq. (129), and Eq. (128), we have

$$R_{CNN(q_1)}(\beta^{(1)}) = \frac{2(1-\epsilon)n}{\sqrt{q_1}}, \quad R_{CNN(q_2)}(\beta^{(1)}) = \frac{2(1-\epsilon)n}{\sqrt{q_2}},$$
$$R_{CNN(q_1)}(\beta^{(2)}) = \frac{2n}{\sqrt{q_2}}, \qquad R_{CNN(q_2)}(\beta^{(2)}) = \frac{2n}{\sqrt{q_2}}. \tag{131}$$

Thus,

$$R_{CNN(q_1)}(\beta^{(1)}) > R_{CNN(q_1)}(\beta^{(2)})$$
$$R_{CNN(q_2)}(\beta^{(1)}) < R_{CNN(q_2)}(\beta^{(2)}). \tag{132}$$

$\square$

Next, we use Eq. (130) to construct a data set and look into the minimum representation cost interpolation of the data. We design our data so that this interpolation changes with $q$.

**Example** We consider a generalized linear regression model. Let $q_1|q_2, q_2|n$. Let

$$K = \mathcal{S}_1(n/q_1) - \mathcal{S}_1(n/q_2), \qquad k = |K|, \tag{133}$$

where

$$\mathcal{S}_1(q) = \left\{1 + j\frac{n}{q} : j \in \{0, 1, \ldots, q-1\}\right\}. \tag{134}$$

For instance, when $n = 12, q_1 = 2$, and $q_2 = 4$, $\mathcal{S}_1(n/q_1) = \{1, 3, 5, 7, 9, 11\}$ and $\mathcal{S}_1(n/q_2) = \{1, 5, 9\}$. Recall that

$$\beta^{(1)} = (1-\epsilon) \sum_{j=0}^{n/q_1-1} e_{1+jq_1}, \qquad \beta^{(2)} = \sum_{j=0}^{n/q_2-1} e_{1+jq_2}, \tag{135}$$

where $e_j \in \mathbb{R}^n$ is the $j$th standard basis vector. Thus, $\mathcal{S}_1(n/q_1)$ and $\mathcal{S}_1(n/q_2)$ are the supports of $\beta^{(1)}$ and $\beta^{(2)}$ respectively. Let $\beta \in \mathbb{R}^n, x \in \mathbb{R}^{3n}$. Let $x = (x_1, x_2, x_3)$ where $x_1, x_2, x_3 \in \mathbb{R}^n$. Consider the model

$$\phi(\beta, x) = \beta^T x_1 + (\beta^2 - (1-\epsilon)\beta)^T x_2 + (\beta^2 - (2-\epsilon)\beta + 1 - \epsilon)^T x_3, \tag{136}$$

where $\beta^2$ denote the entry-wise squaring of $\beta$. We will construct a sample such that the vectors $\beta$ that achieve zero loss are $\beta^{(1)}$ and $\beta^{(2)}$. To achieve this, it suffice to have the following conditions:

$$
\begin{aligned}
&\beta[j] = 0 && \forall j \notin \mathcal{S}_1(n/q_1) \\
&\beta[j] \in \{1, 1-\epsilon\} && \forall j \in \mathcal{S}_1(n/q_2) \\
&\beta[j] \in \{0, 1-\epsilon\} && \forall j \in K \\
&\beta[j] = \beta[j+q_2] && \forall j \in \mathcal{S}_1(n/q_2) \\
&\sum_{j \in \mathcal{S}_1(n/q_2)} \beta[j] + \frac{q_1 \epsilon}{(q_2-q_1)(1-\epsilon)} \sum_{j \in K} \beta[j] = \frac{n}{q_2}.
\end{aligned}
\tag{137}
$$

The only interesting condition is the last one. Note that if we don't have the last condition, then it might be the case that we have some vector which chooses 1 for its coordinates in $\mathcal{S}_1(n/q_2)$ and $1 - \epsilon$ for its coordinates in $K$. We don't want this to happen. With the last condition, for any vector $\beta$ such that $\beta[j] = 1$ for $j \in \mathcal{S}_1(n/q_2)$, $\sum_{j \in \mathcal{S}_1(n/q_2)} \beta[j]$ is already $n/q_2$. Thus, all the remaining entries have to be 0. Also, for any vector $\beta$ such that $\beta[j] = 1 - \epsilon$ for $j \in \mathcal{S}_1(n/q_2)$, $\sum_{j \in \mathcal{S}_1(n/q_2)} \beta[j]$ is less than $n/q_2$ and $\beta[j]$ has to be $1 - \epsilon$ for all $j \in K$ in order to satisfy the last condition.

Now we give the construction for the sample which forces $\beta$ to satisfy the conditions (137). We will use $((x_1, x_2, x_3), y)$ to denote an element in the sample $S$. Let

$$
\begin{aligned}
T_1 &= \{((e_j, 0, 0), 0) : j \notin \mathcal{S}_1(n/q_1)\}, \\
T_2 &= \{((0, 0, e_j), 0) : j \in \mathcal{S}_1(n/q_2)\}, \\
T_3 &= \{((0, e_j, 0), 0) : j \in K\}, \\
T_4 &= \{((e_j - e_{j+q_2}, 0, 0), 0) : j \in \mathcal{S}_1(n/q_2)\}, \\
T_5 &= \left\{ \left( \sum_{j \in \mathcal{S}_1(n/q_2)} e_j + \frac{q_1 \epsilon}{(q_2-q_1)(1-\epsilon)} \sum_{j \in K} e_j, 0, 0 \right), n/q_2 \right) \right\}.
\end{aligned}
\tag{138}
$$

Let

$$
S = \bigcup_{t=1}^{5} T_t
\tag{139}
$$

be the sample. Let

$$
\mathcal{W} = \{\beta \in \mathbb{R}^n : \phi(\beta, x) = y \quad \forall (x, y) \in S\}
\tag{140}
$$

be the set of interpolating solutions. Then

$$
\mathcal{W} = \{\beta^{(1)}, \beta^{(2)}\}.
\tag{141}
$$

For each $q | n$, let

$$
V(q) = \{\beta' \in \mathcal{W} : R_{CNN(q)}(\beta') = \min_{\beta \in \mathcal{W}} R_{CNN(q)}(\beta)\}.
\tag{142}
$$

By equation (130), we see that

$$
V(q_1) = \{\beta^{(2)}\}, \qquad V(q_2) = \{\beta^{(1)}\}.
\tag{143}
$$

### B.3 CNN with sum pooling

In a convolutional neural network with sum pooling, we put an extra *sum pooling* layer before the fully connected layer. The *sum pooling* layer corresponds to a circulant matrix $A$ with respect to a vector $a = (1, 1, \ldots, 1, 0, 0, \ldots, 0) \in \mathbb{R}^n$, which is supported on the first $k$ entries, where $k$ is the width of the pooling region. Recall that a circulant matrix $C$ with respect to a vector $c \in \mathbb{R}^n$ is defined as $C = \frac{1}{\sqrt{n}} \begin{pmatrix} c[1] & c[2] & \cdots & c[n] \\ c[n] & c[1] & \cdots & c[n-1] \\ \vdots & \vdots & & \vdots \\ c[2] & c[3] & \cdots & c[1] \end{pmatrix}$. As before, let $w = (w_1, w_2, \cdots w_{d-1}, W_d)$ be the parameters of a convolutional neural network, where $w_i \in \mathbb{R}^q \times \{0\}^{n-q}$ for $i \in [d-1]$

and $W_d \in \mathbb{R}^{m \times n}$. For each $i \in [d-1]$, let $W_i$ be the circulant matrix with respect to $w_i$. In a convolutional neural network with sum pooling,

$$F_{SCNN(q,k)}(w) = W_d A \prod_{i=1}^{d-1} W_{d+1-i}. \tag{144}$$

For any matrix $\boldsymbol{\beta} \in \mathbb{R}^{m \times n}$ and $q \in [n]$, let $R_{SCNN(q,k)}(\boldsymbol{\beta}) := R_{F_{SCNN(q,k)}}(\boldsymbol{\beta})$ be the representation cost of $\boldsymbol{\beta}$ in a convolutional neural network with sum pooling of filter width $q$ and pooling width $k$. As what we did for CNN without sum pooling, we can use the discrete Fourier transform matrix $\mathsf{F}$ to diagonalize $A$ and $W_i$s. Similar to Eq. (26),

$$\hat{\boldsymbol{\beta}} = \hat{W}_d D \prod_{i=1}^{d-1} D_{d+1-i}, \tag{145}$$

where $D = \mathrm{diag}(\hat{a})$ and $D_i = \mathrm{diag}(\hat{w}_i)$. Now we consider the cases $q = n, m = 1$ and $q = 1, m = 1$. We use $\beta$ in place of $\boldsymbol{\beta}$ in these cases.

### B.3.1  Full filter width case: $q = n, m = 1$

**Theorem 25.** *For any $\beta \in \mathbb{R}^n$,*

$$R_{SCNN(n,k)}(\beta) = d \sum_{i=1}^{n} \left(\frac{|\hat{\beta}_i|}{|\hat{a}_i|}\right)^{2/d}, \tag{146}$$

*where $\hat{\beta}$ and $\hat{a}$ are the Fourier transforms of $\beta$ and $a$ respectively and*

$$|\hat{a}_j|^2 = \frac{1 - \cos(2\pi k(j-1)/n)}{1 - \cos(2\pi(j-1)/n)}. \tag{147}$$

*Proof.* As before, we know that the optimal weights $w_j$ would be identical by the same reason as before. Then we can assume that

$$w_1 = w_2 = \cdots = w_{d-1} = c \tag{148}$$

for some $c$. By (145), we have

$$\hat{\beta}_i = \hat{W}_d[i] \hat{a}_i \hat{c}[i]^{d-1}. \tag{149}$$

Thus,

$$\hat{W}_d[i] = \frac{\hat{\beta}_i}{\hat{a}_i \hat{c}[i]^{d-1}}. \tag{150}$$

Thus,

$$R_{SCNN(n,k)}(\beta) = \min_{\hat{c}} \sum_{i=1}^{n} \left((d-1)|\hat{c}[i]|^2 + \frac{|\hat{\beta}_i|^2}{|\hat{a}_i|^2 |\hat{c}[i]|^{2(d-1)}}\right)$$

$$= \sum_{i=1}^{n} \min_{\hat{c}[i]} \left((d-1)|\hat{c}[i]|^2 + \frac{|\hat{\beta}_i|^2}{|\hat{a}_i|^2 |\hat{c}[i]|^{2(d-1)}}\right) \tag{151}$$

$$\overset{(a)}{=} d \sum_{i=1}^{n} \left(\frac{|\hat{\beta}_i|}{|\hat{a}_i|}\right)^{2/d},$$

where $(a)$ follows from AM-GM inequality. Note that

$$\hat{a}_j = \sum_{t=0}^{k-1} \omega^{(j-1)t} = \frac{1 - \omega^{(j-1)k}}{1 - \omega^{j-1}}, \tag{152}$$

where $\omega := e^{2\pi i/n}$. Since

$$|1 - \omega^p|^2 = (1 - \cos p(2\pi/n))^2 + \sin^2 p(2\pi/n) = 2 - 2\cos p(2\pi/n), \tag{153}$$

we have

$$|\hat{a}_j|^2 = \frac{1 - \cos(2\pi k(j-1)/n)}{1 - \cos(2\pi(j-1)/n)}. \tag{154}$$

$\square$

**B.3.2 Filter width one case:** $q = 1, m = 1$

**Theorem 26.** *For any $\beta \in \mathbb{R}^n$,*

$$R_{SCNN(1,k)}(\beta) = dn^{(d-1)/d}\left(\sum_{i=1}^{n} \frac{|\hat{\beta}_i|^2}{|\hat{a}_i|^2}\right)^{1/d}, \tag{155}$$

*where $\hat{\beta}$ and $\hat{a}$ are the Fourier transforms of $\beta$ and $a$ respectively and*

$$|\hat{a}_j|^2 = \frac{1 - \cos(2\pi k(j-1)/n)}{1 - \cos(2\pi(j-1)/n)}. \tag{156}$$

Note that the induced complexity measure in this case is some weighted $l_2$ norm.

*Proof.* As before, we know that the optimal weights $w_j$ would be identical by the same reason as before. Then we can assume that

$$w_1 = w_2 = \cdots = w_{d-1} = c \tag{157}$$

for some $c$. By (145), we have

$$\hat{\beta}_i = \hat{W}_d[i]\hat{a}_i\hat{c}[i]^{d-1}. \tag{158}$$

Since $q = 1$, we know that

$$\hat{c}[1] = \hat{c}[2] = \cdots = \hat{c}[n] = \frac{1}{\sqrt{n}}c[1]. \tag{159}$$

Thus,

$$\hat{\beta}_i = \hat{W}_d[i]\hat{a}_i\left(\frac{1}{\sqrt{n}}c[1]\right)^{d-1}. \tag{160}$$

Thus,

$$\hat{W}_d[i] = \frac{\hat{\beta}_i}{\hat{a}_i\left(\frac{1}{\sqrt{n}}c[1]\right)^{d-1}}. \tag{161}$$

Thus,

$$R_{SCNN(1,k)}(\beta) = \min_{c[1]}\left((d-1)c[1]^2 + \sum_{i=1}^{n} \frac{n^{d-1}|\hat{\beta}_i|^2}{|\hat{a}_i|^2 c[1]^{2(d-1)}}\right)$$
$$\overset{(a)}{=} dn^{(d-1)/d}\left(\sum_{i=1}^{n} \frac{|\hat{\beta}_i|^2}{|\hat{a}_i|^2}\right)^{1/d}, \tag{162}$$

where $(a)$ follows from AM-GM inequality. Note that

$$\hat{a}_j = \sum_{t=0}^{k-1} \omega^{(j-1)t} = \frac{1 - \omega^{(j-1)k}}{1 - \omega^{j-1}}, \tag{163}$$

where $\omega := e^{2\pi i/n}$. Since

$$|1 - \omega^p|^2 = (1 - \cos p(2\pi/n))^2 + \sin^2 p(2\pi/n) = 2 - 2\cos p(2\pi/n), \tag{164}$$

we have

$$|\hat{a}_j|^2 = \frac{1 - \cos(2\pi k(j-1)/n)}{1 - \cos(2\pi(j-1)/n)}. \tag{165}$$

$\square$

**B.4 CNN with multiple channels**

In a convolutional neural network with $n_c$ channels,

$$F_{MCNN(q)}(W) = \sum_{i=1}^{n_c} F_{CNN(q)}(w_i), \tag{166}$$

where $W = (w_1, \ldots, w_{n_c})$ are the parameters of the $n_c$ parallel convolutional neural networks. For any $\beta \in \mathbb{R}^n$, let $R_{MCNN(q)}(\beta) := R_{F_{MCNN(q)}}(\beta)$ be the representation cost of $\beta$ under $F_{MCNN(q)}$. Surprisingly, CNN with multiple channels have the same representation cost as CNN (with one channel), when $q = n$ or $q = 1$.

### B.4.1 Full filter width case: $q = n$

We first consider the case $q = n$.

**Theorem 27.** *For $\beta \in \mathbb{R}^n$,*

$$R_{MCNN(n)}(\beta) = d \sum_{j=1}^{n} |\hat{\beta}_j|^{2/d} = R_{CNN(n)}(\beta). \tag{167}$$

*Proof.* Let $w^* \in \mathbb{R}^n$ be such that

$$\|w^*\|_2^2 = R_{CNN(n)}(\beta), \quad \text{and} \quad F_{CNN(n)}(w^*) = \beta. \tag{168}$$

Then, taking $W^* = (w^*, 0, \ldots, 0)$, we have

$$\|W^*\|_2^2 = R_{CNN(n)}(\beta), \quad \text{and} \quad F_{MCNN(n)}(W^*) = \beta. \tag{169}$$

Thus,

$$R_{MCNN(n)}(\beta) \leq \|W^*\|_2^2 = R_{CNN(n)}(\beta). \tag{170}$$

Let $\tilde{W} = (\tilde{w}_1, \ldots, \tilde{w}_{n_c})$ be such that

$$\left\|\tilde{W}\right\|_2^2 = R_{MCNN(n)}(\beta). \tag{171}$$

For each $i \in [n_c]$, let

$$p_i = F_{CNN(n)}(\tilde{w}_i). \tag{172}$$

Then

$$\beta = \sum_{i=1}^{n_c} p_i. \tag{173}$$

Thus, Then

$$\hat{\beta} = \sum_{i=1}^{n_c} \hat{p}_i. \tag{174}$$

When $d = 2$, we have

$$\begin{aligned}
R_{MCNN(n)}(\beta) &= \left\|\tilde{W}\right\|_2^2 \\
&= \sum_{i=1}^{n_c} \|\tilde{w}_i\|_2^2 \\
&\geq \sum_{i=1}^{n_c} R_{CNN(n)}(p_i) \\
&= \sum_{i=1}^{n_c} 2\|\hat{p}_i\|_1 \\
&\geq 2\left\|\hat{\beta}\right\|,
\end{aligned} \tag{175}$$

where the last step follows from (174) and triangle inequality.

Now suppose that $d > 2$. Then we have

$$
\begin{aligned}
R_{MCNN(n)}(\beta) &= \left\| \tilde{W} \right\|_2^2 \\
&= \sum_{i=1}^{n_c} \| \tilde{w}_i \|_2^2 \\
&\geq \sum_{i=1}^{n_c} R_{CNN(n)}(p_i) \\
&= \sum_{i=1}^{n_c} d \sum_{j=1}^{n} |\hat{p}_i[j]|^{2/d} \\
&\overset{(a)}{\geq} d \sum_{j=1}^{n} | \sum_{i=1}^{n_c} \hat{p}_i[j]|^{2/d} \\
&= d \sum_{j=1}^{n} |\hat{\beta}_j|^{2/d}.
\end{aligned}
\tag{176}
$$

To see $(a)$, it suffices to show that for each $j$,

$$
\sum_{i=1}^{n_c} |\hat{p}_i[j]|^{2/d} \geq | \sum_{i=1}^{n_c} \hat{p}_i[j]|^{2/d}.
\tag{177}
$$

By triangle inequality, it suffices to show that

$$
\sum_{i=1}^{n_c} |\hat{p}_i[j]|^{2/d} \geq ( \sum_{i=1}^{n_c} |\hat{p}_i[j]|)^{2/d},
\tag{178}
$$

which is equivalent to

$$
( \sum_{i=1}^{n_c} |\hat{p}_i[j]|^{2/d})^{d/2} \geq \sum_{i=1}^{n_c} |\hat{p}_i[j]|.
\tag{179}
$$

This follows directly from Taylor's theorem and the fact that $d > 2$. $\qquad\square$

### B.4.2 Filter width one case: $q = 1$

Now we consider the case $q = 1$.

**Theorem 28.** *For $\beta \in \mathbb{R}^n$,*

$$
R_{MCNN(1)}(\beta) = d n^{(d-1)/d} \|\beta\|_2^{2/d} = R_{CNN(1)}(\beta).
\tag{180}
$$

*Proof.* The proof is exactly the same as in the $q = n$ case. It follows from triangle inequality and the fact that

$$
\sum_{i=1}^{t} |a_i|^c \geq ( \sum_{i=1}^{t} |a_i|)^c
\tag{181}
$$

when $c < 1$. $\qquad\square$

### B.5 An architecture similar to CNN

In this section, we consider the representation cost of an architecture similar to CNN, whose induced complexity measure is $l_{1,2}$ norm on Fourier domain. For simplicity we consider depth two neural network with single output. Let $q$ be a hyper-parameter, which is analogous to the filter width in CNN. We assume that $q|n$. Let $w = (w_2, w_1)$ be the parameters of this architecture, where $w_2 \in \mathbb{R}^n$ and $w_1 \in (\mathbb{R} \times \{0\}^{n/q-1})^q$. Let $W_1$ be the circulant matrix with respect to $w_1$. In this architecture,

$$
F_{LCNN(q)}(w) = w_2^T W_1.
\tag{182}
$$

For $\beta \in \mathbb{R}^n$, let $R_{LCNN(q)}(\beta) := R_{F_{LCNN(q)}}(\beta)$ be the representation cost of $\beta$ under $F_{LCNN(q)}$.

**Theorem 29.** *For $\beta \in \mathbb{R}^n$,*

$$R_{LCNN(q)}(\beta) = 2\sqrt{\frac{n}{q}} \sum_{t=1}^{q} \sqrt{\sum_{j \in \mathcal{S}_t} |\hat{\beta}_j|^2}, \tag{183}$$

*where $\mathcal{S}_t = \{t + kq : k = 0, 1, \ldots, n/q - 1\}$, and $\hat{\beta}$ is the Fourier transform of $\beta$.*

So the induced complexity measure of this architecture is $l_{1,2}$ norm on Fourier domain.

Before giving the proof, we first make some observations which reduce this problem to the CNN case. Let $\mathsf{F}_n \in \mathbb{C}^{n \times n}$ be the $n \times n$ discrete Fourier transform matrix defined by $\mathsf{F}_n[j, k] = \frac{1}{\sqrt{n}}\omega_n^{(j-1)(k-1)}$, where $\omega_n = e^{2\pi i/n}$. Let $\mathsf{F}'$ be the submatrix of $\mathsf{F}_n$ obtained by taking the $1, 1+n/q, 1+2n/q, \ldots, 1+(q-1)n/q$th columns of $\mathsf{F}_n$. Then we have

$$\mathsf{F}' = \sqrt{\frac{q}{n}} \begin{bmatrix} \mathsf{F}_q \\ \mathsf{F}_q \\ \vdots \\ \mathsf{F}_q \end{bmatrix}, \tag{184}$$

where $\mathsf{F}_q$ is the $q \times q$ discrete Fourier transform matrix. In other words, $\mathsf{F}'$ is a stack of smaller discrete Fourier transform matrices up to some scaling. Now let $u \in R^q$ be the subvector of $c$ obtained by taking the $1, 1+n/q, 1+2n/q, \ldots, 1+(q-1)n/q$th entries of $w_1$. Since $\text{supp}(w_1) \subseteq \{1 + vn/q : v = 0, 1, \ldots, q-1\}$,

$$\hat{w}_1 := \mathsf{F}_n w_1 = \mathsf{F}' u = \sqrt{\frac{q}{n}} \begin{bmatrix} \mathsf{F}_q u \\ \mathsf{F}_q u \\ \vdots \\ \mathsf{F}_q u \end{bmatrix} = \sqrt{\frac{q}{n}} \begin{bmatrix} \hat{u} \\ \hat{u} \\ \vdots \\ \hat{u} \end{bmatrix}, \tag{185}$$

where $\hat{u} = \mathsf{F}_q u$ is the Fourier transform of $u$.

Similar to Eq. (26), if $\beta^T = w_2^T C$, then $\hat{\beta}^T = \hat{w}_2^T D$, where $D = \text{diag}\,\hat{w}_1$. Then, we have

$$\hat{\beta} = D\hat{w}_2. \tag{186}$$

Thus, for all $j \in [n]$,

$$\hat{w}_2[j] = \frac{\hat{\beta}_j}{\hat{w}_1[j]} \tag{187}$$

Note that
$$\|w\|_2^2 = \|w_2\|_2^2 + \|w_1\|_2^2 = \|\hat{w}_2\|_2^2 + \|u\|_2^2 = \|\hat{w}_2\|_2^2 + \|\hat{u}\|_2^2, \tag{188}$$
since $u$ is the subvector of $w_1$ by taking the entries in $\text{supp}(w_1)$.

*Proof of Theorem 29.* By Eq. (187) and Eq. (188),

$$
\begin{aligned}
R_{LCNN(q)}(\beta) &= \min_u \sum_{t=1}^{q} |\hat{u}_t|^2 + \sum_{j=1}^{n} \frac{|\hat{\beta}_j|^2}{|\hat{w}_1[j]|^2} \\
&\overset{(a)}{=} \min_u \sum_{t=1}^{q} |\hat{u}_t|^2 + \frac{n}{q} \sum_{t=1}^{q} \frac{\sum_{j \in \mathcal{S}_t} |\hat{\beta}_j|^2}{|\hat{u}_t|^2} \\
&= \min_u \sum_{t=1}^{q} \left( |\hat{u}_t|^2 + \frac{n}{q} \frac{\sum_{j \in \mathcal{S}_t} |\hat{\beta}_j|^2}{|\hat{u}_t|^2} \right) \\
&\overset{(b)}{=} 2\sqrt{\frac{n}{q}} \sum_{t=1}^{q} \sqrt{\sum_{j \in \mathcal{S}_t} |\hat{\beta}_j|^2},
\end{aligned}
\tag{189}
$$

where $(a)$ follows from Eq. (185), and $(b)$ follows from AM-GM inequality. $\qquad\square$

# C  Supplementary materials for residual networks

## C.1  Proofs of general results of residual networks

We give the proofs of Theorem 4 and Theorem 12.

**Theorem 4.** *Suppose that $d_1 < d_2 < \cdots < d_k$. Then, $R_{ResNet}(\lambda\beta)/R_1(\lambda\beta) \longrightarrow 1$ as $\lambda \to 0$, and $R_{ResNet}(\lambda\beta)/R_k(\lambda\beta) \longrightarrow 1$ as $\lambda \to \infty$.*

*Proof.* $\boldsymbol{\lambda \to 0}$ **Case:** We will first show that

$$R_{ResNet}(\beta) \leq R_1(\beta), \tag{190}$$

for all $\beta \in \mathbb{R}^n$. To see this, let $w \in \mathbb{R}^p$ be the weights such that

$$F_{\mathcal{N}_1}(w) = \beta, \quad \text{and} \quad \|w\|_2^2 = R_1(\beta). \tag{191}$$

Since $w$ achieves the minimum representation cost with respect to $\mathcal{N}_1$, for any edge in $\mathcal{N}_{ResNet}$ but not in $\mathcal{N}_1$, its weight has to be 0. Since $d_j > d_1$ for all $j > 1$, each $\mathcal{N}_j$ has at least two adjacent layers whose edges in between are assigned zero weights. Thus, $F_{\mathcal{N}_j}(w) = 0$ for all $j > 1$. Thus, $F_{\mathcal{N}_{ResNet}}(w) = \beta$. Thus, Eq. (190) holds.

Then, fix $\beta \in \mathbb{R}^n$. We will show that

$$\liminf_{\lambda \to 0} \frac{R_{ResNet}(\lambda\beta)}{R_1(\lambda\beta)} \geq 1. \tag{192}$$

Let $f : \mathbb{R}_+ \longrightarrow \mathbb{R}$ be defined as

$$f(\lambda) = R_{ResNet}(\lambda\beta). \tag{193}$$

The restriction to $\lambda \geq 0$ does not compromise our statement since $\lambda\beta = (-\lambda)(-\beta)$ (for $\lambda < 0$ we just apply the argument to $-\lambda$ and $-\beta$). Let $w_\lambda \in \mathbb{R}^p$ be weights such that

$$F_{\mathcal{N}_{ResNet}}(w_\lambda) = \lambda\beta, \quad \text{and} \quad \|w_\lambda\|_2^2 = R_{ResNet}(\lambda\beta). \tag{194}$$

Let $E$ be the set of edges in $\mathcal{N}_{ResNet}$. For each $e \in E$, let $w_\lambda(e)$ be the weights on $e$ in $w_\lambda$. Then,

$$w_\lambda(e)^2 \leq R_{ResNet}(\lambda\beta) \overset{(a)}{\leq} R_1(\lambda\beta) \overset{(c)}{=} \lambda^{2/d_1} R_1(\beta), \tag{195}$$

where $(a)$ follows from Eq. (190) and $(c)$ follows from Lemma 37. Thus,

$$|w_\lambda(e)| = O(\lambda^{1/d_1}) \quad \text{as} \quad \lambda \to 0. \tag{196}$$

Since $\mathcal{N}_j$ is of depth $d_j > d_1$ for all $j > 1$,

$$\left\|F_{\mathcal{N}_j}(w_\lambda)\right\|_2 = O(\lambda^{d_j/d_1}) = o(\lambda) \quad \text{as} \quad \lambda \to 0. \tag{197}$$

Thus,

$$\left\|\sum_{j=2}^{k} F_{\mathcal{N}_j}(w_\lambda))\right\|_2 = o(\lambda) \quad \text{as} \quad \lambda \to 0. \tag{198}$$

Note that

$$F_{\mathcal{N}_1}(w_\lambda) = \lambda\beta - \sum_{j=2}^{k} F_{\mathcal{N}_j}(w_\lambda). \tag{199}$$

Thus,

$$\begin{aligned}
R_{ResNet}(\lambda\beta) &= \|w_\lambda\|_2^2 \\
&\geq R_1(\lambda\beta - \sum_{j=2}^{k} F_{\mathcal{N}_j}(w_\lambda)) \\
&= \lambda^{2/d_1} R_1(\beta - (\sum_{j=2}^{k} F_{\mathcal{N}_j}(w_\lambda))/\lambda) \\
&\overset{(a)}{=} \lambda^{2/d_1} R_1(\beta - o(1)),
\end{aligned} \tag{200}$$

where $(a)$ follows from Eq. (198). Thus,

$$\frac{R_{ResNet}(\lambda\beta)}{R_1(\lambda\beta)} \geq \frac{\lambda^{2/d_1} R_1(\beta - o(1))}{\lambda^{2/d_1} R_1(\beta)} = \frac{R_1(\beta - o(1))}{R_1(\beta)}. \tag{201}$$

Taking $\liminf$ on both sides of Eq. (201), we get

$$\liminf_{\lambda \to 0} \frac{R_{ResNet}(\lambda\beta)}{R_1(\lambda\beta)} \geq \liminf_{\lambda \to 0} \frac{R_1(\beta - o(1))}{R_1(\beta)} = 1, \tag{202}$$

where in the last step we used Lemma 43. By equations (190) and (202),

$$\frac{R_{ResNet}(\lambda\beta)}{R_1(\lambda\beta)} \longrightarrow 1, \quad \text{as} \quad \lambda \to 0. \tag{203}$$

$\boldsymbol{\lambda \to \infty}$ **Case:** Fix $\beta \in \mathbb{R}^n$. We will first show that

$$\limsup_{\lambda \to \infty} \frac{R_{ResNet}(\lambda\beta)}{R_k(\lambda\beta)} \leq 1. \tag{204}$$

First, we assume that $\beta_i \neq 0$ for all $i \in [n]$. Let $w \in \mathbb{R}^p$ be the weights such that

$$F_{\mathcal{N}_k}(w) = \beta, \quad \text{and} \quad \|w\|_2^2 = R_k(\beta). \tag{205}$$

Let $w^{(\lambda)}$ be the weights obtained from $w$ by multiplying all the weights by $\lambda^{1/d_k}$. Then,

$$F_{\mathcal{N}_k}(w^{(\lambda)}) = \lambda\beta, \quad \text{and} \quad \left\|w^{(\lambda)}\right\|_2^2 = \lambda^{2/d_k} \|w\|_2^2 = \lambda^{2/d_k} R_k(\beta) = R_k(\lambda\beta), \tag{206}$$

by Lemma 37. Since the weights $w^{(\lambda)}$ on each edge is $O(\lambda^{1/d_k})$ and the depth of $\mathcal{N}_j$ is $d_j < d_k$,

$$\left\|F_{\mathcal{N}_j}(w^{(\lambda)})\right\|_2 = O(\lambda^{d_j/d_k}) = o(\lambda), \quad \text{as} \quad \lambda \to \infty, \tag{207}$$

for all $j < k$. Thus,

$$\sum_{j=1}^{k-1} \left\|F_{\mathcal{N}_j}(w^{(\lambda)})\right\|_2 = O(\lambda^{(d_k-1)/d_k}) = o(\lambda), \quad \text{as} \quad \lambda \to \infty. \tag{208}$$

Thus, there exists $M > 0$ such that

$$\lambda|\beta_i| > \sum_{j=1}^{k-1} |F_{\mathcal{N}_j}(w^{(\lambda)})[i]| \quad \forall i \in [n], \tag{209}$$

for all $\lambda > M$. Since we will let $\lambda \to \infty$, we can assume that $\lambda$ is always greater than $M$.

If there exists $t \in [d]$ such that $\mathcal{N}_k$ skips the $t$th layer, then the weights $w$ much be zero on the $t$th layer. Hence, $w^{(\lambda)}$ is also zero on the $t$th layer. Thus, any subnetwork $\mathcal{N}_j$ that does not skip the $t$th layer much satisfy $F_{\mathcal{N}_j}(w^{(\lambda)}) = 0$. Thus, to simplify the notation, we might assume without loss of generality that $\mathcal{N}_k$ does not skip any layer.

Let $\mathsf{S}_1 = \{j \in [k] : 1 \in I_j\}$ be the indices of subnetworks that do not skip the first layer. By the last paragraph, $k \in \mathsf{S}_1$. Let $\mathsf{S}_2 = [k] - \mathsf{S}_1$ be the indices of subnetworks that skip the first layer.

Let $s_1, s_2, \ldots s_n$ be some real numbers to be decided later. Let $\tilde{w}^{(\lambda)}(s_1, s_2, \ldots s_n)$ be the weights obtained from $w^{(\lambda)}$ by multiplying the weights of the edges connected to the $i$th input node by $1 + s_i$ for all $i \in [n]$. Since $\tilde{w}^{(\lambda)}(s_1, s_2, \ldots s_n)$ differs from $w^{(\lambda)}$ only in the first layer, for all $i \in [n]$,

$$F_{\mathcal{N}_j}(\tilde{w}^{(\lambda)}(s_1, s_2, \ldots s_n))[i] = (1 + s_i)F_{\mathcal{N}_j}(w^{(\lambda)})[i] \tag{210}$$

for all $j \in \mathsf{S}_1$ and

$$F_{\mathcal{N}_j}(\tilde{w}^{(\lambda)}(s_1, s_2, \ldots s_n))[i] = F_{\mathcal{N}_j}(w^{(\lambda)})[i] \tag{211}$$

for all $j \in \mathsf{S}_2$. By the above equations, for all $i \in [n]$,

$$F_{\mathcal{N}}(\tilde{w}^{(\lambda)}(s_1, s_2, \ldots s_n))[i] = (1 + s_i) \sum_{j \in \mathsf{S}_1} F_{\mathcal{N}_j}(w^{(\lambda)})[i] + \sum_{j \in \mathsf{S}_2} F_{\mathcal{N}_j}(w^{(\lambda)})[i]. \tag{212}$$

By equation (209), $F_{\mathcal{N}}(w^{(\lambda)})[i]$ and $\lambda\beta_i$ have the same sign for all $i \in [n]$. By equations (208) and (212), for all $i \in [n]$,

$$F_{\mathcal{N}}(\tilde{w}^{(\lambda)}(s_1, s_2, \ldots s_n))[i] = (1+s_i)(\lambda\beta_i+o(\lambda))+o(\lambda) = \lambda[(1+s_i)(\beta_i+o(1))+o(1)], \quad \text{as} \quad \lambda \to \infty. \tag{213}$$

By equation (213), if we solve for $F_{\mathcal{N}}(\tilde{w}^{(\lambda)}(s_1, s_2, \ldots s_n))[i] = \lambda\beta_i$, then we get

$$s_i = \frac{o(1)}{\beta_i + o(1)} = o(1), \quad \text{as} \quad \lambda \to \infty. \tag{214}$$

By equation (214), for each $i \in [n]$, there exists $s_i = o(1)$, such that

$$F_{\mathcal{N}_{ResNet}}(\tilde{w}^{(\lambda)}(s_1, s_2, \ldots s_n)) = \lambda\beta. \tag{215}$$

Thus,

$$R_{ResNet}(\lambda\beta) \le \left\|\tilde{w}^{(\lambda)}(s_1, s_2, \ldots s_n)\right\|_2^2 \le (1 + \max_{i \in [n]} |s_i|)^2 \left\|w^{(\lambda)}\right\|_2^2 = (1 + o(1))R_k(\lambda\beta), \tag{216}$$

where the last step follows from Eq. (206) and the fact that $s_i^2 = o(1)$ and $|s_i| = o(1)$ for all $i \in [n]$. Thus,

$$\limsup_{\lambda \to \infty} \frac{R_{ResNet}(\lambda\beta)}{R_k(\lambda\beta)} \le \limsup_{\lambda \to \infty} \frac{(1 + o(1))R_k(\lambda\beta)}{R_k(\lambda\beta)} = \limsup_{\lambda \to \infty} 1 + o(1) = 1. \tag{217}$$

Now, we drop the condition $\beta_i \neq 0$ for all $i \in [n]$. Let $0 < \epsilon < 1$. Let $w \in \mathbb{R}^p$ be the weights such that

$$F_{\mathcal{N}_k}(w) = \beta, \quad \text{and} \quad \|w\|_2^2 = R_k(\beta). \tag{218}$$

Let $w(\epsilon)$ be the weights obtained from $w$ by adding $\epsilon$ to the weights on each edge. Let $\beta(\epsilon) = F_{\mathcal{N}_k}(w(\epsilon))$. Then, there exists a constant $C > 0$, such that for all $i \in [n]$,

$$|\beta(\epsilon)_i - \beta_i| < C\epsilon. \tag{219}$$

Without loss of generality, assume that $\beta(\epsilon)_i \neq 0$ for all $i \in [n]$. Let $w^{(\lambda)}(\epsilon)$ be the weights obtained from $w(\epsilon)$ by multiplying all the weights by $\lambda^{1/d_k}$. Let $w^{(\lambda)}$ be the weights obtained from $w$ by multiplying all the weights by $\lambda^{1/d_k}$. Then,

$$F_{\mathcal{N}_k}(w^{(\lambda)}) = \lambda\beta, \quad \text{and} \quad \left\|w^{(\lambda)}\right\|_2^2 = \lambda^{2/d_k}\|w\|_2^2 = \lambda^{2/d_k}R_k(\beta) = R_k(\lambda\beta), \tag{220}$$

by Lemma 37.

By the same argument as in the previous case, we get something similar to Eq (213)

$$F_{\mathcal{N}}(\tilde{w}^{(\lambda)}(\epsilon)(s_1, s_2, \ldots s_n))[i] = (1+s_i)(\lambda\beta(\epsilon)_i+o(\lambda))+o(\lambda) = \lambda[(1+s_i)(\beta(\epsilon)_i+o(1))+o(1)], \quad \text{as} \quad \lambda \to \infty. \tag{221}$$

Now, we solve for $F_{\mathcal{N}}(\tilde{w}^{(\lambda)}(\epsilon)(s_1, s_2, \ldots s_n))[i] = \lambda\beta_i$. By Equations (221) and (219), we get for all $i \in [n]$,

$$|s_i| = |\frac{\beta_i - \beta(\epsilon)_i + o(1)}{\beta(\epsilon)_i}| \le |\frac{C\epsilon + o(1)}{\beta(\epsilon)_i}| \le C'\epsilon + o(1), \tag{222}$$

for some constant $C' > 0$.

By equation (222), for each $i \in [n]$, there exists $s_i$, such that $|s_i| \le C'\epsilon + o(1)$ and

$$F_{\mathcal{N}_{ResNet}}(\tilde{w}^{(\lambda)}(s_1, s_2, \ldots s_n)) = \lambda\beta. \tag{223}$$

Thus,

$$\begin{aligned} R_{ResNet}(\lambda\beta) &\le \left\|\tilde{w}^{(\lambda)}(\epsilon)(s_1, s_2, \ldots s_n)\right\|_2^2 \\ &\le (1 + \max_{i \in [n]} |s_i|)^2 \left\|w^{(\lambda)}(\epsilon)\right\|_2^2 \\ &\le (1 + \max_{i \in [n]} |s_i|)^2 (\left\|w^{(\lambda)}\right\|_2^2 + C''\lambda^{2/d_k}\epsilon) \\ &= (1 + o(1))(R_k(\lambda\beta) + C''\lambda^{2/d_k}\epsilon), \end{aligned} \tag{224}$$

for some $C'' > 0$, where the last step follows from Eq. (220) and the fact that $s_i^2 = o(1)$ and $|s_i| = o(1)$ for all $i \in [n]$. Thus,

$$
\begin{aligned}
\limsup_{\lambda \to \infty} \frac{R_{ResNet}(\lambda\beta)}{R_k(\lambda\beta)} &\leq \limsup_{\lambda \to \infty} \frac{(1 + o(1))(R_k(\lambda\beta) + C''\lambda^{2/d_k}\epsilon))}{R_k(\lambda\beta)} \\
&= \limsup_{\lambda \to \infty} (1 + o(1) + \frac{C''\lambda^{2/d_k}\epsilon}{\lambda^{2/d_k}R_k(\beta)}) \\
&= 1 + C'''\epsilon,
\end{aligned}
\tag{225}
$$

for some $C''' > 0$. Since $\epsilon \in (0, 1)$ is arbitrary, we get

$$
\limsup_{\lambda \to \infty} \frac{R_{ResNet}(\lambda\beta)}{R_k(\lambda\beta)} \leq 1.
\tag{226}
$$

On the other hand, we will show that

$$
\liminf_{\lambda \to \infty} \frac{R_{ResNet}(\lambda\beta)}{R_k(\lambda\beta)} \geq 1.
\tag{227}
$$

As before, let $\beta \in \mathbb{R}^n$ be fixed and let $f : \mathbb{R}_+ \longrightarrow \mathbb{R}$ be defined as

$$
f(\lambda) = R_{ResNet}(\lambda\beta).
\tag{228}
$$

Let $w_\lambda \in \mathbb{R}^p$ be weights such that

$$
F_{\mathcal{N}_{ResNet}}(w_\lambda) = \lambda\beta, \quad \text{and} \quad \|w_\lambda\|_2^2 = R_{ResNet}(\lambda\beta).
\tag{229}
$$

Let $E$ be the set of edges in $\mathcal{N}_{ResNet}$. For each $e \in E$, let $w_\lambda(e)$ be the weights on $e$ in $w_\lambda$. Then,

$$
w_\lambda(e)^2 \leq R_{ResNet}(\lambda\beta) \overset{(a)}{\leq} (1 + o(1))R_k(\lambda\beta) \overset{(c)}{=} (1 + o(1))\lambda^{2/d_k}R_k(\beta),
\tag{230}
$$

where $(a)$ follows from Eq. (224) and $(c)$ follows from Lemma 37. Thus,

$$
|w_\lambda(e)| = O(\lambda^{1/d_k}) \quad \text{as} \quad \lambda \to \infty.
\tag{231}
$$

Since $\mathcal{N}_j$ is of depth $d_j < d_k$ for all $j < k$,

$$
\|F_{\mathcal{N}_j}(w_\lambda)\|_2 = O(\lambda^{d_j/d_k}) = o(\lambda) \quad \text{as} \quad \lambda \to \infty,
\tag{232}
$$

for all $j < k$. Thus,

$$
\left\| \sum_{j=1}^{k-1} F_{\mathcal{N}_j}(w_\lambda) \right\|_2 = o(\lambda) \quad \text{as} \quad \lambda \to \infty.
\tag{233}
$$

Note that

$$
F_{\mathcal{N}_k}(w_\lambda) = \lambda\beta - \sum_{j=1}^{k-1} F_{\mathcal{N}_j}(w_\lambda).
\tag{234}
$$

Thus,

$$
\begin{aligned}
R_{ResNet}(\lambda\beta) &= \|w_\lambda\|_2^2 \\
&\geq R_k(\lambda\beta - \sum_{j=1}^{k-1} F_{\mathcal{N}_j}(w_\lambda)) \\
&= \lambda^{2/d_k} R_k(\beta - (\sum_{j=1}^{k-1} F_{\mathcal{N}_j}(w_\lambda))/\lambda) \\
&\overset{(a)}{=} \lambda^{2/d_k} R_k(\beta - o(1)),
\end{aligned}
\tag{235}
$$

where $(a)$ follows from Eq. (233). Thus,

$$
\frac{R_{ResNet}(\lambda\beta)}{R_k(\lambda\beta)} \geq \frac{\lambda^{2/d_k}R_k(\beta - o(1))}{\lambda^{2/d_k}R_k(\beta)} = \frac{R_k(\beta - o(1))}{R_k(\beta)}.
\tag{236}
$$

Taking $\liminf$ on both sides of Eq. (236), we get

$$\liminf_{\lambda \to \infty} \frac{R_{ResNet}(\lambda \beta)}{R_k(\lambda \beta)} \geq \liminf_{\lambda \to \infty} \frac{R_k(\beta - o(1))}{R_k(\beta)} = 1, \tag{237}$$

where in the last step we used Lemma 43. By equations (204) and (237),

$$\frac{R_{ResNet}(\lambda \beta)}{R_k(\lambda \beta)} \longrightarrow 1, \quad \text{as} \quad \lambda \to \infty. \tag{238}$$

$\square$

**Theorem 12.** *Suppose that $d_1 < d_2 < \cdots < d_k$. Let $h : \mathbb{R}^n \longrightarrow \mathbb{R}$ be a homogeneous function. If $F_{\mathcal{N}_{ResNet}}$ induces $h$ as induced complexity measure, then $F_{\mathcal{N}_1}$ also induces $h$ as induced complexity measure.*

*Proof.* Suppose that $\mathcal{N}_{ResNet}$ induces $h$ as induced complexity measure. Let $e_1 = (1, 0, \ldots, 0)^T \in \mathbb{R}^n$ and $\beta' \in \mathbb{R}^n$ be any vector such that $h(e_1) = h(\beta')$. Since $h$ is homogeneous, $h(\lambda e_1) = h(\lambda \beta')$ for all $\lambda > 0$. Since $\mathcal{N}_{ResNet}$ induces $h$ as induced complexity measure and $h(\lambda e_1) = h(\lambda \beta')$,

$$R_{ResNet}(\lambda e_1) = R_{ResNet}(\lambda \beta') \tag{239}$$

for all $\lambda > 0$. By Theorem 4,

$$\lim_{\lambda \to 0} \frac{R_{ResNet}(\lambda e_1)}{R_1(\lambda e_1)} = 1, \quad \text{and} \quad \lim_{\lambda \to 0} \frac{R_{ResNet}(\lambda \beta')}{R_1(\lambda \beta')} = 1. \tag{240}$$

By equations (239) and (240),

$$\lim_{\lambda \to 0} \frac{R_1(\lambda \beta')}{R_1(\lambda e_1)} = 1. \tag{241}$$

By Lemma 37 and Eq. (241),

$$1 = \lim_{\lambda \to 0} \frac{R_1(\lambda \beta')}{R_1(\lambda e_1)} = \lim_{\lambda \to 0} \frac{\lambda^{2/d_1} R_1(\beta')}{\lambda^{2/d_1} R_1(e_1)} = \lim_{\lambda \to 0} \frac{R_1(\beta')}{R_1(e_1)} = \frac{R_1(\beta')}{R_1(e_1)}. \tag{242}$$

Let $L$ be the degree of homogeneity of $h$. Since induced complexity measure is invariant up to monotonic transformations, we may assume without loss of generality that $h(e_1) = 1$. Then, for any $\beta \in \mathbb{R}^n$,

$$\begin{aligned}
R_1(\beta) &= R_1(h(\beta)^{1/L} \beta / h(\beta)^{1/L}) \\
&\overset{(a)}{=} h(\beta)^{2/Ld_1} R_1(\beta / h(\beta)^{1/L}) \\
&\overset{(c)}{=} h(\beta)^{2/Ld_1} R_1(e_1) \\
&\overset{(d)}{=} d_1 h(\beta)^{2/Ld_1},
\end{aligned} \tag{243}$$

where $(a)$ follows from Lemma 37, $(c)$ follows from Eq. (242), and $(d)$ follows from Lemma 46. Thus, $\mathcal{N}_1$ induces $h$ as induced complexity measure. $\square$

### C.2   A simple ResNet

Now, we give an example of a simple ResNet. In this simple ResNet,

$$F_{SResNet}(w) = w_2 + \text{diag } w_1 w_2, \tag{244}$$

where $w = (w_1, w_2)$ is the weights of the network. For $i \in [2]$, $w_i \in \mathbb{R}^n$. Let $R_{SResNet} := R_{F_{SResNet}}$ be the representation cost under $F_{SResNet}$ defined in Eq. (1).

In this simple ResNet, there are two component networks $\mathcal{N}_1$ and $\mathcal{N}_2$, with $I_1 = \{1\}$ and $I_2 = \{1, 2\}$. In this case, the first subnetwork $\mathcal{N}_1$ skips the first diagonal layer, while the second subnetwork $\mathcal{N}_2$ goes through both layers. Note that $\mathcal{N}_1$ induces $l_2$ norm while $\mathcal{N}_2$ induces $l_1$ norm. In light of Theorem 4, we should expect that the representation cost $R_{SResNet}$ interpolates between $l_2$ norm and $l_1$ norm.

**Theorem 30.** *For $\beta \in \mathbb{R}^n$, $R_{SResNet}(\beta) = \sum_{i=1}^n (r(\beta_i)^2 + \frac{\beta_i^2}{(r(\beta_i)+1)^2})$, where $r(\gamma)$ is the unique positive real root of the equation $x(x+1)^3 = \gamma^2$.*

*Proof.* Since $\beta = w_2 + \text{diag}(w_1)w_2$, we have

$$\beta_i = w_2[i](w_1[i] + 1). \tag{245}$$

Thus,

$$w_2[i] = \frac{\beta_i}{w_1[i] + 1}. \tag{246}$$

Let $f(x, \gamma) = x^2 + \gamma^2/(x+1)^2$. Note that

$$R_{SResNet}(\beta) = \min_{x_1,\ldots,x_n} \sum_{i=1}^n f(x_i, \beta_i) = \sum_{i=1}^n \min_{x_i} f(x_i, \beta_i). \tag{247}$$

Thus, in order to compute $R_{SResNet}(\beta)$, it suffices to find the minimum of $f(x, \gamma)$ with respect to $x$. Taking the derivative with respect to $x$ and set it to 0, we get

$$2x - \frac{2\gamma^2}{(x+1)^3} = 0, \tag{248}$$

which implies

$$x(x+1)^3 = \gamma^2. \tag{249}$$

Since

$$f''(x) = 2 + \frac{6\gamma^2}{(x+1)^4} > 0, \tag{250}$$

there is a unique minimum for $f$. For each $\gamma \in \mathbb{R}$, let

$$r(\gamma) = \arg\min f(x). \tag{251}$$

Then we claim that

$$r(\gamma) \geq 0. \tag{252}$$

Suppose that $r(\gamma) < 0$. Then

$$\begin{aligned}
f(-r(\gamma), \gamma) &= r(\gamma)^2 + \frac{\gamma^2}{(|r(\gamma)| + 1)^2} \\
&< r(\gamma)^2 + \frac{\gamma^2}{(r(\gamma) + 1)^2} \\
&= f(r(\gamma), \gamma),
\end{aligned} \tag{253}$$

which contradicts the definition of $r(\gamma)$. Then $r(\gamma)$ is a positive root to Eq. (249). We claim that Eq. (249) could only have one positive real roots. Suppose that Eq. (249) have two distinct real roots $x_1 > x_2 > 0$. Let

$$q(x) = x(x+1)^3. \tag{254}$$

Then

$$q(x_1) = q(x_2) = \gamma^2. \tag{255}$$

However,

$$q'(x) = 4x^3 + 9x^2 + 6x + 1 > 0, \tag{256}$$

when $x > 0$. Thus,

$$q(x_1) > q(x_2), \tag{257}$$

which is a contradiction. Thus, $r(\gamma)$ is the unique positive real root of Eq. (249). Let $g(\gamma) = \min_x f(x, \gamma)$. Then we have

$$g(\gamma) = f(r(\gamma), \gamma) = r(\gamma)^2 + \frac{\gamma^2}{(r(\gamma) + 1)^2}. \tag{258}$$

Then the result follows since $R_{SResNet}(\beta) = \sum_{i=1}^n g(\beta_i)$. $\square$

We give some plots to show the behavior of $g(\gamma) := r(\gamma)^2 + \frac{\gamma^2}{(r(\gamma)+1)^2}$ in Figure 2, which shows that the representation cost $R_{SResNet}$ transits from $l_2$ norm to $l_1$ norm. This is in accordance with Theorem 4.

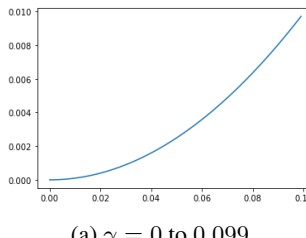 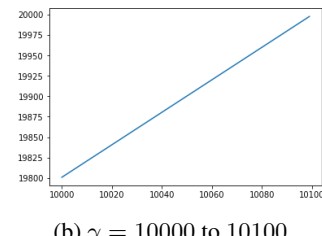 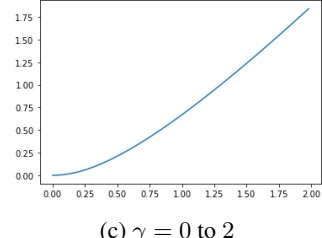

(a) $\gamma = 0$ to $0.099$      (b) $\gamma = 10000$ to $10100$      (c) $\gamma = 0$ to $2$

Figure 2: Representation Cost of a simple ResNet

# D   Supplementary materials of depth two neural networks

In this section, we study the representation cost of two layer neural networks $\mathcal{N}$. In particular, we will show that $R_{\mathcal{N}}$ is always a norm and will characterize $R_{\mathcal{N}}$ through both its primal and dual norms. From the primal characterization, we will find architectures that induce $k-$support norms as induced complexity measure. From the dual characterization, we will find architectures that induce $l_{2,1}$ norm as induced complexity measure.

## D.1   Primal characterization of representation costs of depth two networks

Note that a two layer linear homogeneous feedforward neural network $\mathcal{N}$ can be represented by a graph $G = (V, E)$ with vertex set $V = N_I \cup N_H \cup \{O\}$, where $|N_I| = n$ denotes the nodes in the input layer, $N_H$ denotes the nodes in the hidden layer, and $O$ denote the single output node. Two vertices are adjacent in $G$ if and only if the corresponding nodes are connected by an edge in $\mathcal{N}$. Since $\mathcal{N}$ has only one node in the output layer, we may assume without loss of generality that all nodes in the hidden layer are connected to the node in the output layer. Now for each $h \in N_H$, let

$$S_h = \{i \in N_I : (i, h) \in E\} \tag{259}$$

denote the set of nodes in the input layer that are adjacent to $h$. Note that this definition agrees with the one given in neural networks of general depth in equation (4). Let $w$ be the weights on the neural network $\mathcal{N}$. We define $F_{\mathcal{N}}(w) \in \mathbb{R}^n$ to be the vector corresponding to the linear predictor generated by $w$. For any $\beta \in \mathbb{R}^n$, we define

$$R_{\mathcal{N}}(\beta) = \min\{\|w\|_2^2 : F_{\mathcal{N}}(w) = \beta\} \tag{260}$$

to be the representation cost of $\beta$. Now, we give a characterization of the representation cost.

**Lemma 5.** *For a depth-two linear homogeneous feedforward neural network $\mathcal{N}$ without shared weights, $R_{\mathcal{N}}(\beta) = 2 \min\{\sum_{h \in N_H} \|v_h\|_2 : \operatorname{supp}(v_h) \subseteq S_h, \sum_{h \in N_H} v_h = \beta\}$.*

In the above lemma, as we shall see in the proof, each $v_h$ denote the vector corresponding to the linear predictor generated by part of the network $\mathcal{N}$.

*Proof.* Let $w$ be the weights of this neural network. We partition this network as follows. For each $h \in N_H$, let $w_h$ be the weights obtained from $w$ by keeping the weights of $w$ for all the edges that are adjacent to the node $h$ (including the one between $h$ and the output node $O$) and put all the rest of the weights to $0$. Let

$$v_h = F_{\mathcal{N}}(w_h). \tag{261}$$

Then we have

$$F_{\mathcal{N}}(w) = \sum_{h \in N_H} v_h, \quad \operatorname{supp}(v_h) \subseteq S_h, \quad \text{and} \quad \|w\|_2^2 = \sum_{h \in N_H} \|w_h\|_2^2. \tag{262}$$

Let $w_1, \ldots, w_{|S_h|}$ be the weights in $w_h$ corresponding to edges between nodes in the input layer and $h$. Let $\lambda_h$ be the weights on the edge between $h$ and the output node $O$. Then

$$
\begin{aligned}
\min\{\|w_h\|_2^2 : F_{\mathcal{N}}(w_h) = v_h\} &= \min\left\{\lambda_h^2 + \sum_{i=1}^{|S_h|} w_i^2 : \lambda_h w_i = v_h[i], \quad \forall i \in [n]\right\} \\
&= \min\left\{\lambda_h^2 + \sum_{i=1}^{|S_h|} v_h[i]^2/\lambda_h^2 : \lambda_h > 0\right\} \\
&\stackrel{(a)}{=} 2\|v_h\|_2,
\end{aligned}
\tag{263}
$$

where $(a)$ follows from AM-GM inequality and the fact that $\mathrm{supp}(v_h) \subseteq S_h$. Thus,

$$
\begin{aligned}
R_{\mathcal{N}}(\beta) &= \min\{\|w\|_2^2 : F_{\mathcal{N}}(w) = \beta\} \\
&= \min\left\{\sum_{h \in N_H} \|w_h\|_2^2 : \sum_{h \in N_H} F_{\mathcal{N}}(w_h) = \beta\right\} \\
&\stackrel{(b)}{=} \min\left\{\sum_{h \in N_H} \min\{\|w_h\|_2^2 : F_{\mathcal{N}}(w_h) = v_h\} : \mathrm{supp}(v_h) \subseteq S_h, \sum_{h \in N_H} v_h = \beta\right\} \\
&\stackrel{(c)}{=} \min\left\{\sum_{h \in N_H} 2\|v_h\|_2 : \mathrm{supp}(v_h) \subseteq S_h, \sum_{h \in N_H} v_h = \beta\right\} \\
&= 2\min\left\{\sum_{h \in N_H} \|v_h\|_2 : \mathrm{supp}(v_h) \subseteq S_h, \sum_{h \in N_H} v_h = \beta\right\},
\end{aligned}
\tag{264}
$$

where $(b)$ follows from the fact that $w_h$s have disjoint support, and $(c)$ follows from Eq. (263).

$\square$

Note that the above result implies that $R_{\mathcal{N}}(\cdot)$ is a norm. Next, we study its dual norm. For simplicity, let $\|\cdot\|_{\mathcal{N}} = R_{\mathcal{N}}(\cdot)/2$ and $\|\cdot\|_{\mathcal{N}}^*$ denote the dual norm of $\|\cdot\|_{\mathcal{N}}$. Then, the dual norm of $R_{\mathcal{N}}(\cdot)$ satisfies:

$$
R_{\mathcal{N}}^*(\cdot) = \frac{1}{2}\|\cdot\|_{\mathcal{N}}^*.
\tag{265}
$$

## D.2 Dual characterization of representation costs of depth two networks

**Lemma 6.** *For a depth-two linear homogeneous feedforward neural network $\mathcal{N}$ without shared weights, $R_{\mathcal{N}}^*(\beta) = \frac{1}{2}\max\{(\sum_{i \in S_h} \beta_i^2)^{1/2} : h \in N_H\}$.*

*Proof.* By definition of dual norm,

$$
\|\beta\|_{\mathcal{N}}^* = \max\{(\langle a, \beta\rangle : \|a\|_{\mathcal{N}} \le 1\}.
\tag{266}
$$

First, we pick $a \in \mathbb{R}^n$ such that $\|a\|_{\mathcal{N}} \le 1$. Thus, there exists $v_h$'s such that

$$
\sum_{h \in N_H} v_h = a, \qquad \mathrm{supp}(v_h) \subseteq S_h, \qquad \sum_{h \in N_H} \|v_h\|_2 \le 1.
\tag{267}
$$

Thus,

$$
\langle a, \beta\rangle \stackrel{(267)}{=} \sum_{h \in N_H} \langle v_h, \beta\rangle \stackrel{(a)}{\le} \sum_{h \in N_H} \|v_h\|_2 \left(\sum_{i \in S_h} \beta_i^2\right)^{1/2} \stackrel{(267)}{\le} \max\left\{\left(\sum_{i \in S_h} \beta_i^2\right)^{1/2} : h \in N_H\right\},
\tag{268}
$$

where we used Cauchy's inequality and $\mathrm{supp}(v_h) \subseteq S_h$ (267) in $(a)$. By equations (266) and (268),

$$
\|\beta\|_{\mathcal{N}}^* \le \max\left\{\left(\sum_{i \in S_h} \beta_i^2\right)^{1/2} : h \in N_H\right\}.
\tag{269}
$$

For the other direction, let $h^* \in N_h$ be such that

$$\left( \sum_{i \in S_{h^*}} \beta_i^2 \right)^{1/2} = \max \left\{ \left( \sum_{i \in S_h} \beta_i^2 \right)^{1/2} : h \in N_H \right\}. \tag{270}$$

Then, let $a^* \in \mathbb{R}^n$ be such that

$$a^*[i] = \frac{\mathbf{1}_{i \in S_{h^*}} \beta_i}{\sqrt{\sum_{i \in S_{h^*}} \beta_i^2}}. \tag{271}$$

By Lemma 5,

$$\|a\|_{\mathcal{N}} \leq 1, \tag{272}$$

since $\operatorname{supp}(a) \subseteq S_{h^*}$ and $\|a\|_2 = 1$. By equations (266) and (270),

$$\|\beta\|_{\mathcal{N}}^* \geq \langle a^*, \beta \rangle = \left( \sum_{i \in S_{h^*}} \beta_i^2 \right)^{1/2} = \max \left\{ \left( \sum_{i \in S_h} \beta_i^2 \right)^{1/2} : h \in N_H \right\}. \tag{273}$$

Thus,

$$R_{\mathcal{N}}^*(\beta) = \frac{1}{2} \|\beta\|_{\mathcal{N}}^* = \frac{1}{2} \max \left\{ \left( \sum_{i \in S_h} \beta_i^2 \right)^{1/2} : h \in N_H \right\}. \tag{274}$$

$\square$

By the above result, if there exists $h_1, h_2 \in N_H$ such that $h_1 \neq h_2$ and $S_{h_1} \subseteq S_{h_2}$, then removing $h_1$ from $\mathcal{N}$ would not change the representation cost since

$$\sum_{i \in S_{h_1}} \beta_i^2 \leq \sum_{i \in S_{h_2}} \beta_i^2 \tag{275}$$

for all $\beta \in \mathbb{R}^n$. Thus, we might assume without loss of generality that

$$S_{h_1} \not\subseteq S_{h_2} \qquad \forall h_1 \neq h_2, \quad h_1, h_2 \in N_H. \tag{276}$$

Moreover, the above result shows that the norms that can be induced by two layer neural networks as induced complexity measure are precisely the dual norms of $l_{\infty,2}$ group norms with possibly, overlapping between groups.

Now, we will give some applications of the primal and dual characterizations of the representation cost $R_{\mathcal{N}}(\beta)$ of two layer neural networks.

### D.3 $k$-support norms

In this section, we give an architecture of a two layer linear neural network which induces the $k$-support norm as induced complexity measure. For $\beta \in \mathbb{R}^n$ and $k \in [n]$, define the $k$-support norm as

$$\|\beta\|_k^{sp} = \min \left\{ \sum_{I \in \mathcal{G}_k} \|v_I\|_2 : \operatorname{supp}(v_I) \subseteq I, \sum_{I \in \mathcal{G}_k} v_I = \beta \right\}, \tag{277}$$

where $\mathcal{G}_k$ denotes the set of subsets of $[n]$ of size at most $k$. Now we define the architecture. For each $I \in \mathcal{G}_k, I \neq \emptyset$, there is a node $u_I$ in the hidden layer which is connected to the $i$th node in the input layer if and only if $i \in I$. The output layer has one node which is connected to all nodes in the hidden layer. Note that in the definition of $k$-support norm, we could define $\mathcal{G}_k$ to be the set of subsets of $[n]$ of size exactly $k$ by the remarks after Lemma 6.

Let $w \in \mathbb{R}^N$ be the weights of the networks, where $N$ is the total number of edges. Let $F_{ksp}(w) \in \mathbb{R}^n$ be the predictor obtained by weights $w$ on this network. Let

$$R_{ksp}(\beta) = \min \{ \|w\|_2^2 : F_{ksp}(w) = \beta \} \tag{278}$$

denote the representation cost of $\beta \in \mathbb{R}^n$ for the architecture defined above. By Lemma 5, we immediately get the following result.

**Theorem 31.** *For any $\beta \in \mathbb{R}^n$,*

$$R_{ksp}(\beta) = 2 \|\beta\|_k^{sp}. \tag{279}$$

We prove a result that is stronger than Theorem 11.

**Theorem 32.** *For any $k \in [n]$, there exists a homogeneous feedforward depth two linear neural network without shared weights that induces $k$-support norm as induced complexity measure. Furthermore, a homogeneous feedforward linear depth two neural network $\mathcal{N}$ without shared weights induces $k$-support norm if and only if the mixing depths of subsets $S$ satisfy*

$$M_{\mathcal{N}}(S) = \left\{ \begin{array}{ll} 1 & if \quad |S| \leq k; \\ 2 & if \quad |S| > k. \end{array} \right. \tag{280}$$

*In particular, $k$-balanced network induces $k$-support norm.*

*Proof.* The "if" part is a direct consequence of Theorem 31 and the remarks after Lemma 6.

For the "only if" part, suppose that $\mathcal{N}$ induces $k$-support norm. By Eq. 277, for any $\beta \in \mathbb{R}^n$ such that $|\operatorname{supp}(\beta)| = k$, $\|\beta\|_k^{sp} = \|\beta\|_2$. This implies that for $S := \operatorname{supp}(\beta)$, the subnetwork $\mathcal{N}_S$ induces $l_2$ norm by Lemma 44. Thus, by Theorem 35, $\mathcal{M}_{N_S}(S) = 1$, which implies that there exists $v \in N_1$ such that $S \subseteq S_v$. Thus,

$$M_{\mathcal{N}}(S) = 1 \quad if \quad |S| \leq k. \tag{281}$$

For the other case, suppose that there exist $S' \subseteq [n]$ and $u \in N_1$ such that $|S'| = k + 1$ and $S' \subseteq S_u$. Then, for any $\beta' \in \mathbb{R}^n$ such that $\operatorname{supp}(\beta') = S'$,

$$R_{\mathcal{N}}(\beta') = 2\|\beta'\|_2, \tag{282}$$

by Lemma 46. Now, take $\beta \in \mathbb{R}^n$ such that $|\operatorname{supp}(\beta)| = k$ and $\|\beta\|_2 = \|\beta'\|_2$. By Eq. 281, Eq. 282 and Lemma 46, we have

$$R_{\mathcal{N}}(\beta) = 2\|\beta\|_2 = 2\|\beta'\|_2 = R_{\mathcal{N}}(\beta'). \tag{283}$$

Since $\mathcal{N}$ induces $k$-support norm,

$$\|\beta'\|_k^{sp} = \|\beta\|_k^{sp} = \|\beta\|_2 = \|\beta'\|_2. \tag{284}$$

However, since $|\operatorname{supp}(\beta')| > k$, $\|\beta'\|_k^{sp} > \|\beta'\|_2$ by definition of $k$-support norm in Eq. 277, and triangle inequality. This is a contradiction. Thus,

$$M_{\mathcal{N}}(S) = 2 \quad if \quad |S| > k. \tag{285}$$
$\square$

For the last claim, if $\mathcal{N}$ is a $k$-balanced network, then

$$M_{\mathcal{N}}(S) = \left\{ \begin{array}{ll} 1 & if \quad |S| \leq k; \\ 2 & if \quad |S| > k. \end{array} \right. \tag{286}$$

Thus, $\mathcal{N}$ induces $k$-support norm by arguments above.

### D.4 $l_{2,1}$ norms

This section introduces the architecture with $l_{2,1}$ group norm as the induced complexity measure. Let $G_1, \ldots, G_k$ be a partition of $[n]$. Let

$$\mathcal{C} = \prod_{j=1}^{k} G_j = G_1 \times G_2 \times \cdots \times G_k \tag{287}$$

be the Cartesian product of the $k$ groups. This architecture has a node $u_h$ for each $h \in \mathcal{C}$ such that $u_h$ is connected to the $i$th input node if and only if

$$i \in S_h := \{h[j] : j \in [k]\}, \tag{288}$$

where $h[j]$ denote the element in the $j$th entry of $h$. Then we connect all the nodes in the hidden layer to the output node. See Figure 1b for an example.

Let $w \in \mathbb{R}^p$ be the weights of the networks, where $p = (k+1) \prod_{j=1}^{k} |G_j|$ is the total number of edges. Let $\mathcal{N}_{2,1}$ be the architecture defined above. Let $R_{\mathcal{N}_{2,1}} := R_{F_{\mathcal{N}_{2,1}}}$ be the representation cost under $F_{\mathcal{N}_{2,1}}$ as defined in Eq (1). By Lemma 6, we immediately get the following result.

**Corollary 33.** *For any $\beta \in R^n$,*

$$R^*_{\mathcal{N}_{2,1}}(\beta) = \frac{1}{2} \max \left\{ \left( \sum_{i \in S_h} \beta_i^2 \right)^{1/2} : h \in \mathcal{C} \right\}. \tag{289}$$

Now, we state the result.

**Theorem 34.** $R_{\mathcal{N}_{2,1}}(\beta) = 2\|\beta\|_{2,1}$.

*Proof.* It suffices to show that

$$R^*_{\mathcal{N}_{2,1}}(\beta) = \frac{1}{2}\|\beta\|_{2,\infty} = \frac{1}{2} \sqrt{\sum_{j=1}^{k} \left( \max_{i \in G_j} |\beta_i| \right)^2}. \tag{290}$$

By Corollary 33,

$$R^*_{\mathcal{N}_{2,1}}(\beta) = \frac{1}{2} \max \left\{ \left( \sum_{i \in S_h} \beta_i^2 \right)^{1/2} : h \in \mathcal{C} \right\} = \frac{1}{2} \max \left\{ \left( \sum_{j=1}^{k} \beta_{i_j}^2 \right)^{1/2} : i_j \in G_j \right\}. \tag{291}$$

For each $j \in [k]$, let $i_j^* \in G_j$ be such that

$$|\beta_{i_j^*}| = \max_{i \in G_j} |\beta_i|. \tag{292}$$

Now, in order to maximize $\sum_{j=1}^{k} \beta_{i_j}^2$, we would choose $i_j = i_j^*$ for each $j \in [k]$. Thus,

$$\max \left\{ \left( \sum_{j=1}^{k} \beta_{i_j}^2 \right)^{1/2} : i_j \in G_j \right\} = \left( \sum_{j=1}^{k} \beta_{i_j^*}^2 \right)^{1/2} = \sqrt{\sum_{j=1}^{k} \left( \max_{i \in G_j} |\beta_i| \right)^2}. \tag{293}$$

Thus,

$$R^*_{\mathcal{N}_{2,1}}(\beta) = \frac{1}{2} \sqrt{\sum_{j=1}^{k} \left( \max_{i \in G_j} |\beta_i| \right)^2}. \tag{294}$$

Since the dual of the dual norm is the primal norm,

$$R_{\mathcal{N}_{2,1}}(\beta) = (R^*_{\mathcal{N}_{2,1}})^*(\beta) = 2\|\beta\|^*_{2,\infty} = 2\|\beta\|_{2,1}. \tag{295}$$

$\square$

**Intuition of Optimal Weights**

Now, we consider a special case to show some intuitions of how the weights on the network would attain minimum representation costs and properties of the representation which attains the minimum cost.

**Example** Let $n = 4, G_1 = \{1, 4\}, G_2 = \{2, 3\}$. We give a plot of this architecture:

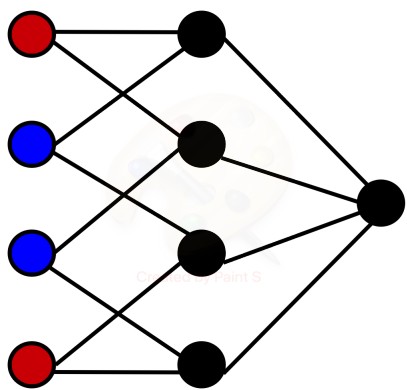

Let $\mathcal{N}$ denote the above architecture. Recall that

$$R_{\mathcal{N}}(\beta) = 2 \min \left\{ \sum_{h \in N_H} \|v_h\|_2 : \text{supp}(v_h) \subseteq S_h, \sum_{h \in N_H} v_h = \beta \right\}. \tag{296}$$

In our case, let $N_H = \{1, 2, 3, 4\}$ and

$$S_1 = \{1, 2\}, \quad S_2 = \{1, 3\}, \quad S_3 = \{2, 4\}, \quad S_4 = \{3, 4\}. \tag{297}$$

Let $\{v_i : i \in [4]\}$ be such that

$$\sum_{i=1}^{4} v_i = \beta, \qquad \text{supp}(v_i) \subseteq S_i, \quad \forall i \in [4]. \tag{298}$$

Then, we have

$$\begin{aligned}
\|\beta\|_{2,1} &:= \sqrt{(|\beta_1| + |\beta_4|)^2 + (|\beta_2| + |\beta_3|)^2} \\
&= \sqrt{(|v_1[1] + v_2[1]| + |v_3[4] + v_4[4]|)^2 + (|v_1[2] + v_3[2]| + |v_2[3] + v_4[3]|)^2} \\
&\leq \sqrt{(|v_1[1]| + |v_2[1]| + |v_3[4]| + |v_4[4]|)^2 + (|v_1[2]| + |v_3[2]| + |v_2[3]| + |v_4[3]|)^2}.
\end{aligned} \tag{299}$$

Now, note that

$$\begin{aligned}
(|v_1[1]| + |v_2[1]| + |v_3[4]| + |v_4[4]|)^2 &= \left( \|v_1\|_2^{\frac{1}{2}} \frac{|v_1[1]|}{\|v_1\|_2^{\frac{1}{2}}} + \|v_2\|_2^{\frac{1}{2}} \frac{|v_2[1]|}{\|v_2\|_2^{\frac{1}{2}}} + \|v_3\|_2^{\frac{1}{2}} \frac{|v_3[4]|}{\|v_3\|_2^{\frac{1}{2}}} + \|v_4\|_2^{\frac{1}{2}} \frac{|v_4[4]|}{\|v_4\|_2^{\frac{1}{2}}} \right)^2 \\
&\overset{(a)}{\leq} \left( \sum_{i=1}^{4} \|v_i\|_2 \right) \left( \frac{v_1[1]^2}{\|v_1\|_2} + \frac{v_2[1]^2}{\|v_2\|_2} + \frac{v_3[4]^2}{\|v_3\|_2} + \frac{v_4[4]^2}{\|v_4\|_2} \right),
\end{aligned} \tag{300}$$

where in $(a)$ we used Cauchy's inequality. Similarly,

$$\begin{aligned}
(|v_1[2]| + |v_2[3]| + |v_3[2]| + |v_4[3]|)^2 &= \left( \|v_1\|_2^{\frac{1}{2}} \frac{|v_1[2]|}{\|v_1\|_2^{\frac{1}{2}}} + \|v_2\|_2^{\frac{1}{2}} \frac{|v_2[3]|}{\|v_2\|_2^{\frac{1}{2}}} + \|v_3\|_2^{\frac{1}{2}} \frac{|v_3[2]|}{\|v_3\|_2^{\frac{1}{2}}} + \|v_4\|_2^{\frac{1}{2}} \frac{|v_4[3]|}{\|v_4\|_2^{\frac{1}{2}}} \right)^2 \\
&\overset{(b)}{\leq} \left( \sum_{i=1}^{4} \|v_i\|_2 \right) \left( \frac{v_1[2]^2}{\|v_1\|_2} + \frac{v_2[3]^2}{\|v_2\|_2} + \frac{v_3[2]^2}{\|v_3\|_2} + \frac{v_4[3]^2}{\|v_4\|_2} \right),
\end{aligned} \tag{301}$$

where in $(b)$ we used Cauchy's inequality. Now, by equation (299), we have

$$\begin{aligned}
\|\beta\|_{2,1} &\leq \sqrt{(|v_1[1]| + |v_2[1]| + |v_3[4]| + |v_4[4]|)^2 + (|v_1[2]| + |v_3[2]| + |v_2[3]| + |v_4[3]|)^2} \\
&\leq \sqrt{\left( \sum_{i=1}^{4} \|v_i\|_2 \right) \left( \frac{v_1[1]^2 + v_1[2]^2}{\|v_1\|_2} + \frac{v_2[1]^2 + v_2[3]^2}{\|v_2\|_2} + \frac{v_3[4]^2 + v_3[2]^2}{\|v_3\|_2} + \frac{v_4[4]^2 + v_4[3]^2}{\|v_4\|_2} \right)} \\
&= \sqrt{\left( \sum_{i=1}^{4} \|v_i\|_2 \right)^2} \\
&= \sum_{i=1}^{4} \|v_i\|_2.
\end{aligned} \tag{302}$$

Thus,

$$R_{\mathcal{N}}(\beta) \geq 2\|\beta\|_{2,1}. \tag{303}$$

For this lower bound to be attained, the vectors $v_i$ should satisfy

$$\begin{aligned}
\mathrm{sign}(v1[1]) &= \mathrm{sign}(v_2[1]) = \mathrm{sign}(\beta_1)\\
\mathrm{sign}(v3[4]) &= \mathrm{sign}(v_4[4]) = \mathrm{sign}(\beta_4)\\
\mathrm{sign}(v1[2]) &= \mathrm{sign}(v_3[2]) = \mathrm{sign}(\beta_2)\\
\mathrm{sign}(v2[3]) &= \mathrm{sign}(v_4[3]) = \mathrm{sign}(\beta_3),
\end{aligned} \tag{304}$$

and

$$\frac{|v_1[1]|}{|v_1[2]|} = \frac{|v_2[1]|}{|v_2[3]|} = \frac{|v_3[4]|}{|v_3[2]|} = \frac{|v_4[4]|}{|v_4[3]|}. \tag{305}$$

The first requirement ensures that the triangle inequality (in (299)) would hold with equality. The second requirement ensures that Cauchy's inequality (in $(a)$, $(b)$ in (300) and (301)) would hold with equality.

The more interesting requirement is the second one, which says that for each node in the hidden layer, the ratio of the weight it distributes to the first group to the weight it distributes to the second group is some constant which is the same for all nodes in the hidden layer. The same holds for more general $l_{2,1}$ architectures. Specifically, let $G_1, \ldots, G_k$ be a partition of $[n]$. For each $h \in \prod_{j=1}^{k} G_j$, let $v_h$ be the vector corresponding to the hidden node that corresponds to $h$. Let $p(v_h)$ be the projection of $v_h$ on the coordinates in $\{h[j] : j \in [k]\}$. Then there exists a vector $u \in \mathbb{R}^k$ such that for each $h \in \prod_{j=1}^{k} G_j$, there exists $\lambda_h \in \mathbb{R}$, such that

$$|p(v_h)| = \lambda_h u, \tag{306}$$

where the absolute value $|\cdot|$ is applied component-wise.

Intuitively, this means that each node in the hidden layer distributes weights to different groups "in the same way"(i.e there is a fixed ratio that is shared across all hidden nodes, of how to distribute weights to input nodes in different groups).

# E    Supplementary materials: mixing depths and basic properties of representation cost and induced complexity measure

We will use the following results in [15] many times:

$$R_{FC}(\beta) = d\|\beta\|_2^{2/d}, \qquad R_{DNN}(\beta) = d\|\beta\|_{2/d}^{2/d}, \tag{307}$$

where $FC$ and $DNN$ are fully connected network and diagonal network of depth $d$, respectively. In this section, we only consider networks without shared weights.

## E.1    Mixing Depths

We begin with the formal definition of *mixing depths*.

**Definition E.1 (Mixing Depths).** For any $S \subseteq [n]$, the mixing depth of $S$ with respect to $\mathcal{N}$ is defined as:

$$M_{\mathcal{N}}(S) := \min\{i \in \mathbb{N} : \text{ there exists } v \in N_i \text{ such that } S \subseteq S_v\}, \tag{308}$$

where $N_i$ is the set of nodes in the $i$-th hidden layer of $\mathcal{N}$, and $S_v$ is defined in Eq. (4).

The notion of mixing depths capture how fast information from a subset of nodes in the input layers is mixed together. Note that $M_{\mathcal{N}}(\{s\}) = 1$ for all $s \in [n]$. Thus, we only consider mixing depths of sets of size at least two.

The following theorem identifies architectures that induce $l_p$ quasi-norms as architectures with *uniform* mixing depths.

**Theorem 35.** *A linear homogeneous feedforward neural network $\mathcal{N}$ without shared weights induces $l_p$ quasi-norm if and only if $M_{\mathcal{N}}(S) = 2/p$, for all $S \subseteq [n], |S| \geq 2$.*

Note that the above theorem implies Theorem 7 since mixing depths are always integers and a diagonal network has *uniform* mixing depths.

Roughly speaking, architectures with small mixing depths usually have small representation costs, and vice versa. The following theorem is a motivating example of this intuition.

**Theorem 36.** *For all linear homogeneous feedforward neural networks $\mathcal{N}$ without shared weights and of depth $d$,*

$$R_{FC}(\beta) = d\|\beta\|_2^{2/d} \leq R_{\mathcal{N}}(\beta) \leq d\|\beta\|_{2/d}^{2/d} = R_{DNN}(\beta). \tag{309}$$

*Furthermore, the lower bound is achieved for all $\beta \in \mathbb{R}^n$ if and only if the mixing depths $M_{\mathcal{N}}(S) = 1$ for all $S \subseteq [n]$, and the upper bound is achieved for all $\beta \in \mathbb{R}^n$ if and only if the mixing depths $M_{\mathcal{N}}(S) = d$ for all $S \subseteq [n]$ such that $|S| \geq 2$.*

Note that for a fixed set of input and output nodes, fully connected networks have the smallest possible mixing depths while diagonal networks have the largest possible mixing depths. The result above shows that architectures with smallest mixing depths have the smallest representation costs while architectures with the largest mixing depths have the largest representation costs.

## E.2 Basic properties of representation cost

We give some basic properties of representation cost for homogeneous feedforward architectures $\mathcal{N}$. First, we show that the representation cost function $R_{\mathcal{N}}(\beta)$ is homogeneous.

**Lemma 37.** *Let $\mathcal{N}$ be a homogeneous feedforward neural network without shared weights and of depth $d$. Then for any $\lambda > 0$,*

$$R_{\mathcal{N}}(\lambda\beta) = \lambda^{2/d} R_{\mathcal{N}}(\beta), \tag{310}$$

*for all $\beta \in \mathbb{R}^n$.*

*Proof.* Let $w$ be weights on $\mathcal{N}$ such that

$$F_{\mathcal{N}}(w) = \beta, \quad \text{and} \quad \|w\|_2^2 = R_{\mathcal{N}}(\beta). \tag{311}$$

Then,

$$F_{\mathcal{N}}(\lambda^{1/d}w) = \lambda\beta, \quad \text{and} \quad \left\|\lambda^{1/d}w\right\|_2^2 = \lambda^{2/d}\|w\|_2^2, \tag{312}$$

since $d$ is the depth of $\mathcal{N}$. Thus,

$$R_{\mathcal{N}}(\lambda\beta) \leq \lambda^{2/d} R_{\mathcal{N}}(\beta). \tag{313}$$

On the other hand, substituting $1/\lambda$ to $\lambda$ and $\lambda\beta$ to $\beta$, we get the other direction

$$R_{\mathcal{N}}(\beta) \leq \frac{R_{\mathcal{N}}(\lambda\beta)}{\lambda^{2/d}}. \tag{314}$$

The result follows from equations (313) and (314).   $\square$

Note that the same result holds with the same proof if we put ReLU activation (or any other homogeneous activation) on $\mathcal{N}$.

Then, we show that for any weights $w$ that attains the minimum representation cost, its magnitude is "uniform" across layers in a sense we define below.

**Lemma 38.** *Let $\mathcal{N}$ be a homogeneous feedforward neural network without shared weights and of depth $d$. Let $\beta \in \mathbb{R}^n$. Let $w = (w_1, \ldots, w_d)$ be the weights on $\mathcal{N}$, where $w_i$ denotes the weights on from the $i-1$th layer $N_{i-1}$ to the $i$th layer $N_i$ such that*

$$F_{\mathcal{N}}(w) = \beta, \quad \text{and} \quad \|w\|_2^2 = R_{\mathcal{N}}(\beta). \tag{315}$$

*Then*

$$\|w_i\|_2^2 = \frac{R_{\mathcal{N}}(\beta)}{d}, \tag{316}$$

*for all $i \in [d]$.*

*Proof.* It suffices to show that $\|w_i\|_2^2 = \|w_{i+1}\|_2^2$ for all $i \in [d]$. Let $v \in N_i$ be an arbitrary node. Let $E_{i-1}^v$ be the set of edges from $N_{i-1}$ to $N_i$ that have $v$ as one of their endpoints. Let $E_i^v$ be the set of edges from $N_i$ to $N_{i+1}$ that have $v$ as one of their endpoints. For any edge $e$, let $w_e$ be the weights on it. Then we claim that

$$\sum_{e \in E_{i-1}^v} w_e^2 = \sum_{e \in E_i^v} w_e^2. \tag{317}$$

Suppose Eq. (317) does not hold. Then, by AM-GM inequality,

$$\left( \sum_{e \in E_{i-1}^v} w_e^2 \right) + \left( \sum_{e \in E_i^v} w_e^2 \right) > 2 \sqrt{\left( \sum_{e \in E_{i-1}^v} w_e^2 \right) \left( \sum_{e \in E_i^v} w_e^2 \right)}. \tag{318}$$

However, we could then scale the weights $w_e$ by $\lambda$ for all $e \in E_{i-1}^v$ and scale the weights $w_e$ by $1/\lambda$ for all $e \in E_i^v$ so that Eq. (317) holds. Let $w'$ be the weights after this modification. Then

$$F_{\mathcal{N}}(w') = \beta, \quad \text{and} \quad \|w'\|_2^2 < \|w\|_2^2 = R_{\mathcal{N}}(\beta), \tag{319}$$

which is a contradiction. Thus, Eq. (317) holds. Since $v \in N_i$ is arbitrary,

$$\|w_i\|_2^2 = \sum_{v \in N_i} \sum_{e \in E_{i-1}^v} w_e^2 = \sum_{v \in N_i} \sum_{e \in E_i^v} w_e^2 = \|w_{i+1}\|_2^2. \tag{320}$$

Since $R_{\mathcal{N}}(\beta) = \sum_{i=1}^d \|w_i\|_2^2 = d\|w_1\|_2^2$,

$$\|w_i\|_2^2 = \|w_1\|_2^2 = \frac{R_{\mathcal{N}}(\beta)}{d}, \tag{321}$$

for all $i \in [d]$. $\qquad\square$

Note that the same result holds with the same proof if we put ReLU activation (or any other homogeneous activation) on $\mathcal{N}$.

Then, we show that the representation cost $R_{\mathcal{N}}(\beta)$ only depends on $|\beta|$, where the absolute value $|\cdot|$ is applied component-wise. Furthermore, $R_{\mathcal{N}}(\beta)$ is strictly increasing in $|\beta_i|$ for all $i \in [n]$.

**Lemma 39.** *Let $\mathcal{N}$ be a homogeneous feedforward neural network without shared weights and of depth $d$. For any $\beta \in \mathcal{N}$,*

$$R_{\mathcal{N}}(\beta) = R_{\mathcal{N}}(|\beta|), \tag{322}$$

*where the absolute value $|\cdot|$ is applied component-wise. Furthermore, $R_{\mathcal{N}}(\beta)$ is strictly increasing in $|\beta_i|$ for all $i \in [n]$.*

*Proof.* Let $\beta, \beta' \in \mathbb{R}^n$ such that $|\beta_i| = |\beta_i'|$ for all $i \in [n]$. We will show that

$$R_{\mathcal{N}}(\beta) = R_{\mathcal{N}}(\beta'). \tag{323}$$

Let $w$ be weights on $\mathcal{N}$ such that

$$F_{\mathcal{N}}(w) = \beta, \quad \text{and} \quad \|w\|_2^2 = R_{\mathcal{N}}(\beta). \tag{324}$$

Then we modify the weights $w$ as follows. For each edge (between the input layer $N_0$ and the first layer $N_1$) in $\mathcal{N}$ that is connected to the $i$th input node, we scale its weight by $\text{sign}(\beta_i \beta_i')$. We do this modification for each $i \in [n]$. Let $w'$ denote the resulting weight. Then

$$F_{\mathcal{N}}(w') = \beta', \quad \text{and} \quad \|w'\|_2^2 = \|w\|_2^2 = R_{\mathcal{N}}(\beta). \tag{325}$$

Thus,

$$R_{\mathcal{N}}(\beta') \leq R_{\mathcal{N}}(\beta). \tag{326}$$

Switching the role of $\beta$ and $\beta'$, we get

$$R_{\mathcal{N}}(\beta) \leq R_{\mathcal{N}}(\beta'). \tag{327}$$

Then, Eq. (323) follows from equations (326) and (327). In particular, Eq. (323) holds for $\beta' = |\beta|$.

For the second claim, fix an $i \in [n]$. Let $\beta'' \in \mathbb{R}^n$ such that $|\beta''_j| = |\beta_j|$ for all $j \neq i$ and $|\beta''_i| < |\beta_i|$. By the first claim we just proved, we may assume that both $\beta$ and $\beta''$ have non-negative entries $(\beta, \beta'' \in \mathbb{R}^n_+)$. Then we modify the weights $w$ as follows. For each edge (between the input layer $N_0$ and the first layer $N_1$) in $\mathcal{N}$ that is connected to the $i$th input node, we scale its weight by $\beta''_i / \beta_i$. We do this modification only for the single index $i$. Let $w''$ denote the resulting weight. Then

$$F_{\mathcal{N}}(w'') = \beta'', \quad \text{and} \quad \|w''\|_2^2 < \|w\|_2^2 = R_{\mathcal{N}}(\beta). \tag{328}$$

Thus,

$$R_{\mathcal{N}}(\beta'') < R_{\mathcal{N}}(\beta). \tag{329}$$

Thus, $R_{\mathcal{N}}(\beta)$ is strictly increasing in $|\beta_i|$ for all $i \in [n]$. $\square$

In light of this result, we can always assume that $\beta \in \mathbb{R}^n_+$ when considering representation cost of homogeneous feedforward neural networks. In addition, if we are given two vectors $\beta, \beta' \in \mathbb{R}^n$ such that $|\beta_i| \leq |\beta'_i|$ for all $i \in [n]$, then $R_{\mathcal{N}}(\beta) \leq R_{\mathcal{N}}(\beta')$.

Next, we show that we can assume that the weights $w$ are always non-negative.

**Lemma 40.** *Let $\mathcal{N}$ be a homogeneous feedforward neural network without shared weights and of depth $d$. Let $\beta \in \mathbb{R}^n_+$. Then there exists non-negative weights $w$ such that*

$$F_{\mathcal{N}}(w) = \beta, \quad \text{and} \quad \|w\|_2^2 = R_{\mathcal{N}}(\beta). \tag{330}$$

*Proof.* Let $w$ be weights on $\mathcal{N}$ such that

$$F_{\mathcal{N}}(w) = \beta, \quad \text{and} \quad \|w\|_2^2 = R_{\mathcal{N}}(\beta). \tag{331}$$

We will show that $|w|$ also satisfies Eq. (331), where $|w|$ is obtained from $w$ by taking absolute values of the weights. First, note that $\|w\|_2^2 = \||w|\|_2^2$. Thus,

$$R_{\mathcal{N}}(F_{\mathcal{N}}(|w|)) \leq \||w|\|_2^2 = \|w\|_2^2 = R_{\mathcal{N}}(\beta). \tag{332}$$

On the other hand, note that

$$F_{\mathcal{N}}(|w|)[i] \geq F_{\mathcal{N}}(w)[i] = \beta_i, \tag{333}$$

for all $i \in [n]$. Thus, by Lemma 39,

$$R_{\mathcal{N}}(F_{\mathcal{N}}(|w|)) \geq R_{\mathcal{N}}(\beta). \tag{334}$$

By equations (332) and (334), we have

$$R_{\mathcal{N}}(F_{\mathcal{N}}(|w|)) = R_{\mathcal{N}}(\beta). \tag{335}$$

Since $R_{\mathcal{N}}(\beta)$ is strictly increasing in $|\beta_i|$ as stated in Lemma 39, we must have

$$F_{\mathcal{N}}(|w|) = \beta \tag{336}$$

by equations (333) and (335). $\square$

In light of Lemma 39 and Lemma 40, it suffices to consider non-negative vectors and non-negative weights when studying representation cost of homogeneous feedforward neural networks.

Then, we consider the following question: "if we perturb the vector $\beta$ a little bit, how does its representation cost change?".

**Lemma 41.** *Let $\mathcal{N}$ be a homogeneous feedforward neural network without shared weights and of depth $d$. Let $\beta, v \in \mathbb{R}^n$ such that*

$$\|v\|_2 \leq \delta < 1, \tag{337}$$

*for some $\delta \in \mathbb{R}$. Then,*

$$R_{\mathcal{N}}(\beta + v) \leq R_{\mathcal{N}}(\beta) + 2\sqrt{n R_{\mathcal{N}}(\beta)}\delta^{1/d} + |E|\delta^{2/d}, \tag{338}$$

*where $E$ is the set of edges in the neural network $\mathcal{N}$. In addition, if there exists some constant $C > 0$ such that*

$$R_{\mathcal{N}}(\beta') \leq C, \tag{339}$$

*for all $\beta' \in B_1(\beta) := \{\beta' \in \mathbb{R}^n : \|\beta' - \beta\|_2 \leq 1\}$, then*

$$|R_{\mathcal{N}}(\beta + v) - R_{\mathcal{N}}(\beta)| \leq 2\sqrt{nC}\delta^{1/d} + |E|\delta^{2/d}. \tag{340}$$

*Proof.* By Lemma 39, we can assume that $\beta_i \geq 0$ for all $i \in [n]$. Since we want to get an upper bound on $R_\mathcal{N}(\beta + v)$, we might assume that $v_i \geq 0$ for all $i \in [n]$, by Lemma 39 (since $|\beta_i + v_i| \leq \beta_i + |v_i|$). Let $w$ be weights on $\mathcal{N}$ such that

$$F_\mathcal{N}(w) = \beta, \quad \text{and} \quad \|w\|_2^2 = R_\mathcal{N}(\beta). \tag{341}$$

By Lemma 40, we can assume that the weights $w$ are non-negative. We modify the weights $w$ by adding $\delta^{1/d}$ to the weights of each edges in $\mathcal{N}$. Let $w'$ denote the resulting weights. Let $\beta' = F_\mathcal{N}(w')$. Since $\mathcal{N}$ is of depth $d$ and all the weights are non-negative,

$$\beta_i' \geq \beta_i + \delta \geq (\beta + v)[i], \tag{342}$$

for all $i \in [n]$. Thus, by Lemma 39,

$$R_\mathcal{N}(\beta + v) \leq R_\mathcal{N}(\beta') \leq \|w'\|_2^2. \tag{343}$$

Let $e$ be an edge in $\mathcal{N}$. Let $w_e$ and $w_e'$ be its weights before and after the modification. Then, note that

$$w_e'^2 = (w_e + \delta^{1/d})^2 = w_e^2 + 2w_e\delta^{1/d} + \delta^{2/d}. \tag{344}$$

Then,

$$\begin{aligned}
\|w'\|_2^2 &= \sum_{e \in E} w_e'^2 \\
&= \sum_{e \in E}(w_e^2 + 2w_e\delta^{1/d} + \delta^{2/d}) \\
&\overset{(a)}{\leq} R_\mathcal{N}(\beta) + 2\sqrt{nR_\mathcal{N}(\beta)}\delta^{1/d} + |E|\delta^{2/d},
\end{aligned} \tag{345}$$

where we used Cauchy's inequality (or $l_1$ norm is bounded by $\sqrt{n}$ times $l_2$ norm) in $(a)$. By equations (345) and (343),

$$R_\mathcal{N}(\beta + v) \leq R_\mathcal{N}(\beta) + 2\sqrt{nR_\mathcal{N}(\beta)}\delta^{1/d} + |E|\delta^{2/d}. \tag{346}$$

Then, if there exists $C > 0$ such that

$$R_\mathcal{N}(\beta') \leq C, \tag{347}$$

for all $\beta' \in B_1(\beta) := \{\beta' \in \mathbb{R}^n : \|\beta' - \beta\|_2 \leq 1\}$, then by Eq. (346), we have

$$R_\mathcal{N}(\beta + v) - R_\mathcal{N}(\beta) \leq 2\sqrt{nC}\delta^{1/d} + |E|\delta^{2/d}. \tag{348}$$

Substituting $\beta + v$ for $\beta$ and $-v$ for $v$, we have

$$R_\mathcal{N}(\beta) - R_\mathcal{N}(\beta + v) \leq 2\sqrt{nC}\delta^{1/d} + |E|\delta^{2/d}. \tag{349}$$

By equations (348) and (349),

$$|R_\mathcal{N}(\beta + v) - R_\mathcal{N}(\beta)| \leq 2\sqrt{nC}\delta^{1/d} + |E|\delta^{2/d}. \tag{350}$$

□

In light of Lemma 41, it is useful to have an upper bound on the representation cost $R_\mathcal{N}(\beta)$ that works for all architectures $\mathcal{N}$. Intuitively, an architecture $\mathcal{N}$ gives rise to high representation cost $R_\mathcal{N}(\cdot)$ if there are very few edges in it. On the other hand, for any valid architecture, there has to be a path from each input node to the output node. Thus, diagonal network seems to be the "sparsest" architecture that satisfies this condition. Indeed, as we shall see, the representation cost of a diagonal network $R_{DNN}(\beta) = d\|\beta\|_{2/d}^{2/d}$ (as shown in [15]) is an upper bound for the representation cost of any architecture of the same depth.

**Lemma 42.** *Let $\mathcal{N}$ be a homogeneous feedforward neural network without shared weights and of depth $d$. For any $\beta \in \mathbb{R}^n$,*

$$R_\mathcal{N}(\beta) \leq d\|\beta\|_{2/d}^{2/d}. \tag{351}$$

*Furthermore, the upper bound is achieved for all $\beta \in \mathbb{R}^n$ if and only if the mixing depths $M_\mathcal{N}(S) = d$ for all $S \subseteq [n], |S| \geq 2$.*

*Proof.* By Lemma 39, it suffices to consider $\beta \in \mathbb{R}_+^n$. For each $i \in [n]$, let $P_i$ be a path from the $i$th input node to the output node. For each edge $e \in \bigcup_{i=1}^n P_i$, assign it the weights $\max\{\beta_i^{1/d} : e \in P_i\}$. For any edge not in $\bigcup_{i=1}^n P_i$, assign it weight 0. Let $w$ denote the resulting weights. Then

$$\|w\|_2^2 \leq \sum_{i=1}^n |P_i| \beta_i^{2/d} = d\|\beta\|_{2/d}^{2/d}, \tag{352}$$

where $|P_i| = d$ is the length of $P_i$ and

$$F_{\mathcal{N}}(w)[i] \geq (\beta_i^{1/d})^d = \beta_i. \tag{353}$$

Thus, by Lemma 39,

$$R_{\mathcal{N}}(\beta) \leq R_{\mathcal{N}}(F_{\mathcal{N}}(w)) \leq \|w\|_2^2 \leq d\|\beta\|_{2/d}^{2/d}. \tag{354}$$

Note that the condition $M_{\mathcal{N}}(S) = d$ for all $S \subseteq [n], |S| \geq 2$ is equivalent to $\mathcal{N}$ being essentially diagonal. Suppose that $\mathcal{N}$ is not essentially diagonal. Then we could choose for each $i \in [n]$, a path $P_i$ from the $i$th input node to the output node such that $P_1, \ldots, P_n$ are not edge disjoint. Then, Eq. (352) is strict inequality as long as $\operatorname{supp}(\beta) = [n]$. For such $\beta$,

$$R_{\mathcal{N}}(\beta) < d\|\beta\|_{2/d}^{2/d}. \tag{355}$$

Now, suppose that $\mathcal{N}$ is essentially diagonal. We partition the network $\mathcal{N}$ as follows. For each $i \in N_0$, let

$$V_i = \{v \in V : \text{ there exists a directed path from } i \text{ to } v\}. \tag{356}$$

Let

$$E_i = \{e \in E : \text{ both endpoints of } e \text{ lie in } V_i\}. \tag{357}$$

Let $\mathcal{N}_i$ be the architecture corresponding to the directed graph $(V_i, E_i)$. Since $\mathcal{N}$ is essentially diagonal,

$$E_i \cap E_j = \emptyset, \tag{358}$$

for all $i \neq j$. Thus,

$$R_{\mathcal{N}}(\beta) \overset{(s)}{=} \sum_{i=1}^n R_{\mathcal{N}_i}(\beta_i) \overset{(a)}{=} \sum_{i=1}^n R_{FC}(\beta_i) = \sum_{i=1}^n d|\beta_i|^{2/d} = d\|\beta\|_{2/d}^{2/d}, \tag{359}$$

where in $(s)$ we used the fact that the networks $\mathcal{N}_i$ are disjoint except at the output node and $\bigcup_{i \in N_0} E_i = E$, and in $(a)$ we used Corollary 47(on the second layer of $\mathcal{N}_i$) to reduce each $\mathcal{N}_i$ to a directed path, which is a fully connected network with one node in each layer.

$\square$

Next, we show that the representation cost function $R_{\mathcal{N}}(\beta)$ is continuous.

**Lemma 43.** *Let $\mathcal{N}$ be a homogeneous feedforward neural network without shared weights and of depth $d$. Then, the representation cost function $R_{\mathcal{N}}(\beta)$ is continuous.*

*Proof.* Let $\beta \in \mathbb{R}^n$. Let $C = d(\|\beta\|_{2/d}^{2/d} + n)$. Then, by Lemma 42,

$$R_{\mathcal{N}}(\beta') \leq d\|\beta'\|_{2/d}^{2/d} \leq d\sum_{i=1}^n (|\beta_i| + 1)^{2/d} \leq d\sum_{i=1}^n (|\beta_i|^{2/d} + 1) = C, \tag{360}$$

for all $\beta' \in B_1(\beta) := \{\beta' \in \mathbb{R}^n : \|\beta' - \beta\|_2 \leq 1\}$. Let $(\beta^{(t)})_{t \in \mathbb{N}}$ be a sequence in $\mathbb{R}^n$ converging to $\beta$. Without loss of generality, assume that $\beta^{(t)} \in B_1(\beta)$ for all $t \in \mathbb{N}$. By Lemma 41,

$$|R_{\mathcal{N}}(\beta^{(t)}) - R_{\mathcal{N}}(\beta)| \leq 2\sqrt{nC}\delta_t^{1/d} + |E|\delta_t^{2/d}, \tag{361}$$

where $\delta_t = \|\beta^{(t)} - \beta\|_2$. Taking $\limsup$ on both sides of Eq. (361), we get

$$\limsup_{t \to \infty} |R_{\mathcal{N}}(\beta^{(t)}) - R_{\mathcal{N}}(\beta)| \leq \limsup_{t \to \infty} 2\sqrt{nC}\delta_t^{1/d} + |E|\delta_t^{2/d} = 0, \tag{362}$$

since $\delta_t \to 0$ as $t \to \infty$. Since $|R_\mathcal{N}(\beta^{(t)}) - R_\mathcal{N}(\beta)| \geq 0$, Eq. (362) implies that

$$\lim_{t \to \infty} |R_\mathcal{N}(\beta^{(t)}) - R_\mathcal{N}(\beta)| = 0. \tag{363}$$

Thus,

$$\lim_{t \to \infty} R_\mathcal{N}(\beta^{(t)}) = R_\mathcal{N}(\beta). \tag{364}$$

Thus, $R_\mathcal{N}(\beta)$ is continuous. □

### E.3 Reference function

A common technique we will use to prove that some certain architecture $\mathcal{N}$ does not induce a certain quasi-norm $\|\cdot\|$ as induced complexity measure is as follows. We first find a function $f : \mathbb{R}^n \longrightarrow \mathbb{R}$ such that $f(\beta)$ depends only on $\|\beta\|$. Then, we find $\beta', \beta'' \in \mathbb{R}^n$ such that

$$\|\beta'\| = \|\beta''\|, \quad R_\mathcal{N}(\beta') = f(\beta'), \quad \text{and} \quad R_\mathcal{N}(\beta'') \neq f(\beta''). \tag{365}$$

Now, if $\mathcal{N}$ induces $\|\cdot\|$ as induced complexity measure, then $R_\mathcal{N}(\beta') = \psi(\|\beta'\|) = \psi(\|\beta''\|) = R_\mathcal{N}(\beta'')$. On the other hand, $R_\mathcal{N}(\beta') = f(\beta') = f(\beta'') \neq R_\mathcal{N}(\beta'')$ by Eq. (365) and the fact that $f(\beta)$ depends only on $\|\beta\|$. Thus, $\mathcal{N}$ does not induce $\|\cdot\|$ as induced complexity measure. We will call such a function $f$ a reference function, since we compare the representation cost $R_\mathcal{N}$ to it in order to get a contradiction.

### E.4 Subnetwork

Another common technique we will use is to consider a subnetwork $\mathcal{N}_S$ of $\mathcal{N}$ corresponding to a certain subset $S \subseteq [n]$ of the input nodes. We obtain $\mathcal{N}_S$ from $\mathcal{N}$ in two steps. First, we remove all input nodes in $\mathcal{N}$ except for those in $S \subseteq [n]$. Then, we remove all nodes that are isolated(cannot be reached by any input node in $S$ via a directed path). Now, we give the formal definition.

**Definition E.2 (Subnetwork).** For $S \subseteq [n]$, the restriction of $\mathcal{N}$ to $S$ is called $\mathcal{N}_S$. The subnetwork $\mathcal{N}_S$ is obtained from $\mathcal{N}$ by first removing all input nodes in $\mathcal{N}$ except for those in $S \subseteq [n]$ and then removing all hidden nodes that are isolated from the remaining input nodes.

Alternatively, $\mathcal{N}_S$ is the subnetwork of $\mathcal{N}$ corresponding to the subgraph generated by nodes $v \in V$ such that

$$S_v \cap S \neq \emptyset. \tag{366}$$

The induced complexity measure of the subnetwork is tightly related to that of the original network.

**Lemma 44.** *Let $\mathcal{N}$ be an architecture. Let $\mathcal{N}_S$ be the subnetwork of $\mathcal{N}$ with respect to the input nodes in $S \subseteq [n]$ (Def E.2). Then for any $\beta \in \mathbb{R}^n$ such that $\mathrm{supp}(\beta) \subseteq S$,*

$$R_{\mathcal{N}_S}(\beta_S) = R_\mathcal{N}(\beta), \tag{367}$$

*where $\beta_S$ is the projection of $\beta$ on coordinates in $S$. Furthermore, if $\mathcal{N}$ induces some quasi-norm $h(\cdot)$ as induced complexity measure, then $\mathcal{N}_S$ induces $h_S(\cdot)$ as induced complexity measure, where $h_S(\cdot) : \mathbb{R}^{|S|} \longrightarrow \mathbb{R}$ is defined as*

$$h_S(\beta') = h(\beta), \tag{368}$$

*where $\beta$ is the lifting of $\beta'$ defined as: $\beta_i = \beta'_i$ for $i \in S$ and $\beta_i = 0$ otherwise.*

*Proof.* Let $w'$ be the weights on $\mathcal{N}_S$ such that $F_{\mathcal{N}_S}(w') = \beta_S$ and $R_{\mathcal{N}_S}(\beta_S) = \|w'\|_2^2$. Then, we extend $w'$ to weights on $\mathcal{N}$ by putting $0$ weights on new edges. Let $w$ denote the resulting weights. Then,

$$F_\mathcal{N}(w) = \beta, \quad \text{and} \quad \|w\|_2^2 = \|w'\|_2^2 = R_{\mathcal{N}_S}(\beta_S). \tag{369}$$

Thus,

$$R_\mathcal{N}(\beta) \leq \|w\|_2^2 = R_{\mathcal{N}_S}(\beta_S). \tag{370}$$

For the other direction, let $\tilde{w}$ be the weights on $\mathcal{N}$ such that $F_\mathcal{N}(\tilde{w}) = \beta$ and $R_\mathcal{N}(\beta) = \|\tilde{w}\|_2^2$. Since $\mathrm{supp}(\beta) \subseteq S$, any edge in $\mathcal{N}$ that is not in $\mathcal{N}_S$ would not contribute to nonzero entries of $\beta$. Thus,

setting the weights of these edges to 0 would not affect $F_{\mathcal{N}}(\tilde{w})$. Since $R_{\mathcal{N}}(\beta) = \|\tilde{w}\|_2^2$, the weights on these edges must already be 0 in $\tilde{w}$. Thus,

$$\|\tilde{w}\|_2^2 = \|\tilde{w}_S\|_2^2, \quad \text{and} \quad F_{\mathcal{N}_S}(\tilde{w}_S) = \beta_S \tag{371}$$

where $\beta_S$ is the projection of $\beta$ on the coordinates in $S$ and $\tilde{w}_S$ is restriction of $\tilde{w}$ to the edges of $\mathcal{N}_S$. Thus,

$$R_{\mathcal{N}_S}(\beta_S) \leq \|\tilde{w}_S\|_2^2 = \|\tilde{w}\|_2^2 = R_{\mathcal{N}}(\beta). \tag{372}$$

Thus,

$$R_{\mathcal{N}_S}(\beta_S) = R_{\mathcal{N}}(\beta). \tag{373}$$

Since $\mathcal{N}$ induces $h(\cdot)$ as induced complexity measure, there exists a strictly increasing function $\psi : \mathbb{R} \longrightarrow \mathbb{R}$ such that

$$R_{\mathcal{N}}(\beta) = \psi(h(\beta)), \tag{374}$$

for all $\beta \in \mathbb{R}^n$ by Def 1.1. Thus,

$$R_{\mathcal{N}_S}(\beta_S) = R_{\mathcal{N}}(\beta) = \psi(h(\beta)) = \psi(h_S(\beta_S)), \tag{375}$$

since $\text{supp}(\beta) \subseteq S$. Finally, note that for any $\beta' \in \mathbb{R}^{|S|}$, $\beta' = \beta''_S$, where $\beta''_i = \beta'_i$ if $i \in S$ and $\beta''_i = 0$ otherwise. Thus,

$$R_{\mathcal{N}_S}(\beta') = \psi(h_S(\beta')), \tag{376}$$

for all $\beta' \in \mathbb{R}^{|S|}$. Thus, $\mathcal{N}_S$ induces $h_S(\cdot)$ as induced complexity measure. □

## E.5   Contraction of paths

Another technique we will use to modify architectures is to contract a path. Let $\mathcal{N}_P$ be an architecture which is a concatenation of some architecture $\mathcal{N}_{0:i}$ of depth $i$, followed by a fully connected layer consisting of one node at the $i + 1$th layer, and then followed by a path $P$ of depth $d - i - 1$. Equivalently, $|N_j| = 1$ for all $j > i$. Then, we claim that contracting $P$ to a single output node would not affect the induced complexity measure. We state this as a lemma.

**Lemma 45.** *Let $\mathcal{N}_P$ be an architecture of depth $d$ such that $|N_j| = 1$ for all $j > i$. Let $P = \mathcal{N}_{i+1:d}$ be the last $d - i$ layers of $\mathcal{N}_P$, which is a path. Let $\mathcal{N}_{0:i+1}$ be the first $i + 1$ layers of $\mathcal{N}_P$. Then, $\mathcal{N}_{0:i+1}$ and $\mathcal{N}_P$ induce the same induced complexity measure.*

Note that $\mathcal{N}_{0:i+1}$ is obtained from $\mathcal{N}_P$ by contracting the path $P$ to a single output node.

*Proof.* Let $t$ be the scaling factor induced by weights on $P$, that is

$$t = \prod_{e \in P} w(e), \tag{377}$$

where $w(e)$ denotes the weights on an edge $e$. Let

$$d_2 = d - i - 1, \tag{378}$$

be the length of $P$. By AM-GM inequality, it is clear that the weights of edges on $P$ would be the same in order to achieve minimum representation cost, that is

$$R_P(t) = d_2 |t|^{2/d_2}. \tag{379}$$

Let $\mathcal{N}_{0:i+1}$ be the architecture obtained from $\mathcal{N}_P$ by contracting $P$ to a single output node. Then, by Lemma 37,

$$R_{\mathcal{N}_{0:i+1}}(\beta/t) = \frac{R_{\mathcal{N}_{0:i+1}}(\beta)}{|t|^{2/(i+1)}}. \tag{380}$$

By equations (379) and (380),

$$R_{\mathcal{N}_P}(\beta) = \min_t(R_{\mathcal{N}_{0:i+1}}(\beta/t) + R_P(t)) = \min_t \left( \frac{R_{\mathcal{N}_{0:i+1}}(\beta)}{|t|^{2/(i+1)}} + d_2|t|^{2/d_2} \right) = d \left( \frac{R_{\mathcal{N}_{0:i+1}}(\beta)}{i+1} \right)^{(i+1)/d}, \tag{381}$$

where the last step follows from AM-GM inequality (write the first term into a sum of $i + 1$ identical terms and write the second term into a sum of $d_2$ identical terms, and then use AM-GM on the resulting $d$ terms). Since $R_{\mathcal{N}_P}(\beta)$ is monotonic in $R_{\mathcal{N}_{0:i+1}}(\beta)$, $\mathcal{N}_P$ and $\mathcal{N}_{0:i+1}$ induce the same induced complexity measure.

□

Since we only care about induced complexity measure of architectures, we can always contract a path $P$ when the conditions in Lemma 45 are met.

## E.6 Reduced architectures

Now, we discuss a notion called reduced architectures. Intuitively, if an architecture $\mathcal{N}$ is not reduced, then we can modify it to make it reduced without changing its induced complexity measure. To motivate the definition, we first make a simple observation. Note that for any $\beta \in \mathbb{R}^n$, and any architecture $\mathcal{N}$, the representation cost of $\beta$ in $\mathcal{N}$ is certainly lower bounded by that in a fully connected network since we can always choose to set the weights of some edges to zero. However, under what condition would that be attained? Now, we give a necessary and sufficient condition for that. Recall that a fully connected network of depth $d$ has representation cost $d\|\beta\|_2^{2/d}$ (See [15]), for all $\beta \in \mathbb{R}^n$.

**Lemma 46.** *For any $\beta \in \mathbb{R}^n$,*

$$R_{\mathcal{N}}(\beta) \geq d\|\beta\|_2^{2/d}. \tag{382}$$

*Moreover, for any $S \subseteq [n]$, $S \neq \emptyset$, the following statements are equivalent:*

*(1) There exists $\beta \in \mathbb{R}^n$ with $\mathrm{supp}(\beta) = S$, such that*

$$R_{\mathcal{N}}(\beta) = d\|\beta\|_2^{2/d}. \tag{383}$$

*(2) There exists a node $v$ in the first hidden layer $N_1$, that is connected to all nodes in the input layer that correspond to the support of $\beta$, or equivalently,*

$$S_v \supset \mathrm{supp}(\beta) = S. \tag{384}$$

*for some $v \in N_1$.*

*(3) For all $\beta \in \mathbb{R}^n$ with $\mathrm{supp}(\beta) = S$, we have*

$$R_{\mathcal{N}}(\beta) = d\|\beta\|_2^{2/d}. \tag{385}$$

*In particular, this implies that $R_{\mathcal{N}}(\beta) = d\|\beta\|_2^{2/d}$, for all $\beta \in \mathbb{R}^n$ if and only if there exists a node $v$ in the first hidden layer $N_1$, that is connected to all nodes in the input layer, or equivalently,*

$$S_v = N_0. \tag{386}$$

*for some $v \in N_1$.*

*Proof.* We will first prove Eq. (382). Then we will prove $(1) \Rightarrow (2) \Rightarrow (3) \Rightarrow (1)$. The last one is trivial. For the first one, we will first expand the trivial inequality $R_{\mathcal{N}}(\beta) \geq d\|\beta\|_2^{2/d}$ into a chain of inequalities. Then, if equality is attained for some $\beta \in \mathbb{R}^n$, then all the inequalities in the chain have to hold with equality. We will use one of these inequalities to prove $(2)$, which is essentially an application of Cauchy Schwartz inequality. To show $(2) \Rightarrow (3)$, we will see that since we are interested in the representation cost of $\beta$, we can delete all input nodes except for those corresponding to the support of $\beta$. Then, in the resulting architecture, we show that we can extract a fully connected subnetwork. Now, since representation cost of a subnetwork is lower bounded by representation cost of the original one, we know that the representation cost of $\beta$ in the original network is upper bounded by that of a fully connected network. On the other hand, the representation cost of $\beta$ in the original network is always bounded below by that of a fully connected network since we can always add edges to the original network to make it fully connected. Now, the new network has a representation cost upper bounded by the original one since it contains the original one as a subnetwork. Note that we are essentially using the fact the representation cost of a fully connected network does not depend on its width. Now, we give the rigorous proof.

**Equation (382) :**   Let $w = (W_1, W_2, \ldots, W_d)$ be such that

$$F_{\mathcal{N}}(w) = \prod_{i=1}^{d} W_{d-i+1} = \beta. \tag{387}$$

$$\|w\|_2^2 = \sum_{i=1}^{d} \|W_i\|_F^2$$

$$\overset{(a)}{\geq} d\left(\prod_{i=1}^{d} \|W_i\|_F\right)^{2/d}$$

$$\overset{(b)}{\geq} d\left(\left\|\prod_{i=1}^{d} W_{d-i+1}\right\|_F\right)^{2/d}$$

$$= d\|\beta\|_2^{2/d}, \tag{388}$$

where we used AM-GM inequality in $(a)$ and submultiplicity of Frobenius norm in $(b)$. Thus,

$$R_{\mathcal{N}}(\beta) \geq d\|\beta\|_2^{2/d}. \tag{389}$$

**(1) $\Rightarrow$ (2):** Let $S \subseteq [n]$, $S \neq \emptyset$. Suppose that

$$R_{\mathcal{N}}(\beta) = d\|\beta\|_2^{2/d}. \tag{390}$$

for some $\beta \in \mathbb{R}^n$ with

$$\text{supp}(\beta) = S. \tag{391}$$

Then there exists some $w = (W_1, W_2, \ldots, W_d)$ such that

$$F_{\mathcal{N}}(w) = \prod_{i=1}^{d} W_{d-i+1} = \beta, \qquad \|w\|_2^2 = d\|\beta\|_2^{2/d}. \tag{392}$$

By $(b)$, this means that

$$d\left(\prod_{i=1}^{d} \|W_i\|_F\right)^{2/d} = d\left(\left\|\prod_{i=1}^{d} W_{d-i+1}\right\|_F\right)^{2/d}. \tag{393}$$

In particular, this implies

$$\left\|\prod_{i=1}^{d-1} W_{d-i+1}\right\|_F^2 \|W_1\|_F^2 = \left\|\left(\prod_{i=1}^{d-1} W_{d-i+1}\right) W_1\right\|_F^2, \tag{394}$$

since

$$\prod_{i=1}^{d} \|W_i\|_F \geq \left\|\prod_{i=1}^{d-1} W_{d-i+1}\right\|_F \|W_1\|_F \geq \left\|\prod_{i=1}^{d} W_{d-i+1}\right\|_F. \tag{395}$$

Let $A = W_1 \in \mathbb{R}^{m \times n}$, $c = (\prod_{i=1}^{d-1} W_{d-i+1})^T \in \mathbb{R}^m$. Let $a_{i,j} = A[i,j]$, $w_i = c[i]$. Then

$$\left\|c^T A\right\|_F^2 = \|c\|_2^2 \|A\|_F^2. \tag{396}$$

Note that

$$\left\|c^T A\right\|_F^2 = \sum_{j=1}^{n}\left(\sum_{i=1}^{m} w_i a_{i,j}\right)^2$$

$$\overset{(e)}{\leq} \sum_{j=1}^{n}\left(\left(\sum_{i=1}^{m} w_i^2\right)\left(\sum_{k=1}^{m} a_{k,j}^2\right)\right)$$

$$= \left(\sum_{i=1}^{m} w_i^2\right)\left(\sum_{j=1}^{n}\sum_{k=1}^{m} a_{k,j}^2\right)$$

$$= \|c\|_2^2 \|A\|_F^2, \tag{397}$$

where we used Cauchy's inequality in $(e)$. Since $c^T A = \beta \neq 0$ by Eq. (392), $c \neq 0$. Thus, there exists $i^* \in [m]$ such that $w_{i^*} \neq 0$. We will show that the $i^*$th node in the first hidden layer $N_1$ is connected to all nodes in the input layer that correspond to the support of $\beta$. Now, for $(e)$ to hold with equality, for each $j \in [n]$, there exists $\lambda_j$ such that

$$a_{i,j} = \lambda_j w_i \tag{398}$$

for all $i \in [m]$. Since $\sum_{i=1}^m w_i a_{i,j} = \beta_j \neq 0$ for all $j \in S = \mathrm{supp}(\beta)$,

$$\lambda_j \neq 0 \tag{399}$$

for all $j \in S$. Then

$$a_{i^*,j} = \lambda_j w_{i^*} \neq 0 \tag{400}$$

for all $j \in S$. Let $v$ be the $i^*$th node in $N_1$. Then $v$ is connected to all nodes in the input layer that correspond to the support of $\beta$.

**(2) $\Rightarrow$ (3) :**  Suppose that there is a node $v \in N_1$ that is connected to all nodes in the input layer that correspond to $S$. Let $\beta \in \mathbb{R}^n$ be a vector with

$$\mathrm{supp}(\beta) = S. \tag{401}$$

Let $\mathcal{N}_S$ be the subnetwork of $\mathcal{N}$ with respect to the input nodes in $S$ (Def E.2). Then, by Lemma 44,

$$R_{\mathcal{N}}(\beta) = R_{\mathcal{N}_S}(\beta_S), \tag{402}$$

where $\beta_S$ is the projection of $\beta$ on coordinates in $S$. Since $v$ is connected to all input nodes of $\mathcal{N}_S$, we can extract a fully connected subnetwork $\mathcal{N}_{FC}$ from $\mathcal{N}_S$ as follows. Let $P$ be a path from $v$ to the output node. Let $\mathcal{N}_{FC}$ be the subnetwork obtained from $\mathcal{N}_S$ by keeping only the path $P$ and the edges between the input nodes and $v$. Then, for any $\beta \in \mathbb{R}^n$,

$$R_{\mathcal{N}}(\beta) = R_{\mathcal{N}_S}(\beta_S) \leq R_{\mathcal{N}_{FC}}(\beta_S) = d\|\beta_S\|_2^{2/d} = d\|\beta\|_2^{2/d}, \tag{403}$$

where $\mathcal{N}_{FC}$ is fully connected. Thus,

$$R_{\mathcal{N}}(\beta) = d\|\beta\|_2^{2/d}. \tag{404}$$

$\square$

By the proof of the previous lemma, we get the following corollary.

**Corollary 47.** *Suppose there exists a node $v \in N_{i+1}$ that is connected to all $u \in N_i$. Let $P$ be a directed path from $v$ to the output node $O$. Then removing all the nodes in $N_{i+1}$ except for $v$ and removing all edges after the $i+1$th layer except for those on $P$ would not change the representation cost. Furthermore, contracting $P$ to a single output node would not change the induced complexity measure.*

*Proof.* Let $d$ be the depth of $\mathcal{N}$. We view $\mathcal{N}$ as a concatenation of two architectures $\mathcal{N}_{0:i}$ and $\mathcal{N}_{i:d}$, where the first one corresponds to the first $i$ layer and second one corresponds to the last $d - i + 1$ layers. Let $\mathcal{N}'_{i:d}$ be the subnetwork obtained from $\mathcal{N}_{i:d}$ by removing all the nodes in $N_{i+1}$ except for $v$ and removing all edges after the $i+1$th layer except for those on P. Then, $\mathcal{N}'_{i:d}$ is fully connected. Thus, by Lemma 46,

$$(d-i)\|\beta\|_2^{2/(d-i)} \leq R_{\mathcal{N}_{i:d}}(\beta) \leq R_{\mathcal{N}'_{i:d}}(\beta) = (d-i)\|\beta\|_2^{2/(d-i)}, \tag{405}$$

where the first term is the representation cost of a fully connected network of depth $d - i$, and the second inequality follows from the fact that $\mathcal{N}'_{i:d}$ is a subnetwork of $\mathcal{N}_{i:d}$. Thus,

$$R_{\mathcal{N}_{i:d}}(\beta) = R_{\mathcal{N}'_{i:d}}(\beta). \tag{406}$$

Let $\mathcal{N}'$ be the concatenation of $\mathcal{N}_{0:i}$ and $\mathcal{N}'_{i:d}$. Then for any $\beta \in \mathbb{R}^n$,

$$\begin{aligned}
R_{\mathcal{N}}(\beta) &= \min_{A,c}\{R_{\mathcal{N}_{0:i}}(A) + R_{\mathcal{N}_{i:d}}(c) : c^T A = \beta\} \\
&= \min_{A,c}\{R_{\mathcal{N}_{0:i}}(A) + R_{\mathcal{N}'_{i:d}}(c) : c^T A = \beta\} \\
&= R_{\mathcal{N}'}(\beta).
\end{aligned} \tag{407}$$

Note that in $\mathcal{N}'$, the architecture of the last $d - i$ layers is a path, which we denote by $P$. By Lemma 45, contracting $P$ to a single output node does not affect the induced complexity measure.

$\square$

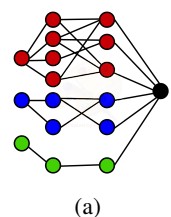
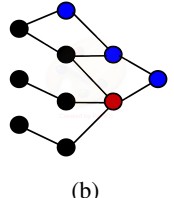

| (a) | (b) |

Figure 3: **Two architectures.** Figure 3a is an essentially diagonal network. Figure 3b is not fully reduced. Figure 1a is a $k$-balanced network, which induces k-support norm with $n = 3, k = 2$.

Now, we are ready to give the definition of reduced architectures.

**Definition E.3** (Reduced Architecture). An architecture $\mathcal{N}$ is reduced if for all $i < d - 1$, there does not exist $v \in N_{i+1}$ that is connected to all $u \in N_i$.

If an architecture is not reduced, then we can always do some modifications as in Corollary 47 without changing its induced complexity measure. Thus, we can always assume that an architecture is reduced.

Now, we prove Theorem 36.

**Theorem 36.** *For all linear homogeneous feedforward neural networks $\mathcal{N}$ without shared weights and of depth $d$,*

$$R_{FC}(\beta) = d\|\beta\|_2^{2/d} \leq R_{\mathcal{N}}(\beta) \leq d\|\beta\|_{2/d}^{2/d} = R_{DNN}(\beta). \tag{309}$$

*Furthermore, the lower bound is achieved for all $\beta \in \mathbb{R}^n$ if and only if the mixing depths $M_{\mathcal{N}}(S) = 1$ for all $S \subseteq [n]$, and the upper bound is achieved for all $\beta \in \mathbb{R}^n$ if and only if the mixing depths $M_{\mathcal{N}}(S) = d$ for all $S \subseteq [n]$ such that $|S| \geq 2$.*

*Proof.* The proof follows from Lemma 42, Lemma 46, Theorem 1, and Theorem 2. □

# F Supplementary materials in Section 4.1.1 : $l_p$ quasi-norms

We will use the following results in [15] many times:

$$R_{FC}(\beta) = d\|\beta\|_2^{2/d}, \qquad R_{DNN}(\beta) = d\|\beta\|_{2/d}^{2/d}, \tag{408}$$

where $FC$ and $DNN$ are fully connected network and diagonal network of depth $d$, respectively. In this section, we only consider networks without shared weights.

## F.1 Essentially diagonal networks

We begin with the definition of essentially diagonal layer and essentially diagonal network.

**Definition F.1** (Essentially Diagonal). A layer $N_i$ is essentially diagonal if for all $v \in N_i$,

$$|S_v| = 1, \tag{409}$$

where $S_v$ is defined as

$$S_v := \{i \in N_0 : \exists \text{ a directed path from } i \text{ to } v\}. \tag{410}$$

An architecture $\mathcal{N}$ of depth $d$ is essentially diagonal if it consists of $d - 1$ essentially diagonal layers followed by a fully connected layer.

Note that if $\mathcal{N}$ is essentially diagonal, then $|S_v| = 1$, for all $v \in \bigcup_{i=1}^{d-1} N_i$. In other words, an essentially diagonal network is the combination of $n$ separated subnetworks that are connected to the same output node in the last layer. See Figure 3a for an example.

Thus, $\mathcal{N}$ is essentially diagonal if and only if the mixing depths $M_{\mathcal{N}}(S) = d$ for all $S \subseteq [n], |S| \geq 2$. This means that essentially diagonal networks achieve the upper bound in Theorem 36. Thus, we immediately get its representation cost.

**Lemma 48.** *If $\mathcal{N}$ is essentially diagonal, then*

$$R_{\mathcal{N}}(\beta) = R_{DNN}(\beta) = d\|\beta\|_{2/d}^{2/d}. \tag{411}$$

## F.2 Fully reduced architectures

Now, we make some observations similar to Lemma 46 and Corollary 47. If we have an architecture whose first $d_1$ layers are essentially diagonal, then its representation cost is certainly lower bounded by that of the architecture which consists of $d_1$ essentially diagonal layers followed by $d - d_1$ fully connected layers. The representation cost of the latter one can be shown to be $d\|\beta\|_{2/(d_1+1)}^{2/d}$. Indeed, by Corollary 47, we can show that the second architecture has the same representation cost as a diagonal network of depth $d_1 + 1$ followed by a path of depth $d - d_1 - 1$. However, when would this lower bound be attained? We give a necessary and sufficient condition in the following lemma.

**Lemma 49.** *Let $\mathcal{N}$ be a depth $d$ linear neural network such that for some $d_1 < d$,*

$$|S_u| = 1 \tag{412}$$

*for all $u \in N_{d_1}$. Then for any $\beta \in \mathbb{R}^n$,*

$$R_{\mathcal{N}}(\beta) \geq d\|\beta\|_{2/(d_1+1)}^{2/d}. \tag{413}$$

*Moreover, the following statements are equivalent:*

*(1) There exists $\beta \in \mathbb{R}^n$ such that $\beta_i \neq 0$ for all $i \in [n]$, and*

$$R_{\mathcal{N}}(\beta) = d\|\beta\|_{2/(d_1+1)}^{2/d}. \tag{414}$$

*(2) There exists a node $v \in N_{d_1+1}$ such that*

$$S_v = N_0. \tag{415}$$

*(3) For all $\beta \in \mathbb{R}^n$,*

$$R_{\mathcal{N}}(\beta) = d\|\beta\|_{2/(d_1+1)}^{2/d}. \tag{416}$$

*Proof.* We will first prove Eq. (413). Then, we will prove $(1) \Rightarrow (2) \Rightarrow (3) \Rightarrow (1)$. The proof strategy is exactly the same as in Lemma 46.

**Eq. (413):** Let $\beta \in \mathbb{R}^n$. For each $i \in [n]$, let

$$W_i = \{v \in N_{d_1} : S_v = \{i\}\}, \qquad w_i = |W_i|. \tag{417}$$

Without loss of generality, assume that the nodes in $N_{d_1}$ are ordered in a way so that the first $w_1$ nodes are in $W_1$, the next $w_2$ nodes are in $W_2$ and so on. Let $w = (w_{1:d_1}, w_{d_1+1:d})$ be the weights of $\mathcal{N}$, where $w_{1:d_1}$ denotes the weights in the first $d_1$ layer and $w_{d_1+1:d}$ denotes the weights in the remaining layers, such that

$$F_{\mathcal{N}}(w) = \beta. \tag{418}$$

We view $\mathcal{N}$ as a concatenation of $\mathcal{N}_{0:d_1}$ and $\mathcal{N}_{d_1:d}$, where $\mathcal{N}_{0:d_1}$ corresponds to the first $d_1$ layers and $\mathcal{N}_{d_1:d}$ corresponds to the remaining $d_2 + 1 := d - d_1 + 1$ layers. Let

$$T = F_{\mathcal{N}_{0:d_1}}(w_{1:d_1}), \qquad a^T = F_{\mathcal{N}_{d_1:d}}(w_{d_1+1:d}). \tag{419}$$

By assumption,

$$T = \begin{bmatrix} t_1 & 0 & \dots & 0 \\ 0 & t_2 & \dots & 0 \\ \vdots & \vdots & \dots & \vdots \\ 0 & 0 & \dots & t_n \end{bmatrix}, \tag{420}$$

where $t_i \in \mathbb{R}^{w_i}$, for all $i \in [n]$. Let

$$a^T = (a_1^T, a_2^T, \dots, a_n^T), \tag{421}$$

where $a_i \in \mathbb{R}^{w_i}$, for all $i \in [n]$. Then we have

$$\beta = a^T T = (a_1^T t_1, a_2^T t_2, \dots, a_n^T t_n). \tag{422}$$

Thus,

$$\beta_i = a_i^T t_i \tag{423}$$

for all $i \in [n]$. Let $(V', E')$ be the directed graph associated with $\mathcal{N}_{0:d_1}$. We partition the network $\mathcal{N}_{0:d_1}$ as follows. For each $i \in N_0$, let

$$V'_i = \{v \in V' : \text{ there exists a directed path from } i \text{ to } v\}. \tag{424}$$

Let

$$E'_i = \{e \in E' : \text{ both endpoints of } e \text{ lie in } V'_i\}. \tag{425}$$

Let $\mathcal{N}_i$ be the architecture corresponding to the directed graph $(V'_i, E'_i)$. Since $|S_v| = 1$ for all $v \in N_{d_1}$,

$$E'_i \cap E'_j = \emptyset, \tag{426}$$

for all $i \neq j$. Thus,

$$R_{\mathcal{N}_{0:d_1}}(T) = \sum_{i=1}^{n} R_{\mathcal{N}_i}(t_i) \overset{(e)}{\geq} \sum_{i=1}^{n} d_1 \|t_i\|_2^{2/d_1}, \tag{427}$$

where in $(e)$ we used the fact that the representation cost of a vector $t_i$ (here the linear transformation is $\mathbb{R} \to \mathbb{R}^{w_i}$) in a fully connected network of depth $d_1$ is $d_1 \|t_i\|_2^{2/d_1}$ (See Theorem 1), which is certainly a lower bound for any other architecture of the same depth. Now, note that

$$
\begin{aligned}
\|w\|_2^2 &= \|w_{1:d_1}\|_2^2 + \|w_{d_1+1:d}\|_2^2 \\
&\geq R_{\mathcal{N}_{0:d_1}}(T) + R_{\mathcal{N}_{d_1:d}}(a) \\
&\overset{(f)}{\geq} \sum_{i=1}^{n} d_1 \|t_i\|_2^{2/d_1} + d_2 \|a\|_2^{2/d_2} \\
&\overset{(g)}{\geq} d\left[\left(\sum_{i=1}^{n} \|t_i\|_2^{2/d_1}\right)^{d_1} \left(\|a\|_2^{2/d_2}\right)^{d_2}\right]^{1/d} \\
&= d\left[\left(\sum_{i=1}^{n} \|t_i\|_2^{2/d_1}\right)^{d_1} \left(\sum_{i=1}^{n} \|a_i\|_2^2\right)\right]^{1/d} \\
&= d\left[\left(\sum_{i=1}^{n} (\|t_i\|_2^{2/(d_1+1)})^{(d_1+1)/d_1}\right)^{d_1/(d_1+1)} \left(\sum_{i=1}^{n} (\|a_i\|_2^{2/(d_1+1)})^{d_1+1}\right)^{1/(d_1+1)}\right]^{(d_1+1)/d} \\
&\overset{(h)}{\geq} d\left[\sum_{i=1}^{n} (\|t_i\|_2^{2/(d_1+1)} \|a_i\|_2^{2/(d_1+1)})\right]^{(d_1+1)/d} \\
&\overset{(r)}{\geq} d\left[\sum_{i=1}^{n} (|a_i^T t_i|)^{2/(d_1+1)}\right]^{(d_1+1)/d} \\
&\overset{(s)}{=} d\left[\sum_{i=1}^{n} |\beta_i|^{2/(d_1+1)}\right]^{(d_1+1)/d} \\
&= d\|\beta\|_{2/(d_1+1)}^{2/d}.
\end{aligned}
\tag{428}
$$

where in $(f)$ we used Eq. (427) and Lemma 46, in $(g)$ we used AM-GM inequality, in $(h)$ we used Holder's inequality, in $(r)$ we used Cauchy's inequality, and in $(s)$ we used Eq. (423). This proves Eq. (413).

**(1) $\Rightarrow$ (2):** Suppose that there exists $\beta \in \mathbb{R}^n$, such that $\beta_i \neq 0$ for all $i \in [n]$, and

$$R_{\mathcal{N}}(\beta) = d\|\beta\|_{2/(d_1+1)}^{2/d}. \tag{429}$$

Then there exists $w$ such that

$$\|w\|_2^2 = d\|\beta\|_{2/(d_1+1)}^{2/d} \quad \text{and} \quad F_{\mathcal{N}}(w) = \beta. \tag{430}$$

By step $(f)$ in equation (428), this implies that

$$R_{\mathcal{N}_{d_1:d}}(a) = d_2 \|a\|_2^{2/d_2}. \tag{431}$$

Now, we will focus on the subnetwork $\mathcal{N}_{d_1:d}$. By Eq. (431) and Lemma 46, there is a node $v$ in $N_{d_1+1}$(which is the first hidden layer in $\mathcal{N}_{d_1:d}$) that is connected to all nodes in $N_{d_1}$ that correspond to the support of $a$. Thus,

$$S_v \supset \bigcup_{i \in \text{supp}(a)} S_{u_i}, \tag{432}$$

where $u_i$ is the $i$th node in $N_{d_1}$. Since $\beta_i \neq 0$ for all $i \in [n]$,

$$a_i^T t_i \neq 0 \tag{433}$$

for all $i \in [n]$ by equation (423). Thus, for each $i \in [n]$, there exists $s_i$ such that

$$a_i[s_i] \neq 0. \tag{434}$$

For each $i \in [n]$, let

$$j_i = s_i + \sum_{l=1}^{i-1} w_i. \tag{435}$$

Then

$$a[j_i] \neq 0, \quad \text{and} \quad S_{u_{j_i}} = \{i\}, \tag{436}$$

for all $i \in [n]$. Then, by Eq. (432),

$$S_v \supset \bigcup_{i \in [n]} S_{u_{j_i}} = [n] = N_0. \tag{437}$$

Thus,

$$S_v = N_0. \tag{438}$$

**(2) $\Rightarrow$ (3) :** Suppose that there exists $v \in N_{d_1+1}$ such that

$$S_v = N_0. \tag{439}$$

Now, for each $i \in [n]$, let $P_i$ be a directed path from $i$ to $v$. Let $P$ be a directed path from $v$ to $O$. Let $\tilde{\mathcal{N}}$ be the subnetwork of $\mathcal{N}$ corresponding to the subgraph $\bigcup_{i=1}^n P_i \cup P$. Now, $\tilde{\mathcal{N}}$ is a diagonal network concatenated with a path. Then for any $\beta \in \mathbb{R}^n$,

$$R_{\mathcal{N}}(\beta) \leq R_{\tilde{\mathcal{N}}}(\beta) = \min_\lambda ((d_1+1)\|\beta/\lambda\|_{2/(d_1+1)}^{2/(d_1+1)} + (d_2-1)\lambda^{2/(d_2-1)}) \overset{(a)}{=} d\|\beta\|_{2/(d_1+1)}^{2/d}, \tag{440}$$

where in $(a)$ we used AM-GM inequality. Thus,

$$R_{\mathcal{N}}(\beta) = d\|\beta\|_{2/(d_1+1)}^{2/d}. \tag{441}$$

$\square$

By the proof of the above lemma, we get the following corollary.

**Corollary 50.** *Let $\mathcal{N}$ be a neural network such that there exists $d_1 < d-1$ such that for all $u \in N_{d_1}$,*

$$|S_u| = 1 \tag{442}$$

*and there exists $v \in N_{d_1+1}$ such that*

$$S_v = N_0. \tag{443}$$

*For each $i \in [n]$, let $P_i$ be a directed path from $i$ to $v$. Then removing all edges from $\mathcal{N}$ except for those on $\bigcup_{i=1}^n P_i$ would not change the induced complexity measure.*

*Proof.* This immediately follows from the last part ($(2) \Rightarrow (3)$) of the proof of Lemma 49 and Lemma 45. $\square$

Now, we give the definition of a fully reduced architecture.

**Definition F.2** (Fully Reduced Architecture). An architecture $\mathcal{N}$ of depth $d$ is fully reduced when the following holds:

For all $0 \leq d_1 < d - 1$, if for all $u \in N_{d_1}$,

$$|S_u| = 1, \tag{444}$$

then for any $v \in N_{d_1+1}$,

$$S_v \neq N_0 = [n]. \tag{445}$$

Note that if an architecture is not fully reduced, then we can do some modifications to make it fully reduced without changing its induced complexity measure by Corollary 50. For instance, network in Figure 3b is not fully reduced because of the red node (here $d = 3$ and $d_1 = 1$). For any $u \in N_1$, $|S_u| = 1$ but for the red node $v \in N_2$, we have $S_v = N_0$. Thus, this architecture is not fully reduced. We can modify it by removing the three blue nodes.

## F.3 Essentially diagonal networks and $l_p$ quasi-norms

Now, we will show that $l_p$ quasi-norms correspond to essentially diagonal networks in the sense that an architecture $\mathcal{N}$ induces $l_p$ quasi-norm as induced complexity measure if and only if $\mathcal{N}$ is essentially diagonal, provided that $\mathcal{N}$ is fully reduced. Before giving the main result, we first make some observations. Suppose that $\mathcal{N}$ induces $l_p$ quasi-norm as induced complexity measure. Then the subnetwork $\mathcal{N}_{i,j}$ (Def E.2) of $\mathcal{N}$ with respect the $i,j$th input nodes also induces $l_p$ quasi-norm as induced complexity measure for the same value of $p$ by Lemma 44. Thus, if we can show that any architecture $\mathcal{N}$ with two input nodes could induce $l_p$ quasi-norm only if $p = 2/d'$ for some $d' \in \mathbb{N}$, then the same statement follows for any architecture. Indeed, as we shall see, all fully reduced architecture $\mathcal{N}$ with two input nodes are essentially diagonal.

**Lemma 51.** *Let $\mathcal{N}$ be a fully reduced architecture with two input nodes. Then, $\mathcal{N}$ is essentially diagonal. Consequently, any architecture with two input nodes induces $l_p$ quasi-norm as induced complexity measure for some $p \in \{2/d' : d' \in \mathbb{N}\}$.*

*Proof.* Let $\mathcal{N}$ be a fully reduced architecture with two input nodes. Let

$$d' = \min\{t : \exists v \in N_t \text{ such that } S_v = N_0 = \{1, 2\}\}, \tag{446}$$

where

$$S_v = \{i \in N_0 : \exists \text{ a directed path from } i \text{ to } v\}. \tag{447}$$

Since $\mathcal{N}$ is fully reduced (Def F.2), $\mathcal{N}$ is essentially diagonal of depth $d'$. By Lemma 48, $\mathcal{N}$ induces $l_{2/d'}$ quasi-norm as induced complexity measure. The second claim follows from the fact that we can always modify an architecture to make it fully reduced without changing its induced complexity measure by Corollary 50.

$\square$

Now, we give the main theorem.

**Theorem 52.** *Suppose that $\mathcal{N}$ induces $l_p$ quasi-norm for some $p$. Then $2/p \in \mathbb{N}$. Moreover, if $\mathcal{N}$ is also fully reduced, then it is essentially diagonal of depth $2/p$.*

The first part of the proof follows directly from the discussion at the beginning of this section. The second part of the proof will use $d' \|\beta\|_{2/d'}^{2/d'}$ as a reference function (Section E.3), where $d' = 2/p$.

*Proof.* Let $\mathcal{N}$ be an architecture which induces $l_p$ quasi-norm as induced complexity measure. Then the subnetwork $\mathcal{N}_{i,j}$ (Def E.2) of $\mathcal{N}$ with respect the $i,j$th input nodes also induces $l_p$ quasi-norm as induced complexity measure for the same value of $p$ by Lemma 44. By Lemma 51, $p = 2/d'$ for some $d' \in \mathbb{N}$.

In addition, suppose $\mathcal{N}$ is also fully reduced. For each $i, j \in [n]$, let

$$d_{i,j} = \min\{t : \exists v \in N_t \text{ such that } S_v \supset \{i, j\}\}, \tag{448}$$

where

$$S_v = \{i \in N_0 : \exists \text{ a directed path from } i \text{ to } v\}. \tag{449}$$

Since $\mathcal{N}_{i,j}$ induces $l_{2/d'}$ quasi-norm as induced complexity measure,

$$d_{i,j} = d' \tag{450}$$

for all $i, j \in [n]$, by Lemma 49. Thus, $|S_v| = 1$ for all $v \in \bigcup_{t=1}^{d'-1} N_t$. Then, we claim that there is $u \in N_{d'}$ such that $S_u = [n] = N_0$. Suppose for the sake of contradiction that

$$S_u \neq N_0 \tag{451}$$

for all $u \in N_{d'}$. Let $\beta = (1, 1, \ldots, 1)$. By Lemma 49 $((1) \Rightarrow (2))$,

$$R_\mathcal{N}(\beta) > d\|\beta\|_{2/d'}^{2/d}, \tag{452}$$

where $d$ is the depth of $\mathcal{N}$. Let $\beta' = \|\beta\|_{2/d'}(1, 0, \ldots, 0)$. Then, by Lemma 46,

$$R_\mathcal{N}(\beta') = d\|\beta'\|_2^{2/d} = d\|\beta\|_{2/d'}^{2/d} < R_\mathcal{N}(\beta). \tag{453}$$

However,

$$\|\beta\|_{2/d'} = \|\beta'\|_{2/d'}. \tag{454}$$

Thus, $\mathcal{N}$ cannot induce $l_{2/d'}$ quasi-norm as induced complexity measure since there are two vector with the same $l_{2/d'}$ quasi-norm but different representation cost with respect to $\mathcal{N}$(See Reference Function in section E.3). Thus, there exists $u \in N_{d'}$ such that $S_u = [n] = N_0$. Since $\mathcal{N}$ is fully reduced (Def F.2), $\mathcal{N}$ is essentially diagonal of depth $d'$.

$\square$

Now, we immediately get the following corollary about $l_{p,q}$ group quasi-norms that can be induced as induced complexity measure by neural networks.

**Corollary 53.** *If $\mathcal{N}$ induces $l_{p,q}$ group quasi-norm for some $p, q$, then $2/p, 2/q \in \mathbb{N}$.*

*Proof.* Suppose that $\mathcal{N}$ induces the following $l_{p,q}$ group quasi-norm as induced complexity measure

$$\|\beta\|_{p,q} = (\sum_{j=1}^k (\sum_{i \in G_j} |\beta_i|^q)^{p/q})^{1/p}, \tag{455}$$

where $\beta \in \mathbb{R}^n$, $k$ denotes the number of groups, and $G_j$ denote the $j$th group. Note that by definition of group quasi-norms, the $G_j$s are disjoint. Now, let $\mathcal{N}_{G_1}$ be the subnetwork (Def E.2) of $\mathcal{N}$ with respect to the input nodes in $G_1$. Then $\mathcal{N}_{G_1}$ induces $l_q$ quasi-norms as induced complexity measure. To see this, substitute $\beta_i = 0$ in equation (455) for all $i$ except for $i \in G_1$. Thus, by Theorem 35,

$$2/q \in \mathbb{N}. \tag{456}$$

Now, for each $j \in [k]$, pick $i_j \in G_j$. Let

$$B = \{i_j : j \in [k]\}. \tag{457}$$

Let $\mathcal{N}_B$ be the subnetwork (Def E.2) of $\mathcal{N}$ with respect to the input nodes in $B$. Then $\mathcal{N}_B$ induces $l_p$ quasi-norms as induced complexity measure. Thus, by Theorem 35,

$$2/p \in \mathbb{N}. \tag{458}$$

$\square$

## F.4 Proof of main results of $l_p$ quasi-norms

Now, we give a proof of Theorem 35.

**Theorem 35.** *A linear homogeneous feedforward neural network $\mathcal{N}$ without shared weights induces $l_p$ quasi-norm if and only if $M_\mathcal{N}(S) = 2/p$, for all $S \subseteq [n], |S| \geq 2$.*

*Proof.* Suppose $\mathcal{N}$ induces $l_p$ quasi-norm. Note that the operations in Corollary 50 do not change mixing depths. Thus, if $\mathcal{N}$ is not fully reduced, we can make it fully reduced using operations in Corollary 50 without changing its mixing depths and induced complexity measures. Without loss of generality, assume that $\mathcal{N}$ is fully reduced. By Theorem 52, it is essentially diagonal of depth $2/p$. Thus, $M_{\mathcal{N}}(S) = d = 2/p$, for all $S \subseteq [n], |S| \geq 2$.

On the other hand, suppose that $M_{\mathcal{N}}(S) = 2/p$, for all $S \subseteq [n], |S| \geq 2$. This implies that $|S_u| = 1$ for all $u \in N_{2/p-1}$ and there exists $v \in N_{2/p}$ such that $S_v = N_0 = [n]$. By Lemma 49, $\mathcal{N}$ induces $l_p$ quasi-norm.

$\square$

Now, we prove Theorem 7.

**Theorem 7.** *There exists a linear homogeneous feedforward neural network $\mathcal{N}$ without shared weights that induces $l_p$ quasi-norm if and only if $2/p \in \mathbb{N}$. In particular, diagonal network of depth $2/p$ induces $l_p$ quasi-norm.*

*Proof.* This immediately follows from Theorem 35, since mixing depths are always integers and a diagonal network has *uniform* mixing depths. $\square$

# G  Supplementary materials in Section 4.1.2 : $l_{p,q}$ group quasi-norms

In this section, we only consider networks without shared weights.

## G.1  Intuitions of group architectures

We give some intuitions on how we design group architectures. The main observation is that *if $\mathcal{N}$ induces $l_{p,q}$ quasi-norm as the complexity measure, then the subnetwork $\mathcal{N}_S$ with $S = \{i, j\}$ ( Def E.2) induces $l_q$ or $l_p$ quasi-norm depending on whether $i$ and $j$ are in the same group or not.* The reasoning for this observation is two-fold: Restricting a network to two nodes gives a subnetwork with "restricted induced complexity measure" as in Eq. (368); and restricting $l_{p,q}$ group quasi-norm to 2 sparse vectors gives $l_p$ or $l_q$ quasi-norms. Theorem 35 identifies the architectures which induce $l_p$ quasi-norm as the ones with *uniform* mixing depths $2/p$. Hence, the mixing depths of any architectures that induces $l_{p,q}$ quasi-norm satisfy

$$M_{\mathcal{N}}(\{i, j\}) = \begin{cases} 2/q & \text{if } i \text{ and } j \text{ are in the same group;} \\ 2/p & \text{if } i \text{ and } j \text{ are in different groups.} \end{cases} \tag{459}$$

Based on this, it is natural to consider architectures that consists of some diagonal layers followed by a *grouping layer* and then followed by a diagonal network (Section 3.1.2). This immediately gives us the group networks.

## G.2  Proof of Theorem 9

**Theorem 9.** *Let $G_1, G_2 \ldots G_k$ be a partition of $[n]$. Let $\beta_{G_j}$ be the projection of $\beta$ on $G_j$. Then for $d_2 > d_1$, $R_{\mathcal{N}^{1;d_1,d_2}}(\beta) = d_2 \sum_{j=1}^{k} \left\| \beta_{G_j} \right\|_{2/d_1}^{2/d_2} = d_2 \| \beta \|_{2/d_2, 2/d_1}^{2/d_2} \cong \| \beta \|_{2/d_2, 2/d_1}.$*

*Proof.* For each $j \in [k]$, let $\mathcal{N}_{G_j}^{1;d_1,d_2}$ be the subnetwork (See Def E.2) of $\mathcal{N}^{1;d_1,d_2}$ with respect to $G_j$. Then $\mathcal{N}_{G_1}^{1;d_1,d_2}, \ldots, \mathcal{N}_{G_k}^{1;d_1,d_2}$ form a partition of $\mathcal{N}^{1;d_1,d_2}$. Thus,

$$R_{\mathcal{N}^{1;d_1,d_2}}(\beta) = \sum_{j=1}^{k} R_{\mathcal{N}_{G_j}^{1;d_1,d_2}}(\beta_{G_j}). \tag{460}$$

Now, note that each $\mathcal{N}_{G_j}^{1;d_1,d_2}$ is an essentially diagonal network of depth $d_1$ concatenated with a path of depth $d_2$. By Lemma 49,

$$R_{\mathcal{N}_{G_j}^{1;d_1,d_2}}(\beta_{G_j}) = d_2 \left\| \beta_{G_j} \right\|_{2/d_1}^{2/d_2}. \tag{461}$$

Thus,

$$R_{\mathcal{N}^{1;d_1,d_2}}(\beta) = d_2 \sum_{j=1}^{k} \left\| \beta_{G_j} \right\|_{2/d_1}^{2/d_2}. \tag{462}$$

$\square$

## G.3 Proof of Theorem 10

**Theorem 10.** *When* $d_2 = d_1 + 1$, $R_{\mathcal{N}^{2;d_1,d_2}}(\beta) = d_2 \|\beta\|_{2/d_1, 2/d_2}^{2/d_2} \cong \|\beta\|_{2/d_1, 2/d_2}$.

*Proof.* Let $w$ be the weights on $\mathcal{N}^{2;d_1,d_1+1}$ such that

$$F_{\mathcal{N}^{2;d_1,d_1+1}}(w) = \beta, \qquad R_{\mathcal{N}^{2;d_1,d_1+1}}(\beta) = \|w\|_2^2. \tag{463}$$

Let $w_1, w_2$ denote the weights corresponding to the first $d_1 - 1$ diagonal layers and the remaining layers respectively. Let $T = \mathrm{diag}(t_1, \ldots, t_n)$ denote the diagonal matrix generated by $w_1$ in the first $d_1 - 1$ diagonal layers. Since $R_{\mathcal{N}^{2;d_1,d_1+1}}(\beta) = \|w\|_2^2$ ($w$ attains the minimum representation cost),

$$\|w_1\|_2^2 = \sum_{i=1}^{n} R_{\mathcal{N}_{FC}}(t_i) = (d_1 - 1) \sum_{i=1}^{n} |t_i|^{2/(d_1-1)}, \tag{464}$$

where $\mathcal{N}_{FC}$ denotes a fully connected neural network and the first equality follows from the fact that the first $d_1 - 1$ hidden layers are $n$ disjoint paths each of which is a fully connected network with one input node and one output node. Let $\mathcal{N}_{d_1-1:d_1+1}$ be the subnetwork of $\mathcal{N}$ corresponding to the last three layers. Note that $\mathcal{N}_{d_1-1:d_1+1}$ has the same architecture as $\mathcal{N}_{2,1}$ in section D.4. Let $a = F_{\mathcal{N}}(w_2)$. Then since

$$a^T T = \beta, \tag{465}$$

we have

$$a_i = \beta_i / t_i \tag{466}$$

for all $i \in [n]$. Since $R_{\mathcal{N}^{2;d_1,d_1+1}}(\beta) = \|w\|_2^2$ ($w$ attains the minimum representation cost),

$$\|w_2\|_2^2 = R_{\mathcal{N}_{d_1-1:d_1+1}}(a) = R_{\mathcal{N}_{2,1}}(a) = 2\sqrt{\sum_{j=1}^{k} \left( \sum_{i \in G_j} |a_i| \right)^2} = 2\sqrt{\sum_{j=1}^{k} \left( \sum_{i \in G_j} \frac{|\beta_i|}{|t_i|} \right)^2}. \tag{467}$$

Therefore,

$$\|w\|_2^2 = \|w_1\|_2^2 + \|w_2\|_2^2$$

$$= (d_1 - 1) \sum_{i=1}^{n} |t_i|^{2/(d_1-1)} + 2\sqrt{\sum_{j=1}^{k} \left( \sum_{i \in G_j} \frac{|\beta_i|}{|t_i|} \right)^2}$$

$$\overset{(a)}{\geq} (d_1 + 1) \left[ \left( \sum_{i=1}^{n} |t_i|^{2/(d_1-1)} \right)^{d_1-1} \left( \sum_{j=1}^{k} \left( \sum_{i \in G_j} \frac{|\beta_i|}{|t_i|} \right)^2 \right) \right]^{1/(d_1+1)}$$

$$= (d_1 + 1) \left\{ \left[ \sum_{j=1}^{k} \left( \left( \sum_{i \in G_j} |t_i|^{2/(d_1-1)} \right)^{(d_1-1)/d_1} \right)^{d_1/(d_1-1)} \right]^{(d_1-1)/d_1} \left[ \sum_{j=1}^{k} \left( \left( \sum_{i \in G_j} \frac{|\beta_i|}{|t_i|} \right)^{2/d_1} \right)^{d_1} \right]^{1/d_1} \right\}^{\frac{d_1}{d_1+1}}$$

$$\overset{(b)}{\geq} (d_1 + 1) \left\{ \sum_{j=1}^{k} \left[ \left( \sum_{i \in G_j} |t_i|^{2/(d_1-1)} \right)^{(d_1-1)/d_1} \left( \sum_{i \in G_j} \frac{|\beta_i|}{|t_i|} \right)^{2/d_1} \right] \right\}^{d_1/(d_1+1)}$$

$$= (d_1 + 1) \left\{ \sum_{j=1}^{k} \left[ \left( \sum_{i \in G_j} (|t_i|^{\frac{2}{d_1+1}})^{\frac{d_1+1}{d_1-1}} \right)^{\frac{d_1-1}{d_1+1}} \left( \sum_{i \in G_j} \left( \left( \frac{|\beta_i|}{|t_i|} \right)^{\frac{2}{d_1+1}} \right)^{\frac{d_1+1}{2}} \right)^{\frac{2}{d_1+1}} \right]^{\frac{d_1+1}{d_1}} \right\}^{\frac{d_1}{d_1+1}}$$

$$\overset{(c)}{\geq} (d_1 + 1) \left\{ \sum_{j=1}^{k} \left[ \sum_{i \in G_j} |\beta_i|^{\frac{2}{d_1+1}} \right]^{\frac{d_1+1}{d_1}} \right\}^{\frac{d_1}{d_1+1}}$$

$$= (d_1 + 1) \|\beta\|_{2/d_1, 2/(d_1+1)}^{2/(d_1+1)}, \tag{468}$$

where in $(a)$ we used AM-GM inequality, in $(b), (c)$ we used Holder's inequality.

Now, we show that this bound can be attained. To show this, it suffices to find a $t \in \mathbb{R}^n$ such that the bound in Eq. (468) is attained. We do this by first start with some arbitrary vector $t = (t_1, \ldots, t_n)$ and modify it step by step such that the inequalities in each step of Eq. (468) can be achieved with equality. This would imply that the bound in Eq. (468) is achievable. For each $j \in [k]$, let

$$t^{(j)} \in \mathbb{R}^{|G_j|} \tag{469}$$

denote the subvector of $t$ such that $t_i$ is an entry in $t^{(j)}$ if and only $i \in G_j$. Now, in order for $(c)$ to hold with equality, we modify $t^{(j)}$ such that

$$\frac{|t^{(j)}[i_1]|}{|t^{(j)}[i_2]|} = \frac{|\beta_{i_1}|}{|\beta_{i_2}|}^{(d_1+1)/(d_1-1)} \tag{470}$$

for all $i_1, i_2 \in G_j$, for all $j \in [k]$. Note that this requirement only depends on the ratio between entries in $t^{(j)}$ for each $j$. In other words, it will remain to hold if the ratio within-group does not change. Now, in order for $(b)$ to hold with equality, we scale each $t^{(j)}$ by $\lambda_j$ such that the ratio between-groups satisfy some certain requirement. Note that this does not affect the ratio within-groups and thus $(c)$ continues to hold with equality. Lastly, for $(a)$ to holds with equality, we scale the whole vector $t$ by some constant $\lambda$, which does not change the ratio between groups or the ratio within groups. Thus, $(b), (c)$ continue to hold with equality. Thus,

$$R_{\mathcal{N}^{2;d_1,d_1+1}}(\beta) = (d_1 + 1) \|\beta\|_{2/d_1, 2/(d_1+1)}^{2/(d_1+1)}. \tag{471}$$
$\square$

### G.4  Proof of Theorem 8

**Theorem 8.** *If there exists a linear homogeneous feedforward neural network $\mathcal{N}$ without shared weights that induces $l_{p,q}$ group quasi-norms, then $2/p, 2/q \in \mathbb{N}$. On the other hand, if $2/p, 2/q \in \mathbb{N}$ and $2/p \geq 2/q - 1$, then there exists a linear homogeneous feedforward neural network $\mathcal{N}$ without shared weights that induces $l_{p,q}$ group quasi-norms.*

*Proof.* This follows immediately from Corollary 53, Theorem 9 and Theorem 10.  $\square$

## G.5 Negative results of $\mathcal{N}^{2;d_1,d_2}$ when $d_2 > d_1 + 1$

Now, we give some negative results. We will show that $\mathcal{N}^{2;1,d_2}$ does not induce $l_{2,2/d_2}$ quasi-norm as induced complexity measure for all $d_2 \geq 3$. We do this in two steps. First, we will define a new quasi-norm $f$, which is monotonic in $l_{2,2/d_2}$ quasi-norm. Then, we use $f$ as a reference function (Section E.3) to show that $\mathcal{N}^{2;1,d_2}$ does not induce $l_{2,2/d_2}$ quasi-norm as induced complexity measure. Note that $f$ is not a parametrization here.

**Lemma 54.** *Let* $f : \mathbb{R}^n \longrightarrow \mathbb{R}_+$ *be defined as*

$$f(\beta) = d_2 \min \left\{ \sum_{h \in N_1} \|v_h\|_2^{2/d_2} : v_h \in \mathbb{R}^n, \text{supp}(v_h) \subseteq S_h, \left( \sum_{h \in N_1} v_h^{2/d_2} \right)^{d_2/2} = \beta \right\}, \quad (472)$$

*where* $N_1$ *and* $S_h$ *are defined as in section D.4 Eq.* (287) *and* (288) *respectively, and the exponents are applied component-wise(i.e* $a^k[i] = a[i]^k$*). Then*

$$f(\beta) = d_2 \|\beta\|_{2,2/d_2}^{2/d_2}. \quad (473)$$

*Proof.* Let $\phi : \mathbb{R}^n \longrightarrow \mathbb{R}^n$ be defined as

$$\phi(x)[i] = x[i]^{2/d_2}. \quad (474)$$

Then

$$\|v_h\|_2^{2/d_2} = \left( \sum_{i=1}^n v_h[i]^2 \right)^{1/d_2} = \left( \sum_{i=1}^n |\phi(v_h)[i]|^{d_2} \right)^{1/d_2} = \|\phi(v_h)\|_{d_2}. \quad (475)$$

Also, note that

$$\sum_{h \in N_1} \phi(v_h) = \phi(\beta) \quad (476)$$

if and only if

$$\left( \sum_{h \in N_1} v_h^{2/d_2} \right)^{d_2/2} = \beta. \quad (477)$$

Let $g(\phi(\beta)) = f(\beta)/d_2$. Then

$$g(\phi(\beta)) = \min \left\{ \sum_{h \in N_1} \|\phi(v_h)\|_{d_2} : \text{supp}(v_h) \subseteq S_h, \sum_{h \in N_1} \phi(v_h) = \phi(\beta) \right\}. \quad (478)$$

Note that $g$ is a norm. Let $g^*$ denote its dual norm. Let $d^* > 0$ be such that

$$\frac{1}{d_2} + \frac{1}{d^*} = 1. \quad (479)$$

By the similar arguments (with $l_2$ norm changing to $l_{d_2}$ norm) in Lemma 6 and Theorem 34, we have

$$\begin{aligned} g^*(\phi(\beta)) &= \max \left\{ \left( \sum_{i \in S_h} |\phi(\beta)[i]|^{d^*} \right)^{1/d^*} : h \in N_1 \right\} \\ &= \left( \sum_{j=1}^k \left( \max_{i \in G_j} |\phi(\beta)[i]| \right)^{d^*} \right)^{1/d*} \\ &= \|\phi(\beta)\|_{d^*,\infty}. \end{aligned} \quad (480)$$

Thus,

$$g(\phi(\beta)) = \|\phi(\beta)\|_{d_2,1}. \quad (481)$$

Then

$$\begin{aligned}
f(\beta) &= d_2 g(\phi(\beta)) \\
&= d_2 \|\phi(\beta)\|_{d_2,1} \\
&= d_2 \left( \sum_{j=1}^{k} \left( \sum_{i \in G_j} |\phi(\beta)[i]| \right)^{d_2} \right)^{1/d_2} \\
&= d_2 \left( \sum_{j=1}^{k} \left( \sum_{i \in G_j} |\beta_i|^{2/d_2} \right)^{d_2} \right)^{1/d_2} \\
&= d_2 \|\beta\|_{2,2/d_2}^{2/d_2}.
\end{aligned}$$ (482)

$\square$

**Lemma 55.** *Let $d_2 \geq 3$. For any $\beta \in \mathbb{R}^n$,*

$$R_{\mathcal{N}^{2;1,d_2}}(\beta) \geq f(\beta),$$ (483)

*where equality is attained only if there exists $v_h \in \mathbb{R}^n$ for each $h \in N_1$ such that the supports for $v_h$ are disjoint,*

$$\sum_{h \in N_1} v_h = \beta, \qquad \operatorname{supp}(v_h) \subseteq S_h \quad \forall h \in N_1,$$ (484)

*and*

$$\sum_{h \in N_1} \|v_h\|_2^{2/d_2} = \|\beta\|_{2,2/d_2}^{2/d_2}.$$ (485)

*Proof.* By the same argument as in Lemma 5, we have

$$R_{\mathcal{N}^{2;1,d_2}}(\beta) = d_2 \min \left\{ \sum_{h \in N_1} \|v_h\|_2^{2/d_2} : \operatorname{supp}(v_h) \subseteq S_h, \sum_{h \in N_1} v_h = \beta \right\},$$ (486)

for any $\beta \in \mathbb{R}^n$. Fix a $\beta \in \mathbb{R}^n$. Let $\{v_h : h \in N_1\}$ be such that

$$\sum_{h \in N_1} v_h = \beta, \qquad \operatorname{supp}(v_h) \subseteq S_h,$$ (487)

and

$$R_{\mathcal{N}^{2;1,d_2}}(\beta) = d_2 \sum_{h \in N_1} \|v_h\|_2^{2/d_2}.$$ (488)

For each $i \in [n]$, let

$$\lambda_i = \frac{\sum_{h \in N_1} v_h[i]}{\left( \sum_{h \in N_1} v_h[i]^{2/d_2} \right)^{d_2/2}}.$$ (489)

Since $d_2 \geq 3$,

$$\left( \sum_{h \in N_1} v_h[i]^{2/d_2} \right)^{d_2/2} \geq \sum_{h \in N_1} v_h[i],$$ (490)

where equality is attained only if all but one of the $v_h[i]$s are zero. Thus,

$$\lambda_i \leq 1,$$ (491)

for all $i \in [n]$. Now, for each $i \in [n]$, for each $h \in N_1$ let

$$w_h[i] = \lambda_i v_h[i].$$ (492)

Then

$$\left( \sum_{h \in N_1} w_h^{2/d_2} \right)^{d_2/2} = \beta, \qquad \operatorname{supp}(w_h) \subseteq S_h,$$ (493)

and

$$f(\beta) \le d_2 \sum_{h \in N_1} \|w_h\|_2^{2/d_2} \le d_2 \sum_{h \in N_1} \|v_h\|_2^{2/d_2} = R_{\mathcal{N}^{2;1,d_2}}(\beta), \tag{494}$$

where equality is attained only if

$$\lambda_i = 1, \tag{495}$$

for all $i \in [n]$. This occurs only if all but one of the $v_h[i]$s are zero for all $i \in [n]$, which implies that the supports of $v_h$s are disjoint. $\qquad\square$

Now, we are going to prove that $\mathcal{N}^{2;1,d_2}$ does not induce $l_{2,2/d_2}$ quasi-norm as induced complexity measure. Note that if $\mathcal{N}^{2;1,d_2}$ does not induce $l_{2,2/d_2}$ quasi-norm as induced complexity measure for any nontrivial grouping, then there have to be at least two groups and one of which should contain at least one element. Then, without loss of generality, we can assume that

$$\{1,3\} \subseteq G_1, \qquad 2 \in G_2. \tag{496}$$

Then, if we remove all input nodes except for the first three of them, then the resulting architecture would also induce $l_{2,2/d_2}$ quasi-norm as induced complexity measure. Thus, in order to show that $\mathcal{N}^{2;1,d_2}$ does not induce $l_{2,2/d_2}$ quasi-norm as induced complexity measure, we can assume without loss of generality that

$$n = 3, \qquad k = 2, \qquad G_1 = \{1,3\}, \qquad G_2 = \{2\}. \tag{497}$$

Note that the above argument still holds if we change $\mathcal{N}^{2;1,d_2}$ to any other candidate architecture $\mathcal{N}$ that is supposed to induce $l_{2,2/d_2}$ quasi-norm as induced complexity measure. Now, we state the result.

**Lemma 56.** *If $d_2 \ge 3$, then $\mathcal{N}^{2;1,d_2}$ does not induce $l_{2,2/d_2}$ quasi-norm as induced complexity measure.*

*Proof.* As we just discussed, we can assume without loss of generality that

$$n = 3, \qquad k = 2, \qquad G_1 = \{1,3\}, \qquad G_2 = \{2\}. \tag{498}$$

For any $\beta \in \mathbb{R}^n$,

$$\|\beta\|_{2,2/d_2} = \sqrt{\sum_{j=1}^k \left( \sum_{i \in G_j} |\beta_i|^{2/d_2} \right)^{d_2}} = \sqrt{(|\beta_1|^{2/d_2} + |\beta_3|^{2/d_2})^{d_2} + |\beta_2|^2}. \tag{499}$$

Note that $N_1 = (h_1, h_2)$ and

$$S_{h_1} = \{1,2\}, \qquad S_{h_2} = \{2,3\}. \tag{500}$$

Let

$$\beta' = (1, 2^{d_2/2}, 1). \tag{501}$$

Suppose for the sake of contradiction that

$$f(\beta') = R_{\mathcal{N}^{2;1,d_2}}(\beta'). \tag{502}$$

Then, by Lemma 55, there exists $v_1, v_2$ such that

$$\mathrm{supp}(v_1) \subseteq S_{h_1}, \quad \mathrm{supp}(v_2) \subseteq S_{h_2}, \quad \mathrm{supp}(v_1) \cap \mathrm{supp}(v_2) = \emptyset, \tag{503}$$

and

$$\sum_{h \in N_1} v_h = \beta', \qquad \sum_{h \in N_1} \|v_h\|_2^{2/d_2} = \|\beta'\|_{2,2/d_2}^{2/d_2}. \tag{504}$$

Without loss of generality, assume that

$$\mathrm{supp}(v_1) = \{1\}, \qquad \mathrm{supp}(v_2) = \{2,3\}. \tag{505}$$

Then

$$v_1 = (1, 0, 0), \qquad v_2 = (0, 2^{d_2/2}, 1). \tag{506}$$

Then
$$\sum_{h \in N_1} \|v_h\|_2^{2/d_2} = 1 + (2^{d_2} + 1)^{1/d_2} > 1 + 2 = 3. \tag{507}$$

However,
$$\|\beta'\|_{2,2/d_2}^{2/d_2} = ((|\beta_1|^{2/d_2} + |\beta_3|^{2/d_2})^{d_2} + |\beta_2|^2)^{1/d_2} = 2^{(d_2+1)/d_2} < 2\sqrt{2} < 3, \tag{508}$$

since $d_2 > 2$. Thus,
$$R_{\mathcal{N}^{2;1,d_2}}(\beta') > f(\beta'). \tag{509}$$

Now, let
$$\beta'' = \lambda(1, 0, 0), \tag{510}$$

where $\lambda > 0$ is chosen such that
$$\|\beta''\|_{2,2/d_2} = \|\beta'\|_{2,2/d_2}. \tag{511}$$

Recall that
$$R_{\mathcal{N}^{2;1,d_2}}(\beta'') = d_2 \min \left\{ \sum_{h \in N_1} \|v_h\|_2^{2/d_2} : \text{supp}(v_h) \subseteq S_h, \sum_{h \in N_1} v_h = \beta'' \right\}. \tag{512}$$

Now, if we choose $v_1 = \beta''$ and $v_2 = 0$, then
$$R_{\mathcal{N}^{2;1,d_2}}(\beta'') \leq d_2 \|\beta''\|_2^{2/d_2} = d_2 \lambda^{2/d_2} = d_2 \|\beta''\|_{2,2/d_2}^{2/d_2} = d_2 \|\beta'\|_{2,2/d_2}^{2/d_2} = f(\beta') < R_{\mathcal{N}^{2;1,d_2}}(\beta'). \tag{513}$$

Thus, $\mathcal{N}^{2;1,d_2}$ does not induce $l_{2,2/d_2}$ quasi-norm as induced complexity measure. $\square$

By a similar argument as in the proof of Theorem 10, we can show that $\mathcal{N}^{2;d_1,d_2}$ does not induce $l_{2/d_1,2/d_2}$ quasi-norm when $d_2 > d_1 + 1$. Roughly speaking, we can get a similar chain of inequality as in Eq. 468. However, this time it cannot be achieved since $\mathcal{N}^{2;1,d_2-d_1+1}$ does not induce $l_{2,2/(d_2-d_1+1)}$ quasi-norm as induced complexity measure, and the equality in the second step of Eq. 468 becomes strict inequality.

# H  Supplementary materials in Section 4.2 : negative results on homogeneous neural networks

In this section, we only consider networks without shared weights.

Elastic nets is defined as
$$\|\beta\|_{EN} = \|\beta\|_1 + \alpha \|\beta\|_2, \tag{514}$$
where $\alpha > 0$ is some constant.

To show the impossibility of designing architectures with these induced complexities, we make the following observation.

**Lemma 57.** *Let $h : \mathbb{R}^n \longrightarrow \mathbb{R}_+$ be a function that is the induced complexity measure of some linear homogeneous feedforward neural network $\mathcal{N}$ without shared weights. Let $i, j \in [n], i < j$ be two distinct indices. Let $h_{i,j} : \mathbb{R}^2 \longrightarrow \mathbb{R}_+$ be defined as*
$$h_{i,j}(\beta') = h(\beta) \tag{515}$$
*where $\beta_i = \beta_1'$, $\beta_j = \beta_2'$, and $\beta_k = 0$ for all $k \notin \{i, j\}$ ($\beta$ is the lifting of $\beta'$). Then, there exists a strictly increasing function $\phi : \mathbb{R}_+ \longrightarrow \mathbb{R}_+$ and $p = 2/d$ for some $d \in \mathbb{N}$ such that*
$$h_{i,j}(\beta') = \phi(\|\beta'\|_p), \tag{516}$$

*for all $\beta' \in \mathbb{R}^2$, where $\|\cdot\|_p$ denotes the $l_p$ quasi-norm.*

*Proof.* Let $\mathcal{N}_{i,j}$ be the subnetwork of $\mathcal{N}$ with respect to the $i, j$th input nodes (Def E.2). By Lemma 44, $\mathcal{N}_{i,j}$ induces $h_{i,j}$ as induced complexity measure. On the other hand, $\mathcal{N}_{i,j}$ induces $l_{2/d}$ quasi-norm as induced complexity measure for some $d \in \mathbb{N}$ by Lemma 51. Thus, the result follows.

$\square$

The above result shows that any function that is the induced complexity measure of some linear homogeneous feedforward neural network always behaves like a $l_p$ quasi-norm on 2-sparse vectors. The result is a direct consequence of Lemma 44 and Lemma 51.

Since neither elastic nets nor $l_{p,q}$ quasi-norms with overlapping between groups satisfy this property, they are not induced complexity measure of any architecture $\mathcal{N}$.

**Theorem 58.** *For any $\alpha > 0$, elastic nets $\|\cdot\|_{EN}$ defined in Eq. (514) is not the induced complexity measure of any linear homogeneous feedforward neural network without shared weights.*

Next, we give the negative result for $l_{p,q}$ group quasi-norm with overlapping groups.

*Proof.* Let $h(\cdot) = \|\cdot\|_{EN}$. Let $h_{1,2} : \mathbb{R}^2 \longrightarrow \mathbb{R}_+$ be defined as

$$h_{1,2}(\beta') = h(\beta) \tag{517}$$

where $\beta$ is obtained from $\beta'$ by putting zeros to the $k$th entries for all $k \notin \{1, 2\}$. Then

$$h_{1,2}(\beta_1, \beta_2) = |\beta_1| + |\beta_2| + \alpha\sqrt{\beta_1^2 + \beta_2^2}. \tag{518}$$

Suppose $h_{1,2}$ is monotonic in $l_p$ quasi-norm for some $p = 2/d$. Then

$$h_{1,2}(1, 1) = h_{1,2}(2^{1/p}, 0), \tag{519}$$

since

$$\|(1, 1)\|_p = \left\|(2^{1/p}, 0)\right\|_p. \tag{520}$$

Note that

$$h_{1,2}(2^{1/p}, 0) = 2^{1/p}(1 + \alpha), \qquad h_{1,2}(1, 1) = 2 + \alpha\sqrt{2}. \tag{521}$$

Thus, we have

$$2 + \alpha\sqrt{2} = 2^{1/p}(1 + \alpha). \tag{522}$$

If $d \geq 2$, then $p \leq 1$ and

$$2^{1/p}(1 + \alpha) \geq 2(1 + \alpha) > 2 + \alpha\sqrt{2}. \tag{523}$$

Thus, $d = 1$ and $p = 2$. However,

$$2^{1/p}(1 + \alpha) = \sqrt{2}(1 + \alpha) < 2 + \alpha\sqrt{2}. \tag{524}$$

Thus, $h_{1,2}$ is not monotonic in $l_p$ quasi-norm for any $p = 2/d$. Thus, by Lemma 57, $h$ cannot be induced as induced complexity measure by any linear neural network. $\square$

The $l_{p,q}$ group quasi-norm with overlapping groups is defined as

$$\|\beta\|_{p,q} = \Big(\sum_{j=1}^k \Big(\sum_{i \in G_j} |\beta_i|^q\Big)^{p/q}\Big)^{1/p}, \tag{525}$$

where $G_1, G_2 \ldots G_k \subseteq [n]$.

**Theorem 59.** *Let $G_1, \ldots, G_k \subseteq [n]$ such that*

$$\bigcup_{j=1}^k G_j = [n]. \tag{526}$$

*Let $\|\cdot\|_{p,q}$ be the $l_{p,q}$ group quasi-norm with respect to $G_1, \ldots, G_k$ for some $p, q > 0, p \neq q$, defined in Eq. (525). Suppose that there exists $i, j \in [n]$ and $s, t \in [k]$ such that*

$$\{i, j\} \subseteq G_s, \qquad i \in G_t, \qquad j \notin G_t. \tag{527}$$

*Then $\|\cdot\|_{p,q}$ is not the induced complexity measure of any linear homogeneous feedforward neural network without shared weights.*

Note that the assumption in the above result is necessary. If the assumption does not hold, then for all $i, j$, $x_i$ and $x_j$ are either always in the same group or always in different groups. However, this implies that the resulting quasi-norm is $l_{p,q}$ quasi-norm without overlapping between groups, provided that

$$G_s \neq G_t \quad \forall s \neq t. \tag{528}$$

Indeed, the assumption in the above result exactly characterizes $l_{p,q}$ quasi-norm with overlapping groups.

*Proof.* Let $h(\cdot) = \|\cdot\|_{p,q}$. Without loss of generality, assume that $i = 1, j = 2$. Let $w_1$ denote the number of groups that only contains 1. Let $w_2$ denote the number of groups that only contains 2. Let $w_3$ be the number of groups that contain both 1 and 2. Let $h_{1,2} : \mathbb{R}^2 \longrightarrow \mathbb{R}_+$ be defined as

$$h_{1,2}(\beta') = h(\beta) \tag{529}$$

where $\beta$ is obtained from $\beta'$ by putting zeros to the $k$th entries for all $k \notin \{1, 2\}$. Then

$$h_{1,2}(x)^p = w_3(|x_1|^q + |x_2|^q)^{p/q} + w_1|x_1|^p + w_2|x_2|^p. \tag{530}$$

Since we only care about the value of $h_{1,2}$ up to monotonic transformation, we can assume that

$$w_3 = 1. \tag{531}$$

Note that at least one of $w_1$ and $w_2$ is nonzero by assumption. Now, suppose for the sake of contradiction that $h_{1,2}$ is monotonic in $l_{2/d}$ quasi-norm for some $d \in \mathbb{N}$. Then

$$h_{1,2}(1, 0) = h_{1,2}(0, 1), \tag{532}$$

which implies that

$$w_1 = w_2 = c \tag{533}$$

for some $c > 0$. Now,

$$h_{1,2}(x)^p = (|x_1|^q + |x_2|^q)^{p/q} + c(|x_1|^p + |x_2|^p). \tag{534}$$

Since $h_{1,2}$ is monotonic in $l_{2/d}$ quasi-norm, it is constant on

$$\{(t^{d/2}, (1 - t)^{d/2}) : t \in (0, 1)\}. \tag{535}$$

Let

$$g(t) = h_{1,2}((t^{d/2}, (1 - t)^{d/2}))^p = (t^{qd/2} + (1 - t)^{qd/2})^{p/q} + c(t^{pd/2} + (1 - t)^{pd/2}). \tag{536}$$

Let

$$r = p/q, \qquad s = qd/2. \tag{537}$$

Then

$$g(t) = (t^s + (1 - t)^s)^r + c(t^{rs} + (1 - t)^{rs}). \tag{538}$$

Since $p \neq q$,

$$r \neq 1. \tag{539}$$

Now, if $s = 1$, then

$$g(t) = 1 + c(t^r + (1 - t)^r), \tag{540}$$

which is clearly not constant since $r \neq 1$ and $c > 0$. Thus,

$$s \neq 1. \tag{541}$$

Since $g(t)$ is constant, its derivative is 0:

$$g'(t) = rs(t^s + (1 - t)^s)^{r-1}(t^{s-1} - (1 - t)^{s-1}) + crs(t^{rs-1} - (1 - t)^{rs-1}) = 0, \tag{542}$$

which implies that

$$g'(t)/rs = (t^s + (1 - t)^s)^{r-1}(t^{s-1} - (1 - t)^{s-1}) + c(t^{rs-1} - (1 - t)^{rs-1}) = 0, \tag{543}$$

for all $t \in (0, 1)$. Then

$$\lim_{t \to 0} g'(t)/rs = 0. \tag{544}$$

However,

$$\lim_{t \to 0} g'(t)/rs = -1 - c < 0 \tag{545}$$

if $s > \max(1, 1/r)$,

$$\lim_{t \to 0} g'(t)/rs = \infty \tag{546}$$

if $s < 1$ or $s < 1/r$,

$$\lim_{t \to 0} g'(t)/rs = -1 < 0 \tag{547}$$

if $s = 1/r > 1$. This is a contradiction. Thus, $h_{1,2}$ is not monotonic in $l_{2/d}$ quasi-norm for any $d \in \mathbb{N}$. Thus, by Lemma 57, $h$ cannot be induced as induced complexity measure by any linear neural network. $\qquad \square$

# I  Supplementary materials: homogeneous parameterizations

In this section, we consider general homogeneous parametrizations. We will show several homogeneous parameterizations whose induced complexity measure cannot be induced by any linear neural network.

First, we recall the setup. We consider parameterized mappings $f : \mathcal{X} \times \mathbb{R}^p \longrightarrow \mathbb{R}^m$, from input $x \in \mathcal{X}$ and parameters $w \in \mathbb{R}^p$ to predictions $f(x; w)$. We denote the predictor implemented with parameters $w$ by $F(w) : \mathcal{X} \longrightarrow \mathbb{R}^m$ defined as $F(w)(x) := f(x; w)$. Then $\text{image}(F)$ is the set of functions from $\mathcal{X}$ to $\mathbb{R}^m$ which can be obtained from this class of parameterized models. In this section, we consider the single output linear models, i.e., $m = 1$, $\mathcal{X} = \mathbb{R}^n$, and $\text{image}(F)$ is the set of linear transformations, which can be identified as $\mathbb{R}^n$.

*The representation cost* ([15, 31, 27]) of a function $g$ in $\text{image}(F)$ under the parametrization $F$ is

$$R_F(g) = \min\{\|w\|_2^2 : F(w) = g\}. \tag{548}$$

## I.1  Elastic nets

In this section, we consider the Elastic Nets penalty defined as:

$$\|\beta\|_1 + \alpha\|\beta\|_2, \tag{549}$$

for $\beta \in \mathbb{R}^n$, where $\alpha > 0$ is some constant. Let $w_1 = (w_1, w_2), w_2 = (W_3, w_4)$, where $w_1, w_2, w_4 \in \mathbb{R}^n$ and $W_3 \in \mathbb{R}^{n \times n}$. Let

$$W_1 = \text{diag}(w_1). \tag{550}$$

Let $w = (w_1, w_2)$ be the parameter, and $\mathcal{X} = \mathbb{R}^n$. Let

$$f_{EN}(x; w) = \text{sign}(w_2^T W_1) \min(2|w_2^T W_1|, 2\alpha^{-1}|w_4^T W_3|)x, \tag{551}$$

where $\min(\cdot)$, $\text{sign}(\cdot)$, and absolute value $|\cdot|$ are applied component-wise.

**Theorem 60.** *For any $\beta \in \mathbb{R}^n$,*

$$R_{F_{EN}}(\beta) = \|\beta\|_1 + \alpha\|\beta\|_2 = \|\beta\|_{EN}. \tag{552}$$

Thus, the induced complexity measure induced by $F_{EN}$ is an elastic net.

*Proof.* Let $w = (w_1, w_2, W_3, w_4)$ be such that $\beta = f_{EN}(\cdot; w)$. Let

$$\beta'^T = w_2^T W_1, \qquad \beta''^T = w_4^T W_3. \tag{553}$$

By results in linear fully connected networks and linear diagonal networks, we have

$$R_{F_{EN}}(\beta) = \min\{2\|\beta'\|_1 + 2\|\beta''\|_2 : \text{sign}(\beta') \min(2|\beta'|, 2\alpha^{-1}|\beta''|) = \beta\}. \tag{554}$$

Since $|\beta| = \min(2|\beta'|, 2\alpha^{-1}|\beta''|)$,

$$R_{F_{EN}}(\beta) \geq 2\|\beta/2\|_1 + 2\|\alpha\beta/2\|_2 = \|\beta\|_1 + \alpha\|\beta\|_2, \tag{555}$$

where equality is achieved when $2\beta' = 2\alpha^{-1}\beta'' = \beta$. $\qquad\square$

In the proof of Theorem 60, the key step is to answer the follow question: Given two parameterizations $F_1, F_2$, how can we find another parameterization $F$ such that $R_F(\cdot) = R_{F_1}(\cdot) + R_{F_2}(\cdot)$? The answer is taking component-wise minimum $\min(\cdot)$. Similarly, we can extend this result to an arbitrary number of parameterizations.

**Lemma 61.** *Let $f_1(\cdot; w_1), \ldots, f_k(\cdot; w_k)$ be $k$ linear predictors such that $R_{F_j}(\beta)$ is strictly increasing in $|\beta_i|$ for all $i \in [n]$ and only depends on $|\beta|$, for all $j \in [k]$, where $|\cdot|$ is component-wise absolute value. Let*

$$f(x; w) = \text{sign}(f_1(\cdot; w_1)) \min_{j \in [k]}(|f_j(\cdot; w_j)|)^T x, \tag{556}$$

*where $\min(\cdot)$ and absolute value $|\cdot|$ are taken component-wise, and $w = (w_1, \ldots, w_k)$. Then,*

$$R_F(\beta) = \sum_{j=1}^{k} R_{F_j}(\beta). \tag{557}$$

*In addition, if $F_j$s are all positively homogeneous of degree $L > 0$, then $F$ is also positively homogeneous of degree L.*

*Proof.* For any $\beta \in \mathbb{R}^n$,

$$
\begin{aligned}
R_F(\beta) &= \min \left\{ \sum_{j=1}^{k} \|w_j\|_2^2 : \text{sign}(f_1(\cdot; w_1)) \min_{j \in [k]}(|f_j(\cdot; w_j)|) = \beta \right\} \\
&= \min \left\{ \sum_{j=1}^{k} \min\{\|w_j\|_2^2 : f_j(\cdot; w_j) = \gamma_j\} : \text{sign}(\gamma_1) \min_{j \in [k]}(|\gamma_j|) = \beta \right\} \\
&= \min \left\{ \sum_{j=1}^{k} R_{F_j}(\gamma_j) : \text{sign}(\gamma_1) \min_{j \in [k]}(|\gamma_j|) = \beta \right\} \\
&\overset{(a)}{=} \min \left\{ \sum_{j=1}^{k} R_{F_j}(|\gamma_j|) : \text{sign}(\gamma_1) \min_{j \in [k]}(|\gamma_j|) = \beta \right\} \\
&\overset{(b)}{=} \sum_{j=1}^{k} R_{F_j}(|\beta|) \\
&\overset{(c)}{=} \sum_{j=1}^{k} R_{F_j}(\beta),
\end{aligned}
\tag{558}
$$

where in $(a), (c)$ we used the fact that $R_{F_j}(\beta)$ depends only on $|\beta|$, and in $(b)$ we used the fact that $R_{F_j}(\beta)$ is strictly increasing in $|\beta|$.

In addition, suppose that $f_j$ is positively homogeneous of degree $L > 0$ for all $j \in [k]$. Then for all $\lambda > 0$, for all $j \in [k]$,

$$
f_j(\cdot; \lambda w)^T x = f_j(x; \lambda w) = \lambda^L f_j(x; w) = \lambda^L f_j(\cdot; w)^T x,
\tag{559}
$$

for all $x \in \mathbb{R}^n$. Thus,

$$
f_j(\cdot; \lambda w) = \lambda^L f_j(\cdot; w)
\tag{560}
$$

for all $j \in [k]$. Then for all $\lambda > 0$,

$$
\begin{aligned}
f(x; \lambda w) &= \text{sign}(f_1(\cdot; \lambda w_1)) \min_{j \in [k]}(|f_j(\cdot; \lambda w_j)|)^T x \\
&= \text{sign}(\lambda^L f_1(\cdot; w_1)) \min_{j \in [k]}(\lambda^L |f_j(\cdot; w_j)|)^T x \\
&= \lambda^L \text{sign}(f_1(\cdot; w_1)) \min_{j \in [k]}(|f_j(\cdot; w_j)|)^T x \\
&= \lambda^L f(x; w).
\end{aligned}
\tag{561}
$$

Thus, $f$ is positively homogeneous of degree $L$.

$\square$

## I.2 $l_p$ quasi-norms

Let $p \in (0, \infty)$, and $\mathcal{X} = \mathbb{R}^n$. Our goal is to find a homogeneous parameterization which induces $l_p$ quasi-norm as induced complexity measure. Let $\phi_p : \mathbb{R} \longrightarrow \mathbb{R}$ be defined as

$$
\phi_p(z) = \text{sign}(z)|z|^{2/p}.
\tag{562}
$$

Let

$$
f_p(x; w) = \phi_p(w)^T x,
\tag{563}
$$

where $x; w \in \mathbb{R}^n$ and $\phi$ is applied component-wise.

**Theorem 62.** *For any $\beta \in \mathbb{R}^n$,*

$$
R_{F_p}(\beta) = \|\beta\|_p^p \cong \|\beta\|_p.
\tag{564}
$$

Thus, $F_p$ induces $l_p$ quasi-norm as induced complexity measure.

*Proof.* Let $w \in \mathbb{R}^p$ such that

$$f_p(\cdot; w) = \beta \quad \text{and} \quad \|w\|_2^2 = R_{F_p}(\beta). \tag{565}$$

Then since $\phi_p(w) = \beta$,

$$|w[i]| = |\beta_i|^{p/2} \tag{566}$$

for all $i \in [n]$. Then, we would have

$$R_{F_p}(\beta) = \|w\|_2^2 = \|\beta\|_p^p. \tag{567}$$

$\square$

Note that this parameterization cannot be realized by any linear neural network since it is not multilinear. Indeed, as we have seen in Theorem 35, not all $l_p$ quasi-norms can be induced as induced complexity measure by neural networks.

### I.3  $l_{p,q}$ group quasi-norm with overlapping between groups

In Corollary 59, we showed that $l_{p,q}$ quasi-norm with overlapping between groups cannot be induced as induced complexity measure by any linear neural networks. In contrast, we will show that with homogeneous parameterization, we can indeed induce $l_{p,q}$ quasi-norm with overlapping between groups as induced complexity measure for all $p, q > 0$. We will give two classes of homogeneous parameterizations for the case $p < q$ and $p > q$ respectively.

We begin with the case $p < q$. The strategy is to first find a parameterization $f_q^p$ such that

$$R_{F_q^p}(\beta) = c\|\beta\|_q^p, \tag{568}$$

for some constant $c$. Then, we will use something similar to Lemma 61 to get the parameterization $F_{p,q}$ whose representation cost is

$$R_{F_{p,q}}(\beta) = \sum_{j=1}^{k} cR_{F_q^p}(\beta_{G_j}) = c\sum_{j=1}^{k}\left(\sum_{i \in G_j}|\beta_i|^q\right)^{p/q} = c\sum_{j=1}^{k}\|\beta_{G_j}\|_q^p, \tag{569}$$

where $G_1, \ldots, G_k$ are the groups and $\beta_{G_j}$ denotes the projection of $\beta$ onto the coordinates that correspond to elements in $G_j$. Thus, $F_{p,q}$ induces $l_{p,q}$ quasi-norm with overlapping between groups as induced complexity measure.

Now, we give the parameterization $F_q^p$ whose representation cost satisfies equation (568) for $p < q$. We will use the following generalized AM-GM inequality:

$$x^t y^{1-t} \leq tx + (1-t)y \tag{570}$$

for any $x, y \geq 0$ and $t \in [0, 1]$. Fix $t \in [0, 1]$. We will find parameterization $F_{p,q}$ for $p = tq$. Based on the previous paragraph, we will first find parameterization $F_q^{tq}$ such that

$$R_{F_q^{tq}}(\beta) = c\|\beta\|_q^{tq}, \tag{571}$$

for some constant $c$. Let $q \in (0, \infty)$. Let $\mathcal{X} = \mathbb{R}^n$. Let $\phi_q : \mathbb{R} \longrightarrow \mathbb{R}$ be defined as

$$\phi_q(z) = \text{sign}(z)|z|^{2/q}. \tag{572}$$

Let $w = (w_1, w_2)$ be the parameters where $w_1 \in \mathbb{R}^n$ and $w_2 \in \mathbb{R}$. For any $w = (w_1, w_2) \in \mathbb{R}^{n+1}$ and $x \in \mathbb{R}^n$, let

$$f_q^{tq}(x; w) = \phi_q(w_2^{(1-t)/t}w_1)^T x, \tag{573}$$

where $\phi_q$ is applied component-wise.

**Lemma 63.** *For any $\beta \in \mathbb{R}^n$,*

$$R_{F_q^{tq}}(\beta) = c_t\|\beta\|_q^{tq} \cong \|\beta\|_q, \tag{574}$$

*where $c_t = 1/(t^t(1-t)^{1-t})$.*

*Proof.* Let $w = (w_1, w_2) \in \mathbb{R}^{n+1}$ be such that $\phi(w_2^{(1-t)/t} w_1) = \beta$. Let $\beta' = w_2^{(1-t)/t} w_1$. Then

$$\|w_1\|_2^2 + w_2^2 = \frac{\|\beta'\|^2}{w_2^{2(1-t)/t}} + w_2^2 \overset{(a)}{\geq} \frac{\|\beta'\|_2^{2t}}{t^t(1-t)^{1-t}}, \tag{575}$$

where in $(a)$ we used the generalized AM-GM inequality (570), which can be attained when

$$\frac{\|\beta'\|^2}{t w_2^{2(1-t)/t}} = \frac{w_2^2}{1-t}. \tag{576}$$

Now, since $\phi(\beta') = \beta$,

$$|\beta_i'| = |\beta_i|^{q/2} \tag{577}$$

for all $i \in [n]$. Thus,

$$R_{F_q^{tq}}(\beta) = \frac{\|\beta'\|_2^{2t}}{t^t(1-t)^{1-t}} = \frac{\|\beta\|_q^{tq}}{t^t(1-t)^{1-t}}. \tag{578}$$

$\square$

Using some technique similar to Lemma 61, we get the following result.

**Theorem 64.** *Let $G_1, \ldots, G_k \subseteq [n]$ be the groups. For $x \in \mathbb{R}^n$, let $x_{G_j}$ denote the projection of $x$ on coordinates that correspond to elements in $G_j$. Let $w = (w_1, \ldots, w_k)$ be the parameters, where $w_j \in \mathbb{R}^{|G_j|+1}$ for all $j \in [k]$. For each $j \in [k]$, let*

$$f_j(x; w_j) = f_q^{tq}(x_{G_j}; w_j), \tag{579}$$

*where $f_q^{tq}$ is defined in equation (573). So $f_j(\cdot; w_j)$ is the lifting of $f_q^{tq}(\cdot; w_j)$ by putting zeroes to coordinates not in $G_j$. For each $i \in [n]$, pick $j_i \in [k]$ such that $i \in G_{j_i}$. Let*

$$f_{tq,q}(x; w) = \sum_{i=1}^n \text{sign}(f_{j_i}(\cdot; w_{j_i})[i]) \min_{j:i \in G_j} (|f_j(\cdot; w_j)[i]|) x_i, \tag{580}$$

*Then*

$$R_{F_{tq,q}}(\beta) = c_t \sum_{j=1}^k \left\|\beta_{G_j}\right\|_q^{tq} \cong \|\beta\|_{tq,q}, \tag{581}$$

*where $F_{tq,q}$ is the parameterization induced by $f_{tq,q}$ and $c_t = 1/(t^t(1-t)^{1-t})$.*

Thus, $F_{tq,q}$ induces $l_{tq,q}$ quasi-norm as induced complexity measure.

*Proof.* Let $\beta \in \mathbb{R}^n$. Suppose that $f_{tq,q}(\cdot; w) = \beta$, and $\|w\|_2^2 = R_{F_{tq,q}}(\beta)$. Then,

$$|\beta_i| = \min_{j:i \in G_j} (|f_j(\cdot; w_j)[i]|) \tag{582}$$

for each $i \in [n]$. Since $R_{F_j}(a) = R_{F_q^{tq}}(a_{G_j})$ ($a_{G_j}$ is the projection of $a$ on coordinates in $G_j$) is strictly increasing in $|a_i|$ for $i \in G_j$ by Lemma 63, and $w$ attains the minimum representation cost,

$$|f_j(\cdot; w_j)[i]| = |\beta_i|, \tag{583}$$

for all $j$ such that $i \in G_j$. Thus, by Lemma 63,

$$\|w_j\|_2^2 = R_{F_q^{tq}}(\beta_{G_j}) = \frac{1}{t^t(1-t)^{1-t}} \left\|\beta_{G_j}\right\|_q^{tq}, \tag{584}$$

where $\beta_{G_j}$ is the projection of $\beta$ on coordinates in $G_j$. Therefore,

$$R_{F_{tq,q}}(\beta) = \sum_{j=1}^k \|w_j\|_2^2 = \frac{1}{t^t(1-t)^{1-t}} \sum_{j=1}^k \left\|\beta_{G_j}\right\|_q^{tq}. \tag{585}$$

$\square$

Next, we turn to the $p > q$ case and aim to find a homogeneous parameterization that induces $l_{q,tq}$ quasi-norm as induced complexity measure. First, we recall the construction of hidden layer used to induce $l_{2,1}$ norm (section D.4) as induced complexity measure. Let $G_1, \ldots, G_k$ be a partition of $[n]$. Let

$$\mathcal{C} = \prod_{j=1}^{k} G_j = G_1 \times G_2 \times \cdots \times G_k \tag{586}$$

be the Cartesian product of the $k$ groups. For each $h \in \mathcal{C}$, let $S_h := \{h[j] : j \in [k]\}$. Let $w = (w_h)_{h \in \mathcal{C}}$ be the parameters, where $w_h \in \mathbb{R}^{k+1}$ for all $h \in \mathcal{C}$. For each $x \in \mathbb{R}^n$, let $x_{S_h}$ be the projection of $x$ on coordinates that correspond to elements in $S_h$. For each $h \in \mathcal{C}$, let $g_h(f_q^{tq}(\cdot; w_h)) \in \mathbb{R}^n$ be the vector defined as

$$g_h(f_q^{tq}(\cdot; w_h))[i] = f_q^{tq}(\cdot; w_h)[i] \tag{587}$$

if $i \in S_h$ and 0 otherwise. Now, let

$$f_{q,tq}(x; w) = \left( \left( \sum_{h \in \mathcal{C}} g_h(f_q^{tq}(\cdot; w_h))^{tq} \right)^{1/tq} \right)^T x, \tag{588}$$

where the exponents are taken component-wise(i.e $a^k[i] = a[i]^k$).

**Theorem 65.** *For $\beta \in \mathbb{R}^n$,*

$$R_{F_{q,tq}}(\beta) = c_t \|\beta\|_{q,tq}^{tq} \cong \|\beta\|_{q,tq}, \tag{589}$$

*where $c_t = 1/(t^t(1-t)^{1-t})$.*

Thus, $F_{q,tq}$ induces $l_{q,tq}$ quasi-norm as induced complexity measure.

*Proof.* By Lemma 63,

$$R_{F_{q,tq}}(\beta) = c_t \min \left\{ \sum_{h \in \mathcal{C}} \|v_h\|_q^{tq} : \mathrm{supp}(v_h) \subseteq S_h, \left( \sum_{h \in \mathcal{C}} v_h^{tq} \right)^{1/tq} = \beta \right\}, \tag{590}$$

where $v_h = g_h(f_q^{tq}(\cdot; w_h))$ and $c_t = 1/(t^t(1-t)^{1-t})$. Let $\phi : \mathbb{R}^n \longrightarrow \mathbb{R}^n$ be defined as

$$\phi(x)[i] = x[i]^{2/d}. \tag{591}$$

Then

$$\|v_h\|_q^{tq} = \left( \sum_{i=1}^{n} v_h[i]^q \right)^t = \left( \sum_{i=1}^{n} \phi(v_h)[i]^{1/t} \right)^t = \|\phi(v_h)\|_{1/t}. \tag{592}$$

Also, note that

$$\sum_{h \in \mathcal{C}} \phi(v_h) = \phi(\beta) \tag{593}$$

if and only if

$$\left( \sum_{h \in \mathcal{C}} v_h^{tq} \right)^{1/tq} = \beta. \tag{594}$$

Let $g(\phi(\beta)) = R_{F_{q,tq}}(\beta)/c_t$. Then

$$g(\phi(\beta)) = \min \left\{ \sum_{h \in \mathcal{C}} \|\phi(v_h)\|_{1/t} : \mathrm{supp}(v_h) \subseteq S_h, \sum_{h \in \mathcal{C}} \phi(v_h) = \phi(\beta) \right\}. \tag{595}$$

Note that $g$ is a norm since $1/t > 1$. Let $g^*$ denote its dual norm. Let $t^* > 0$ be the conjugate of $1/t$, that is

$$t + \frac{1}{t^*} = 1. \tag{596}$$

By the same arguments in Lemma 6 and Theorem 34, we have

$$g^*(\phi(\beta)) = \max\left\{\left(\sum_{i \in S_h} |\phi(\beta)[i]|^{t^*}\right)^{1/t^*} : h \in \mathcal{C}\right\}$$

$$= \left(\sum_{j=1}^{k}\left(\max_{i \in G_j}|\phi(\beta)[i]|\right)^{t^*}\right)^{1/t*} \tag{597}$$

$$= \|\phi(\beta)\|_{t^*,\infty}.$$

Thus,

$$g(\phi(\beta)) = \|\phi(\beta)\|_{1/t,1}. \tag{598}$$

Then

$$R_{F_{q,tq}}(\beta) = c_t g(\phi(\beta)) = c_t\|\phi(\beta)\|_{1/t,1} = c_t\left(\sum_{j=1}^{k}\left(\sum_{i \in G_j}|\phi(\beta)[i]|\right)^{1/t}\right)^{t}$$

$$= c_t\left(\sum_{j=1}^{k}\left(\sum_{i \in G_j}|\beta_i|^{tq}\right)^{1/t}\right)^{t} = c_t\|\beta\|_{q,tq}^{tq}. \tag{599}$$

$\square$

To summarize, we can induce all $l_{p,q}$ group quasi-norms with overlapping between groups as induced complexity measure using homogeneous parameterizations for all $p$ and $q$. In particular, we can induce all $l_{p,q}$ group quasi-norms without overlapping between groups as induced complexity measure using homogeneous parameterizations for all $p$ and $q$.