# OpenReview forum: "Representation Costs of Linear Neural Networks: Analysis and Design"
_NeurIPS.cc/2021/Conference — NeurIPS 2021 Poster_

### Official Review · Reviewer_o8fi · 2021-07-12

**Rating:** 7
**Confidence:** 4

**Summary:**

The paper studies representation costs of linear networks and their induced complexity measures. First the representation costs of some standard architecture are given (fullly-connected nets, diagonal nets, convolutional nets, and residual nets), then for shallow networks a general formula for the representation cost (and its dual norm, given that the representation cost is convex) of any network is given. The second part of the paper is dedicated to the characterization of which complexity measures can appear from the representation cost of a neural network. Though this question is not answered in general, the authors study a number of related questions: a characterization of which $l_p$ can be represented by a neural network is given, then some example architectures are given to obtain group $l_{p,q}$ norms or $k$-support norms, then it is shown that $l_{p,q}$ group norm with overlapping groups and elastic nets cannot be recovered by a neural network representation cost (though it is possible for the representation cost of a general homogeneous parametrization).

**Limitations And Societal Impact:**

This paper is theoretical and as such has no direct societal impact.

**Main Review:**

This paper gives a quite complete overview of the representation costs induced by linear networks. It is well-written though the statements of some Lemmas/Theorems (on finite filter width conv nets, the description of the grouping layers for example) are quite technical but no intuition/explanation is given, to help understand them (I imagine that this is due to a lack of space: this paper is made up of a large number of results, and there is not enough room to discuss the smaller ones).

The results of Section 3 are mostly slight generalizations/improvements of known results, as noted by the authors. The description of the "residual nets" is new though if I understand well, it represent the case of multiple parallel networks which is quite different to the traditional residual nets, so maybe it would be best to not call it residual networks. While this section lacks novelty it offers a good overview.

Section 4 contains more novel results: to my knowledge the description of architectures leading to the $l_p$ norm, group $l_{p,q}$ norm and $k$-support norm are new. The discussion of negative results (norms that cannot be represented by a linear network) is also new and important to obtain a complete characterization of the complexity measures induced by DNNs, though this paper is only a first step in this direction.

Overall, the strength of this paper is that it gives a good overview of the complexity measures induced by linear networks, what types of measures can and cannot be represented. Recent results have shown the important role that the representation cost of neural network plays, and this paper is a good step towards a characterization of the complexity measures induced by linear networks.

The weakness of this paper is that it is a collection of a many results, some of them small or not so novel and maybe some of the results could be dropped (for example the discussion of finite filter width conv nets is quite long and technical, yet it did not give me any particular insight on the effect of the finite width of the filter).

**Time Spent Reviewing:**

5

---

> ### Author Response · Authors · 2021-08-11
> **ResNet Definition**
>
> Thanks for the review!
>
> Our current definition of ResNet is indeed not clear now. We will modify the definition to make it clear in a new version of this paper. Roughly speaking, we define the ResNet as a sum over several subnetworks. Each of the subnetworks only passes through some of the layers and skip the rest. One example is the network F_N(w_1, w_2)(x) = w_2^T x + w_2^T diag(w_1) x, where N is the ResNet, w_1, w_2 are the weights in the first and second layers and x is the input. Here, the first layer is diagonal and the second layer is fully connected. Also, there are two subnetworks in this ResNet. One of the subnetworks skips the first layer and the other one goes through both layers. The important thing is that the weights in the second layer w_2 are shared for both subnetworks. So this model is not multiple parallel networks. Sorry for the confusion!

---

### Official Review · Reviewer_ZwDT · 2021-07-15

**Rating:** 7
**Confidence:** 3

**Summary:**

This paper studies the representation cost in the setting of linear networks. It is shown that an l^2 regularization on the parameters corresponds to adding a penalty on the functional space governed by the representation cost. This paper shows how the representation cost depends on the architecture and how the bias depends on the representation cost.

**Ethical Concerns:**

No ethical concerns

**Limitations And Societal Impact:**

No societal impact

**Main Review:**

This is an interesting paper. The dictionaries between architectures and complexity measures are interesting. In particular, we see that certain penalty functions such as elastic net cannot be viewed as representation costs of certain architectures.

**Time Spent Reviewing:**

1

---

> ### Author Response · Authors · 2021-08-11
> **Thanks**
>
> Thanks for your appreciation for this paper!

---

### Official Review · Reviewer_n4py · 2021-07-24

**Rating:** 6
**Confidence:** 3

**Summary:**

This paper investigates the correspondence between the architecture of a linear neural network and the complexity measure induced by putting $l_2$ regularization on the weights. On one hand, the authors study the complexity measure induced by fully-connected networks, diagonal networks, convolutional networks with full or limited filter width, as well as residual networks. On the other hand, the authors also present the architectures that induce $l_p$, $l_{p, q}$ as well as $k$-support norms as complexity measures.

**Limitations And Societal Impact:**

Yes.

**Main Review:**

To my knowledge, this work is a novel and nice contribution that extends previous work on the norms corresponding to linear neural networks with simpler architectures. The results are both mathematically interesting and potentially helpful for better understanding the implicit regularizations in linear neural networks. The writing is mostly clear.

In addition to the examples given on page 5, it could be helpful to give more examples of linear functions (matrices) whose relative complexities depend on what architectures we choose, e.g. convolutional versus fully-connected versus diagonal.

Since the authors refer to [1] as a prior work that studies similar linear neural network models with single outputs or full convolutional filter widths, could the authors comment on how similar the techniques are between the two papers?

Are there scenarios where grouping layers may be interesting practically?


Reference: 1. Suriya Gunasekar, Jason Lee, Daniel Soudry, and Nathan Srebro. Implicit bias of gradient descent on linear convolutional networks.

**Time Spent Reviewing:**

3

---

> ### Author Response · Authors · 2021-08-11
> **Technique and example**
>
> Thanks for the review.
>
> To answer your question about [1]: The technique we used in Proposition 2 and Proposition 3 is similar to the one used in [1] in the sense that we both reduce the case of linear convolutional neural networks with full filter width to the case of linear diagonal neural networks. The approach we used for convolutional neural network with finite filter width (i.e. Theorem 4, Corollary 5 and Theorem 6) is only loosely related to [1] in the sense that we used Discrete Fourier Transform as a first step. The technique we used for other results in this paper is different from [1].
>
> We are not aware of any application of grouping layers in practice since this is new to our work to the best of our knowledge. However, we think it can be useful in practice since grouped l_pq norms have been used as regularizers in many areas such as transfer learning. Indeed, we are actually thinking of applying these grouping layers to some transfer learning problems.
>
> Another example of a vector/linear predictor whose relative complexities depend on what architecture we choose are the vectors b = (1,1,0,0) and  b’ = (1,0,0,1). Say we have two groups of the indices/coordinates: (1,2) and (3,4). Then ||b||_{1,2} = \sqrt(2) < 2 = ||b’||_{1,2} whereas ||b’||_{2,1} = \sqrt(2) < 2 = ||b||_{2,1}. Let N_1 be an architecture that induces l_1,2 norm and N_2 be an architecture that induces l_2,1 norm. Then R_{N_1}(b) < R_{N_1}(b’) whereas R_{N_2}(b) > R_{N_2}(b’). This example shows that N_1 is biased toward vectors which are concentrated in a small number of groups whereas N_2 is biased toward vectors which are more “spread out” over the different groups.
>
> Reference:
> [1]. Suriya Gunasekar, Jason Lee, Daniel Soudry, and Nathan Srebro. Implicit bias of gradient descent on linear convolutional networks.

---

> > ### Comment · Reviewer_n4py · 2021-09-01
> > **Follow-up**
> >
> > Thanks for your helpful response. The connection between the grouping layers and transfer learning sounds quite interesting to explore further. I intend to keep my score, and would be happy to see this paper accepted.

---

### Official Review · Reviewer_GxMP · 2021-07-26

**Rating:** 8
**Confidence:** 4

**Summary:**

This paper studies the representation cost under different neural network models and the design of parameterization that leads to $l_2$ regularization.

**Limitations And Societal Impact:**

As described in the main review.

**Main Review:**

This is a well-written paper with solid theoretical contributions. It first analyzes the representation cost (proposed in [1,2,3]) of different neural network models:

1) fully connected network (Prop. 1 in Sec. 3.1.3);
2) diagonal network (Prop. 2 in Sec. 3.1.2);
3) convolutional network (Prop. 3 in Sec. 3.1.3);
4) linear convolutional network (Thm. 4-6 in Sec. 3.2);
5) linear non-homogeneous residual network (Thm. 7 in Sec. 3.3);
6) depth 2 network (Lemma 8-9 in Sec. 3.4).

Then the paper studies the design of parameterization that leads to certain regulations:

1) $l_p$ norm (Thm. 10 and Prop. 11 in Sec 4.2);
2) $l_{p,q}$ (Thm. 12 and 13 in Sec. 4.3);
3) $k-$support norms (Thm. 14 in Sec 4.4);
4) some other induced complexity measures (Sec. 4.5-4.6).

Comments:
The contents of this work are very solid and intense, it would have been great if the authors can spend more effort to build better connections between the results in different models. For example, how do the results in section 3 connect with each other in terms of assumptions/representation loss, perhaps the authors can provide a table or some experiments for better presentation.

Questions:
How does the result in section 3.4 relate to the result in section 3.1? It seems that the results in section 3.1 have already characterized the cases for general homogeneous networks with widths larger than 2, why do we need extra results on $d=2$?

citation:

[1] Suriya Gunasekar, Jason Lee, Daniel Soudry, and Nathan Srebro. Implicit bias of gradient descent on linear convolutional networks. arXiv preprint arXiv:1806.00468, 2018.

[2] Greg Ongie, Rebecca Willett, Daniel Soudry, and Nathan Srebro. A function space view of bounded norm infinite width relu nets: The multivariate case. arXiv preprint arXiv:1910.01635, 2019.

[3] Pedro Savarese, Itay Evron, Daniel Soudry, and Nathan Srebro. How do infinite width bounded norm networks look in function space? pages 2667–2690, 2019.

**Time Spent Reviewing:**

7 hours

---

> ### Author Response · Authors · 2021-08-11
> **Relation between results in section 3.1 and section 3.4**
>
>     Thanks for the review and encouraging comments!
>
>     We will work on adding illustrative experiments to help understand these connections better.  We do provide a table on top of page 2 of the paper which summarizes the results in this paper (specific suggestions about the table would be appreciated) but we agree with your comments that the presentation still has much room for improvement.
>
>     To answer your question: In section 3.1, we only characterize the representation cost of fully connected networks while in section 3.4, we consider networks with arbitrary architecture of depth two. Fully connected networks of depth two are both a special case for the results in section 3.4 and a special case for the result in section 3.1.

---

### Decision · Program_Chairs · 2021-09-28

**Decision:**

Accept (Poster)

**Comment:**

The paper gives a fairly complete list of results covering the representation cost for linear networks of many different architectures. The paper then studies how to design parametrization to encourage certain regularization effects. The reviewers find the results to be a solid theoretical contribution to the understanding of different parametrization.

**Consistency Experiment:**

NeurIPS has a long history of experimentation. In 2014, NeurIPS ran an experiment in which 10% of submissions were reviewed by two independent committees to quantify the randomness in the review process. This year, we repeated a variant of this experiment to see how the quality of the review process has changed over time.  This paper was part of the experiment and was therefore assigned to two committees (consisting of reviewers, an Area Chair, and a Senior Area Chair) that reached independent decisions.  If both committees made the same recommendation, this recommendation was followed. If a single committee recommended acceptance, the paper was accepted (with the exception of a few cases in which the other committee identified what we considered a fatal flaw, e.g., an error in a key result).

This copy’s committee reached the following decision: **Accept (Poster)**

The other committee assigned to the paper recommended **Reject**.  You can find the other set of reviews, along with any follow up discussion with the authors here:
https://openreview.net/forum?id=3oQyjABdbC8